# On the Global Convergence Rates of Decentralized Softmax Gradient Play in Markov Potential Games

**Runyu Zhang**
Harvard University
runyuzhang@fas.harvard.edu

**Jincheng Mei**
Google Research, Brain Team
jcmei@google.com

**Bo Dai**
Google Research, Brain Team
bodai@google.com

**Dale Schuurmans**
University of Alberta
Google Research, Brain Team
schuurmans@google.com

**Na Li**
Harvard University
nali@seas.harvard.edu

## Abstract

Softmax policy gradient is a popular algorithm for policy optimization in single-agent reinforcement learning, particularly since projection is not needed for each gradient update. However, in multi-agent systems, the lack of central coordination introduces significant additional difficulties in the convergence analysis. Even for a stochastic game with identical interest, there can be multiple Nash Equilibria (NEs), which disables proof techniques that rely on the existence of a unique global optimum. Moreover, the softmax parameterization introduces non-NE policies with zero gradient, making it difficult for gradient-based algorithms in seeking NEs. In this paper, we study the finite time convergence of decentralized softmax gradient play in a special form of game, Markov Potential Games (MPGs), which includes the identical interest game as a special case. We investigate both gradient play and natural gradient play, with and without $\log$-barrier regularization. The established convergence rates for the unregularized cases contain a trajectory dependent constant that can be *arbitrarily large*, whereas the $\log$-barrier regularization overcomes this drawback, with the cost of slightly worse dependence on other factors such as the action set size. An empirical study on an identical interest matrix game confirms the theoretical findings.

## 1 Introduction

Multi-agent systems encounter vast application in real world scenarios, such as network routing [35, 8], social and economic decision making [36, 30], and robotic swarms [22, 14]. In these problems, a system consists of a group of agents interacting in a shared environment. Given the recent success of reinforcement learning (RL), increasing attention has been drawn to the possibility of applying RL algorithms, such as policy gradient, to multi-agent systems. However, the theoretical foundations for multi-agent reinforcement learning (MARL) remain limited. Unlike single-agent RL, the actions of other agents affect the dynamics and the decision making outcome for each individual in the system, raising additional theoretical challenges when analyzing joint performance.

The stochastic game (SG) is a classical multi-agent model that has received extensive attention in recent MARL studies. In a stochastic game, the environment is represented by a state space that evolves based on the joint actions of agents. Each agent in a stochastic game tries to maximize its own total reward by making decisions *independently*, based on state information shared between agents. The stochastic game model was first introduced in [31], with a series of followup works proposing NE-seeking algorithms, particularly in the RL setting (e.g. [21, 5, 32, 6, 17, 38] and citations therein). Given recent progress in the underlying theory of RL, many recent works have investigated finite

time iteration and sample complexity for learning NE or other general equilibria notions, such as correlated and coarse correlated equilibria (e.g. [33]).

There are different types of SGs, some with attributes that merit special attention; for example, two-player zero sum games [4, 9], which are widely used to model two player competitive games such as GO. In this paper, we will focus on another type of SG, the Markov potential game (MPG) [23, 27, 39, 18], which includes the identical interest game as a special case. The structure of a MPG enables efficient learning through the use of gradient-based algorithms such as gradient play. Recent work [39, 18] has focused on the iteration and sample complexity of finding a NE in an MPG under the *direct* policy parameterization, which is not practical in most real world scenarios, given the cost of projecting back to the probability simplex on every iteration. This drawback has motivated consideration of the *softmax* parameterization, which bypasses the projection step in the gradient update, and is perhaps the most popular approach to parameterizing policies in practice. [11] have studied natural gradient play for MPG under softmax parameterization, but only address asymptotic behavior and leave finite time complexity open.

From the perspective of analysis and practical performance, the extension from the direct to the softmax parameterization in policies is nontrivial. Even in the single agent case, as shown by [2, 24], there are policies in the softmax parameterization that have near-zero gradient and yet are far from being globally optimal, which creates difficulty for a gradient-based algorithm to escape suboptimal points. A similar issue exists for MPGs: due to the more complex interaction between agents, there is even a greater set of policies that obtain small gradient norm but are far from a NE. Based on our analysis and numerical results, even for natural gradient play—which is known to enjoy dimension free convergence in single agent learning [2]—we find in the multiagent setting that it can still become stuck in these undesirable regions. Such evidence suggests that preconditioning according to the Fisher information matrix [29, 3] is not sufficient to ensure fast convergence in multi-agent learning. A stronger form of regularization is required, which motivates the introduction of $\log$-barrier regularization to avoid undesirable regions of policy space.

| Algorithm | Single-agent MDP | Multi-agent MPG |
|---|---|---|
| Gradient play, direct parameterization | $O\left(\frac{\|\mathcal{A}\|M^2}{(1-\gamma)^4\epsilon^2}\right)$ [2] | $O\left(\frac{(\phi_{\max}-\phi_{\min})\sum_{i=1}^n \|\mathcal{A}_i\|M^2}{(1-\gamma)^4\epsilon^2}\right)$ [39, 18] |
| Gradient play, softmax parameterization | $O\left(\frac{M^2}{(1-\gamma)^3 c^2\epsilon}\right)$ [24] | $\mathbf{O\left(\frac{n\max_i \|\mathcal{A}_i\|(\phi_{\max}-\phi_{\min})M^2}{(1-\gamma)^4 c^2\epsilon^2}\right)}^*$ |
| Natural gradient play, softmax parameterization | $O\left(\frac{1}{(1-\gamma)^2\epsilon}\right)$ [2] | $\mathbf{O\left(\frac{n(\phi_{\max}-\phi_{\min})^2 M}{(1-\gamma)^3 c\epsilon^2}\right)}^*$ |
| Gradient play + $\log$-barrier reg., softmax parameterization | $O\left(\frac{\|\mathcal{A}\|^2 M^2}{(1-\gamma)^4\epsilon^2}\right)$ [2] | $\mathbf{O\left(\frac{n\max \|\mathcal{A}_i\|^2(\phi_{\max}-\phi_{\min})M^2}{(1-\gamma)^4\epsilon^2}\right)}$ |
| Natural gradient play + $\log$-barrier reg, softmax parameterization | Unknown | $\mathbf{O\left(\frac{n\max_i \|\mathcal{A}_i\|(\phi_{\max}-\phi_{\min})M^2}{(1-\gamma)^4\epsilon^2}\right)}$ |

Table 1: Summary of known convergence rate results for gradient based methods in Markov decision processes (MDPs) and MPGs respectively. The new results proved in this paper for MPGs are displayed in bold font. Complexity bounds with '*' depend on an additional assumption on the MPG (See Theorem 4 and 5). The definitions of variables $M$ and $c$ appearing in some bounds can be found in (5) and (11). *Note that the definition of $M$ is slightly different from the "distribution mismatch coefficient" $D_\infty$ defined in [2] (see more details in descriptions that follows Assumption 1). To make the complexity results more comparable, we slightly modify and re-derive the results in [2, 24, 39, 18].*

**Our contribution:** In this paper, we provide *finite time* iteration complexity results for gradient play and natural gradient play under the softmax parameterization, considering both unregularized and $\log$-barrier regularized dynamics. We summarize the convergence rates and compare them to existing results for the direct parameterization and to the corresponding single agent cases in Table 1. These findings suggest that regularization is crucial for obtaining fast convergence to a NE under the softmax parameterization in a MPG. In Table 1, the results for the two unregularized algorithms in the multi-agent case rely on the assumption that the set of stationary policies is isolated (which is also assumed in [11] when establishing the asymptotical convergence for natural policy gradient), and the corresponding complexity bounds contain an initialization dependent factor $c$. By contrast, the $\log$-barrier regularized algorithms overcome both drawbacks, but as a tradeoff, their bounds

incur a slightly worse dependence on $|\mathcal{A}_i|$ and $M$. We observe numerically that the log-barrier regularized algorithms are indeed more robust against becoming trapped near undesirable non-NE stationary points. To the best of our knowledge, the finite-time iteration complexity results are the first such results for MPGs under the softmax parameterization. Though the analysis for the gradient play follows their single-agent counterparts [2, 24], the results for natural gradient play are highly non-trivial, requiring very different analysis tools which have their own merits to the literature (see Remark 2 and Remark 3 for more details on the technical novelty in the analysis). Our results also convey the following two messages. First, finding the NE of a multi-agent MPG is harder than finding the global optimum for the single-agent case, because multi-agent learning suffers greater risk of becoming trapped near undesirable stationary points. This is reflected in the dependence of the complexity bounds on $\epsilon$ in Table 1. Second, natural gradient play outperforms gradient play counterparts, suggesting that natural gradient play captures useful information about the geometry of the parameter space that accelerates the learning process.

## 2 Problem settings

We consider an infinite time horizon $n$-agent stochastic game (SG, [31]) $\mathcal{M} = (N, \mathcal{S}, \mathcal{A} = \mathcal{A}_1 \times \ldots \times \mathcal{A}_n, P, r = (r_1, \ldots, r_n), \gamma, \rho)$ which is specified by an agent set $N = \{1, 2, \ldots, n\}$, a finite state space $\mathcal{S}$, a finite action space $\mathcal{A}_i$ for each agent $i \in N$, a transition model $P$ (such that $P(s'|s, a) = P(s'|s, a_1, \ldots, a_n)$ is the probability of transitioning into state $s'$ upon taking action $a := (a_1, \ldots, a_n)$ in state $s$ where $a_i \in \mathcal{A}_i$ is action of agent $i$), a reward function $r_i : \mathcal{S} \times \mathcal{A} \to [0, 1]$ for each agent $i$, a discount factor $\gamma \in [0, 1)$, and an initial state distribution $\rho$ over $\mathcal{S}$. We use $s(t) \in \mathcal{S}$ to denote the state at time step $t$, and $a(t) = (a_1(t), \ldots, a_n(t)) \in \mathcal{A}$ to denote the total action.

A stochastic policy $\pi : \mathcal{S} \to \Delta(\mathcal{A})$ (where $\Delta(\mathcal{A})$ is the probability simplex over $\mathcal{A}$) specifies a strategy, where agents choose their actions *jointly* based on the current state in a stochastic fashion; i.e. $\Pr(a(t)|s(t)) = \pi(a(t)|s(t))$. A *decentralized* stochastic policy is a special subclass of stochastic policies with $\pi = \pi_1 \times \ldots \times \pi_n$, such that $\pi_i : \mathcal{S} \to \Delta(\mathcal{A}_i)$, where $\pi_i$ is agent $i$'s own local policy. For decentralized stochastic policies, each agent takes its action based on the current state $s$ *independently of* other agents' action choices; i.e.,

$$\Pr(a(t)|s(t)) = \pi(a(t)|s(t)) = \prod_{i=1}^{n} \pi_i(a_i(t)|s(t)).$$

For notation simplicity, we define $\pi_I(a_I|s) := \prod_{i \in I} \pi_i(a_i|s)$, where $I \subseteq N$ is an index set. Further, we use the notation $-i$ to denote the index set $N \backslash \{i\}$. In this paper we focus on tabular softmax parameterization for a policy, where policy $\pi_\theta = (\pi_{\theta_1}, \ldots, \pi_{\theta_n})$ is parameterized by a set of parameters $\theta = (\theta_1, \ldots, \theta_n)$, with $\theta_i = \{\theta_{s,a_i}\}_{s \in \mathcal{S}, a_i \in \mathcal{A}_i}$, and where

$$\pi_{\theta_i}(a_i|s) = \frac{\exp(\theta_{s,a_i})}{\sum_{a_i'} \exp(\theta_{s,a_i'})}. \tag{1}$$

We denote agent $i$'s total reward starting from initial states $s(0) \sim \rho$ as: $J_i(\theta) := \mathbb{E}_{s(0) \sim \rho} \left[ \sum_{t=0}^{\infty} \gamma^t r_i(s(t), a(t)) \big| \pi_\theta, s(0) = s \right]$. Agent $i$'s objective is to maximize its own total reward $J_i$. A Nash equilibrium (NE) is often used to characterize the equilibrium (a joint policy) where no agent has a unilateral incentive to deviate from it.

**Definition 1.** *(Nash equilibrium) A policy $\theta^* = (\theta_1^*, \ldots, \theta_n^*)$ is called a (Markov perfect) Nash equilibrium (NE) if*

$$J_i(\theta_i^*, \theta_{-i}^*) \geq J_i(\theta_i', \theta_{-i}^*), \quad \forall \theta_i', \quad i \in N \tag{2}$$

*Further, we define the 'NE-gap' of a policy $\theta$ to be:*

$$\texttt{NE-gap}_i(\theta) := \sup_{\theta_i'} J_i(\theta_i', \theta_{-i}) - J_i(\theta_i, \theta_{-i}); \quad \texttt{NE-gap}(\theta) := \max_i \texttt{NE-gap}_i(\theta).$$

*A policy $\theta$ is an $\epsilon$-Nash equilibrium if:* $\texttt{NE-gap}(\theta) \leq \epsilon$.

We define the value function with respect to stage cost $r_i$ as:

$$V_i^\theta(s) := \mathbb{E}\left[ \sum_{t=0}^{\infty} \gamma^t r_i(s(t), a(t)) \big| \pi_\theta, s(0) = s \right].$$

We define agent $i$'s $Q$-function and advantage function $Q_i^\theta, A_i^\theta : \mathcal{S} \times \mathcal{A} \to \mathbb{R}$,

$$Q_i^\theta(s, a) := \mathbb{E}\left[ \sum_{t=0}^{\infty} \gamma^t r_i(s(t), a(t)) \big| \pi_\theta, s(0) = s, a(0) = a \right], \quad A_i^\theta(s, a) := Q_i^\theta(s, a) - V_i^\theta(s).$$

We further define agent $i$'s *'averaged' Q-function* $\overline{Q_i^\theta} : \mathcal{S} \times \mathcal{A}_i \to \mathbb{R}$ and *'averaged' advantage-function* $\overline{A_i^\theta} : \mathcal{S} \times \mathcal{A}_i \to \mathbb{R}$ as:

$$\overline{Q_i^\theta}(s, a_i) := \sum_{a_{-i}} \pi_{\theta_{-i}}(a_{-i}|s) Q_i^\theta(s, a_i, a_{-i}), \quad \overline{A_i^\theta}(s, a_i) := \sum_{a_{-i}} \pi_{\theta_{-i}}(a_{-i}|s) A_i^\theta(s, a_i, a_{-i}).$$

Finally, define the *discounted state visitation distribution* $d_\theta$ of a policy $\pi_\theta$ given an initial state distribution $\rho$ as:

$$d_\theta(s) := \mathbb{E}_{s(0) \sim \rho} (1 - \gamma) \sum_{t=0}^\infty \gamma^t \mathrm{Pr}^\theta(s(t) = s|s(0)), \tag{3}$$

where $\mathrm{Pr}^\theta(s(t) = s|s(0))$ is the state visitation probability that $s(t) = s$ when executing $\pi_\theta$ starting at state $s(0)$. From the policy gradient theorem [34], we have that (proof given in Appendix B):

$$\frac{\partial J_i(\theta)}{\partial \theta_{s,a_i}} = \frac{1}{1-\gamma} d_\theta(s) \pi_{\theta_i}(a_i|s) \overline{A_i^\theta}(s, a_i). \tag{4}$$

For the remainder of the paper, we make the following assumptions on the stochastic games we study.

**Assumption 1.** *The stochastic game $\mathcal{M}$ satisfies:* $\inf_\theta \min_{s \in \mathcal{S}} d_\theta(s) > 0$.

Assumption 1 requires that every state is visited with positive probability for any policy, which is a standard assumption for convergence proofs in the RL literature (e.g. [2, 24]). We will use $M$ to denote the following quantity

$$M := \sup_\theta \max_s \frac{1}{d_\theta(s)}. \tag{5}$$

Note that $M$ can be viewed as a measure of exploration sufficiency in the stochastic game, which is slightly different from the "distributional mismatch coefficient" introduced in [2] defined by $\sup_{\theta,\theta'} \max_s \frac{d_{\theta'}(s)}{d_\theta(s)}$; however, both can be upper bounded by $\max_s \frac{1}{(1-\gamma)\rho(s)}$.

We primarily focus on the following subclass of stochastic games in this paper:

**Definition 2.** *A stochastic game is called a Markov potential game (MPG, [37, 23, 39, 18, 26]) if there exists a potential function $\phi : \mathcal{S} \times \mathcal{A}_1 \times \cdots \times \mathcal{A}_n \to \mathbb{R}$ such that for any agent $i$ and any pair of policy parameters $(\theta_i', \theta_{-i}), (\theta_i, \theta_{-i})$ :*

$$\mathbb{E}\left[\sum_{t=0}^\infty \gamma^t r_i(s(t), a(t)) \Big| \pi = (\theta_i', \theta_{-i}), s(0) = s\right] - \mathbb{E}\left[\sum_{t=0}^\infty \gamma^t r_i(s(t), a(t)) \Big| \pi = (\theta_i, \theta_{-i}), s(0) = s\right]$$
$$= \mathbb{E}\left[\sum_{t=0}^\infty \gamma^t \phi(s(t), a(t)) \Big| \pi = (\theta_i', \theta_{-i}), s(0) = s\right] - \mathbb{E}\left[\sum_{t=0}^\infty \gamma^t \phi(s(t), a(t)) \Big| \pi = (\theta_i, \theta_{-i}), s(0) = s\right], \forall s. \tag{6}$$

Without loss of generality, we assume that $\phi_{\min} \le \phi(s, a) \le \phi_{\max}$ for all $(s, a)$. The definition of MPG is a generalization of the notion potential game in the one-shot setting [28]. Note that identical reward game where agents share a same reward function naturally satisfies the above condition and serves as one important special case of MPG. For non-identical reward settings, [23, 12] found that continuous MPGs can model applications such as the great fish war [19], the stochastic lake game [10], medium access control [23] etc. For tablular MPGs, [39, 18] also discuss necessary/sufficient conditions that implies a MPG, as well as its application and counterexamples.

Given a MPG, we define the *total potential function* $\Phi$ as:

$$\Phi(\theta) := \mathbb{E}_{s(0) \sim \rho} \left[ \sum_{t=0}^\infty \gamma^t \phi(s(t), a(t)) \Big| \pi_\theta, s(0) = s \right].$$

Given the property in (6), it is straightforward to verify that the NE condition (2) is equivalent to $\Phi(\theta_i^*, \theta_{-i}^*) \ge \Phi(\theta_i', \theta_{-i}^*), \forall \theta_i', i \in N$ and that for the policy gradient, $\frac{\partial J_i(\theta)}{\partial \theta_{s,a_i}} = \frac{\partial \Phi(\theta)}{\partial \theta_{s,a_i}}$ for all $i, s, a_i$.

*Remark* 1 (**Differences between MPG and single-agent/centralized MDP**). Because of the existence of the total potential function $\Phi$, it is natural to ask whether MPG renders the multi-agent policy gradient similar to single agent policy gradient and thus results and analysis tools developed for single agent policy gradient in e.g., [2, 24] would be easily extended to the multiagent case. Unfortunately, this is not the case. To illustrate how it differs from single-agent/centralized case, we can focus on the special type of MPGs where every agent has the same reward function, namely the identical interest case. In the single agent/centralized case, there is a unique global optimal solution which corresponds to the convergent stationary policy. However, in the multiagent case, even if the rewards are identical, because the policy is *decentralized*, i.e., agents taking *independent* policies $\pi := \pi_1 \times \ldots \times \pi_n$, we loose the connection between stationary policies and optimal policies. As we shown later, the

convergent stationary policies are Nash equilibria, which are unfortunately non-unique even for the identical interest case. Moreover, a key condition that is used in establishing the convergence rate, Łojasiewicz condition (Lemma 1), is also much weaker for the multiagent case compared to single agent [24]: the left hand side is the *Nash gap* $\max_{i,\theta_i^*} \Phi(\theta_i^*, \theta_{-i}) - \Phi(\theta)$ instead of the *optimality gap* $\max_{\theta^*} \Phi(\theta^*) - \Phi(\theta)$). Note that zero Nash gap does not imply zero optimality gap, as there exists *many* NEs of *different* values. These differences disable many proof technique used for single agent case and make the analysis harder and lead to different performance results, as demonstrated in the rest of the paper.

## 3 Relationship between first order stationary point and Nash equilibrium

Before studying convergence performance of gradient play algorithms, it is important to first understand the relationship between the stationary points and the NEs. Unfortunately, equivalence cannot be established in this setting. Standard optimization theory guarantees that all NEs are stationary points, but unfortunately not vice versa. Under softmax parameterization, there exist non-NE stationary points. For example, from the gradient formulation (4), it can be shown that any non-NE deterministic policies are also stationary points. However, the notion of NE and stationarity are indeed closely related. This section aims to characterize some differences between NE and non-NE stationary points. This differentiation of the NE and non-NE stationary points is established by the non-uniform Łojasiewicz condition (also known as gradient domination) for stochastic games.

**Lemma 1.** *(Non-uniform Łojasiewicz inequality; proof given in Appendix D) Define*

$$M(\theta) := \max_s \frac{1}{d_\theta(s)}, \quad c(\theta) := \min_s \sum_{a_i^* \in \mathrm{argmax}_{a_i} \overline{Q_i^\theta}(s,a_i)} \pi_{\theta_i}(a_i^*|s). \tag{7}$$

*Then we have that*

$$\mathtt{NE\text{-}gap}_i(\theta) \leq \frac{\sqrt{|\mathcal{A}_i|}M(\theta)}{c(\theta)} \|\nabla_{\theta_i} J_i(\theta)\|_2.$$

The Łojasiewicz condition (gradient domination) implies that the NE-gap of a policy can be bounded by the norm of its gradient, whereas the term 'non-uniform' refers to the factor $\frac{\sqrt{|\mathcal{A}_i|}M(\theta)}{c(\theta)}$, which cannot be bounded uniformly for all $\theta$. The counterpart of Lemma 1 for a single-agent MDP was first introduced in [24, Lemma 8]. One major difference between Lemma 1 and [24, Lemma 8] is how $c(\theta)$ is defined. In [24], $c(\theta) := \min_s \pi_\theta(a^*(s)|s)$, where $a^*(s)$ is the optimal action on state $s$ (i.e., $a^* = \mathrm{argmax}_a Q^*(s,a)$), whereas in MPG, because there's no globally defined $Q^*$, the $a_i^*$ in (7) is chosen as the *greedy* optimal action of the *current* averaged $Q$-function (i.e., $a_i^* \in \mathrm{argmax}_{a_i} \overline{Q_i^\theta}(s,a_i)$).

Note that because $c(\theta)$ on the denominator can be zero for certain policies (e.g. one can verify that any non-NE deterministic policy have $c(\theta) = 0$), which implies that a $\theta$ with gradient norm close to zero is not necessarily near a NE. Given this observation, we could differentiate the non-NE stationary points with NEs by whether $c(\theta^*)$ equals to zero, which is formally stated in the following lemma:

**Lemma 2.** *(Proof given in Appendix D) Suppose $\theta^*$ is a stationary point, i.e. $\|\nabla\Phi(\theta^*)\| = 0$, then $\theta^*$ is a NE if and only if $c(\theta^*) = 1$, $\theta^*$ is not a NE if and only if $c(\theta^*) = 0$.*

## 4 Unregularized gradient play

We first investigate the convergence to NE for gradient and natural gradient play, respectively. Under the softmax parameterization, the two schemes are given by

$$\textit{Gradient Play:} \quad \theta_i^{(t+1)} = \theta_i^{(t)} + \eta \nabla_{\theta_i} J_i(\theta_i^{(t)}) \tag{8}$$

$$\textit{Natural Gradient Play:} \quad \theta_i^{(t+1)} = \theta_i^{(t)} + \eta F_i(\theta^{(t)})^\dagger \nabla_{\theta_i} J_i(\theta_i^{(t)}) \tag{9}$$

where † denotes the Moore-Penrose inverse and $F_i(\theta)$ is the Fisher information matrix for $\pi_{\theta_i}$:

$$F_i(\theta) := \mathbb{E}_{s \sim d_\theta(\cdot)} \mathbb{E}_{a_i \sim \pi_{\theta_i}(\cdot|s)} \left[ \nabla_{\theta_i} \log \pi_{\theta_i}(a_i|s) \nabla_{\theta_i} \log \pi_\theta(a_i|s)^\top \right].$$

For notational simplicity, we abbreviate the variables $d_{\theta^{(t)}}$, $A_i^{\theta^{(t)}}$ and $\overline{A_i^{\theta^{(t)}}}$ as $d^{(t)}$, $A_i^{(t)}$ and $\overline{A_i^{(t)}}$ respectively; and denote $\pi_{\theta^{(t)}}(a|s)$ and $\pi_{\theta_i^{(t)}}(a_i|s)$ as $\pi^{(t)}(a|s)$ and $\pi_i^{(t)}(a_i|s)$ respectively. For the softmax parameterization, we can establish the equivalence of natural gradient play and soft Q-learning [13], formally stated in the following lemma.

**Lemma 3.** *(Proof given in Appendix C) Natural gradient play is equivalent to*

$$\pi_i^{(t+1)}(a_i|s) \propto \pi_i^{(t)}(a_i|s) \exp\left(\eta \overline{A_i^{(t)}}(s, a_i)/(1-\gamma)\right) \tag{10}$$

**Asymptotic convergence to Nash Equilibrium.**    As stated in Section 3, there exist stationary points that are not NEs. It is not immediately obvious why running gradient methods can avoid converging to these points, thus before studying convergence rate to NE, it is necessary to first examine whether asymptotic convergence holds. Moreover, the asymptotic convergence result is used to establish the finite time convergence rate results later (see the subsection 4.1).

**Theorem 4.** *(Proof given in Appendix E) Suppose Assumption 1 holds and that the stationary policies are isolated, gradient play (8) with $\eta \leq \frac{(1-\gamma)^3}{6n}$ guarantees that $\lim_{t \to +\infty} \theta^{(t)} = \theta^{(\infty)}$, where $\theta^{(\infty)}$ is a NE. The same argument also holds for natural gradient play (10) with $\eta \leq \frac{(1-\gamma)^2}{2n(\phi_{\max} - \phi_{\min})}$.*

The proof of Theorem 4 resembles the technique used in [2] for the single agent case, where the additional assumption on the isolated stationary policies is introduced due to some specific technical difficulties encountered in multi-agent learning (see more discussion in Appendix E, which is also introduced in [11] for establishing the asymptotic convergence of NPG. We believe it is a conservative condition for ensuring the asymptotic convergence. It remains an interesting open question to establish convergence without this assumption.

## 4.1   Finite time convergence rate

This section considers finite time convergence rate for gradient play and natural gradient play. Corresponding results for the single-agent setting can be found in [24] (for gradient play) and [2, 16, 25] (for natural gradient play). Some aspects of these analyses can be carried over to the multi-agent MPG setting; however, as will be discussed later, there are several fundamental differences that make the multi-agent case more challenging.

Our convergence results rely on the observation from Section 3 and the asymptotic convergence to NE. Combining Theorem 4 and Lemma 2, we know that $c(\theta^{(t)})$ asymptotically converges to 1 for (natural) gradient play, and since $c(\theta^{(t)}) > 0$ for any softmax policy (because $\pi_{\theta_i}(a_i|s) > 0$),

$$c := \inf_t c(\theta^{(t)}) > 0. \tag{11}$$

We are now ready to give formal convergence rates for gradient and natural gradient play respectively.

**Theorem 5.** *(Gradient play and natural gradient play; proof given in F) Suppose Assumption 1 holds and that the stationary policies are isolated, gradient play (8) with $\eta = \frac{(1-\gamma)^3}{6n}$ will guarantee that for all $T$,*

$$\frac{\sum_{t=0}^{T-1} \mathtt{NE\text{-}gap}(\theta^{(t)})^2}{T} \lesssim O\left(\frac{n \max_i |\mathcal{A}_i|(\phi_{\max} - \phi_{\min})M^2}{(1-\gamma)^4 c^2 T}\right), \tag{12}$$

*Natural gradient play (10) with $\eta = \frac{(1-\gamma)^2}{2n(\phi_{\max} - \phi_{\min})}$ will guarantee that for all $T$,*

$$\frac{\sum_{t=0}^{T-1} \mathtt{NE\text{-}gap}(\theta^{(t)})^2}{T} \lesssim O\left(\frac{n(\phi_{\max} - \phi_{\min})^2 M}{(1-\gamma)^3 c T}\right). \tag{13}$$

*Here $O(\cdot)$ hides constant factors, $M$ and $c$ are defined as in (5) and (11), respectively.*

*Remark 2* (**Proof sketch and novelty**).  The proof for gradient play is relatively straightforward from the non-uniform Łojasiewicz inequality and standard non-convex optimization results, which we refer readers to the appendix for more details. However, the proof for natural gradient play is more involved and existing analysis on NPG cannot be generalized to this setting. For single-agent MDP, the analysis on NPG leverages the unique existence of optimal value function $V^*$ so that similar analysis for mirror-descent can also carry over to NPG analysis, and thus obtain dimension

free convergence. However, in the multi-agent setting, there's no well-defined $V^*$ as NEs can be non-unique with different potential values, thus, we need to further deploy additional structures of the total potential function $\Phi$. Our analysis rely on the sufficient ascent lemma (Lemma 20) that lower bounds the ascent amount $\Phi(\theta^{(t+1)}) - \Phi(\theta^{(t)})$ for each natural gradient step (we would like to further note that this sufficient ascent lemma cannot be trivially obtained by the smoothness of $\Phi$). Then, we further lower bound the ascent amount in terms of NE-gap (Lemma 21). Lastly, the theorem follows by conducting standard telescoping techniques.

**Discussion on $\frac{1}{c}$:** The complexity results in Theorem 5 both depend on $\frac{1}{c}$. However, this term can become arbitrarily large. In fact, [20] show that $c$ can be exponentially small in terms of the number of states $|\mathcal{S}|$ for a general finite MDP, even under uniform initialization, hence convergence can be very slow. This conclusion is also confirmed by numerical evidence. As pointed out by [24], even for single agent settings, policy gradient can get stuck at regions with small gradient yet far from being global optimal. Similar or even worse phenomena can be observed for multi-agent MPG, as shown in Figure 1(a)-(c): even for a single state game ($|\mathcal{S}| = 1$) with uniform initialization, unregularized gradient based algorithms can still enter regions with a relatively large NE-gap while the gradient norm and $c(\theta)$ are close to zero.

**More comparison with learning for single-agent MDP:** For gradient play, we have established an iteration complexity of $O\left(\frac{n \max_i |\mathcal{A}_i|(\phi_{\max} - \phi_{\min})M^2}{(1-\gamma)^4 c^2 \epsilon^2}\right)$ to find an $\epsilon$-NE, whereas [24] show a complexity of $O\left(\frac{(\phi_{\max} - \phi_{\min})M^2}{(1-\gamma)^4 c^2 \epsilon}\right)$ to reach an $\epsilon$-global optimum for policy gradient in a single agent MDP. The dependence on $\frac{1}{\epsilon}$ is better in the single agent case because of the existence of a global optimal policy $\pi^*$ and optimal total reward $V^*$, which justify the definition of optimality gap $\delta_t = V(\theta^{(t)}) - V^*$. This, combined with the non-uniform Łojasiewicz condition which bounds $\delta_t$ by the gradient norm, allows one to use techniques from convex smooth analysis to show that $\delta_t$ is on the scale of $\frac{1}{t}$. By contrast, for multi-agent learning, there can be multiple NEs with different values, hence $\delta_t$ is ill-defined. Further, note that the NE-gap is different from the optimality gap, hence gradient ascent no longer guarantees monotonic decreasing of NE-gap (Figure 1(a)), and we can only exploit non-convex optimization techniques that yield $O(\frac{1}{\epsilon^2})$ complexities.

For the same reason, the rate of convergence we obtain for natural gradient play is $O\left(\frac{n(\phi_{\max} - \phi_{\min})^2 M}{(1-\gamma)^3 c \epsilon^2}\right)$, which is worse than the dimension free convergence rate of $O\left(\frac{1}{(1-\gamma)^2 \epsilon}\right)$ given in [2] for single-agent MDPs. (A better exponential convergence rate for natural PG has also been proved in [16, 25] with the exponential factor being problem dependent.) Nevertheless, the dependence on $\frac{1}{c}$, $\frac{1}{1-\gamma}$ and $M$ is better than gradient play, suggesting that the preconditioning of natural gradient play at least partially captures the geometry of the parameter space. We also note that the quadratic dependence on $(\phi_{\max} - \phi_{\min})$ might be a proof artifact. It remains an open question whether this can be reduced to a linear dependence.

## 5 Gradient play with $\log$-barrier regularization

The previous section has shown that, for unregularized objectives, the convergence rate for gradient based algorithms depends on a factor $\frac{1}{c}$ that can be arbitrarily large for bad initializations. This motivates us to investigate regularization, in hopes of removing the dependence on $\frac{1}{c}$. For this purpose, we consider $\log$-barrier regularization:

$$\widetilde{J}_i(\theta) = J_i(\theta) + \lambda \sum_{s, a_i} \log \pi_{\theta_i}(a_i|s).$$

Define:

$$\widetilde{\Phi}(\theta) = \Phi(\theta) + \lambda \sum_{i=1}^{n} \sum_{s, a_i} \log \pi_{\theta_i}(a_i|s). \tag{14}$$

It is not hard to verify that the gradient with respect to $J_i$ is:

$$\frac{\partial \widetilde{J}_i(\theta)}{\partial \theta_{s, a_i}} = \frac{\partial \widetilde{\Phi}(\theta)}{\partial \theta_{s, a_i}} = \frac{1}{1-\gamma} d_\theta(s) \pi_{\theta_i}(a_i|s) \overline{A}_i^\theta(s, a_i) + \lambda - \lambda |\mathcal{A}_i| \pi_{\theta_i}(a_i|s).$$

**Discussion on the choice of the regularizer:** Before analyzing the resulting algorithm we first discuss the motivation for this regularizer. First, note that for each agent, the additional regularizer only depends on an agent's own local policy, which is desirable for multiagent RL. As an alternative, one might impose regularization by choosing

$$\widetilde{\Phi}(\theta) = \Phi(\theta) + \lambda \mathbb{E}_{s \sim d_\theta(\cdot)} \sum_{i=1}^n \sum_{a_i} \log \pi_{\theta_i}(a_i|s);$$

i.e., so that the regularization weight imposed on a state $s$ depends on the state visitation probability $d_\theta(s)$. However, in this case the gradient of the $i$-th agent $\nabla_{\theta_i} \widetilde{\Phi}(\theta)$ will not only depend on its own policy parameter $\theta_i$, but also on other parameters of the other agents' policies $\theta_{-i}$. Thus, running gradient based algorithms with such a regularization scheme can no longer be executed in a fully decentralized manner using local policy information. Therefore, we prefer regularization (14) which does not depend on $d_\theta(s)$. Secondly, we adopt the $\log$-barrier instead of entropy regularization due to technical rather than practical considerations. Although entropy regularization achieves fast exponential convergence in single agent learning [7, 24], for multi-agent learning, we haven't been able to obtain results as strong as the $\log$-barrier regularization. Intuitively, the $\log$-barrier regularized gradient field repels the trajectory from regions with small $\pi_i(a_i|s)$ values (where the geometry becomes close to singular) more strongly, which enables us to obtain our current analysis. However, we emphasize that our result does not imply that $\log$-barrier is better than entropy regularization in practice. It remains future work to determine whether entropy regularization, or other methods such as trust region based methods, can achieve the same, or even better convergence rates.

## 5.1 Gradient play

We first consider gradient play algorithm, i.e.,

$$\theta_i^{(t+1)} = \theta_i^{(t)} + \eta \nabla_{\theta_i} \widetilde{J}_i(\theta^{(t)}). \tag{15}$$

Fortunately, similar analysis from [2] for single-agent MDP can be generalized to MPG with slight modifications. Here we only state the result and defer the proof to Appendix G.1.

**Theorem 6.** *Under Assumption 1, for* $\eta = \frac{(1-\gamma)^3}{6n+2\lambda \max_i |\mathcal{A}_i|(1-\gamma)^3}$, *and* $\lambda = \frac{\epsilon}{M \max_i |\mathcal{A}_i|}$, *let* $\theta^{(0)}$ *be the uniform random policy, i.e.,* $\theta^{(0)} = \mathbf{0}$, *then running gradient play* (15) *for* $T$ *steps, where* $T \gtrsim O\left(\frac{n \max_i |\mathcal{A}_i|^2 (\phi_{\max} - \phi_{\min}) M^2}{(1-\gamma)^4 \epsilon^2}\right)$ *will guarantee that* $\min_{0 \le t \le T-1} \mathtt{NE\text{-}gap}(\theta^{(t)}) \le \epsilon$.

Note that compared to the unregularized case in Theorem 5, it only requires Assumption 1, while the convergence rate is accelerated by eliminating the dependence on $\frac{1}{c}$. However, as a (worthy) tradeoff, the dependence on the action space size $\max_i |\mathcal{A}_i|$ now becomes quadratic. The key reason for these differences is that $\log$-barrier regularization assures that any policy with sufficiently small gradient norm cannot be close to the boundary of the probability simplex where the non-uniform Łojasiewicz constant is large.

## 5.2 Natural gradient play

In the unregularized setting, we have seen that natural gradient play enjoys a better convergence rate than gradient play, which motivates us to consider whether a similar advantage still holds for the regularized case. In this section we consider natural gradient play

$$\theta_i^{(t+1)} = \theta_i^{(t)} + \eta F_i(\theta^{(t)})^\dagger \nabla_{\theta_i} \widetilde{J}_i(\theta_i^{(t)}), \tag{16}$$

which is equivalent to (see the proof in Appendix C)

$$\pi_i^{(t+1)}(a_i|s) \propto \pi_i^{(t)}(a_i|s) \exp\left(\frac{\eta}{1-\gamma} \overline{A_i^{(t)}}(s, a_i) + \frac{\eta \lambda}{d^{(t)}(s)\pi_i^{(t)}(a_i|s)} - \frac{\eta \lambda |\mathcal{A}_i|}{d^{(t)}(s)}\right). \tag{17}$$

**Theorem 7.** *(Proof given in Appendix G.2) Under Assumption 1, for*

$$\eta = \min\left\{\frac{1}{15\left(\frac{1}{(1-\gamma)^2} + \lambda|\mathcal{A}_i|M\right)}, \frac{1}{4\left(4\lambda \max_i |\mathcal{A}_i|M^2 + \frac{4M}{(1-\gamma)^2} + \frac{3nM}{(1-\gamma)^3}\right)}\right\}, \text{ the natural gradient play (17)}$$

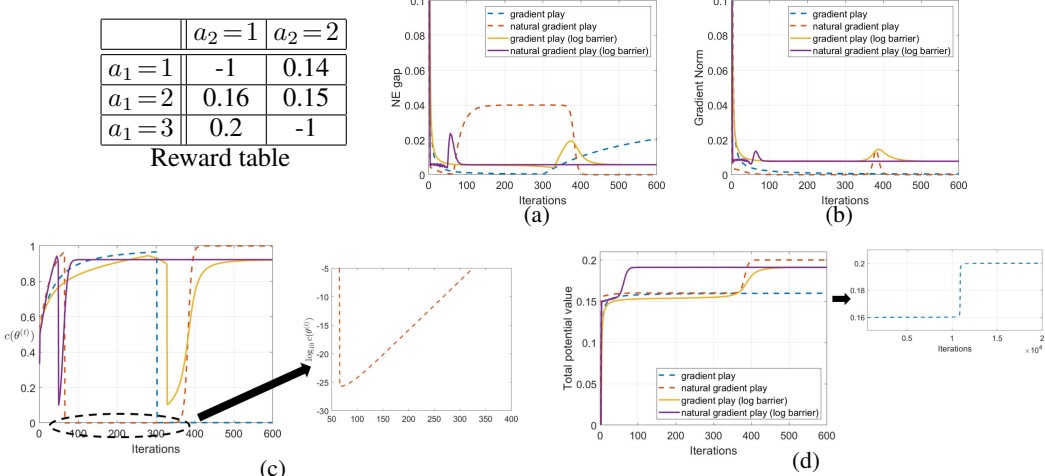

| | | $a_2=1$ | $a_2=2$ |
|---|---|---|---|
| $a_1=1$ | | -1 | 0.14 |
| $a_1=2$ | | 0.16 | 0.15 |
| $a_1=3$ | | 0.2 | -1 |

Reward table

Figure 1: We consider a two-player identical reward matrix game as shown in the reward table. We run gradient play and natural gradient play (with and without $\log$-barrier regularization) with initial policies being the uniform distribution (i.e., $\pi_1 = [\frac{1}{3}, \frac{1}{3}, \frac{1}{3}]$, $\pi_2 = [\frac{1}{2}, \frac{1}{2}]$). The subfigures (a)-(d) show how the $\texttt{NE-gap}(\theta^{(t)})$, $\|\nabla_\theta \Phi(\theta^{(t)})\|_2$, $c(\theta^{(t)})$ (defined in (7)) and $\Phi(\theta^{(t)})$ change with each iteration respectively. In Figure (c), we zoom in on the $\log_{10} c(\theta)$ factor for natural gradient play. In Figure (d), we also zoom out the trajectory for running gradient play to iteration $2 \times 10^4$. Here the step sizes were chosen to be $\eta = 5$ while the regularization weight $\lambda$ was chosen to be $\lambda = 0.003$. In consideration of numerical stability issues, we truncate the update step of natural gradient play with $\log$-barrier regularization by a maximum absolute value of 1 for each entry. For more numerical results and corresponding analysis see Appendix A.

*will guarantee that* $\frac{\sum_{t=0}^{T-1} \texttt{NE-gap}(\theta^{(t)})}{T} \leq \frac{9\left(\widetilde{\Phi}(\theta^{(T)}) - \widetilde{\Phi}(\theta^{(0)})\right)}{\eta \lambda T} + \lambda \max_i |\mathcal{A}_i| M$, *Further, by setting* $\lambda = \frac{\epsilon}{2 \max_i |\mathcal{A}_i| M}$, $\theta^{(0)} = \mathbf{0}$, *for* $T \gtrsim O\left(\frac{n \max_i |\mathcal{A}_i| (\phi_{\max} - \phi_{\min}) M^2}{(1-\gamma)^4 \epsilon^2}\right)$, *we have* $\frac{\sum_{t=0}^{T-1} \texttt{NE-gap}(\theta^{(t)})}{T} \leq \epsilon$.

*Remark* 3. (**Proof sketch and novelty**) As also stated for unregularized natural gradient play, there's no direct analysis tools we could borrow from literature for the analysis of natural gradient play. Our analysis depends on two key lemmas. The first is a sufficient ascent lemma on $\widetilde{\Phi}(\theta^{(t+1)}) - \widetilde{\Phi}(\theta^{(t)})$ for each natural gradient step (Lemma 26). Another key lemma (Lemma 24) states that the algorithm implicitly ensures that the policies never go near the boundary of the probability simplex, i.e., it can be uniformly lower-bounded by $\pi_i^{(t)}(a_i|s) \geq \frac{\lambda}{4\left(\lambda |\mathcal{A}_i| M + \frac{1}{(1-\gamma)^2}\right)}$, $\forall t$. Combining the two lemmas, it can be concluded that the ascent value $\widetilde{\Phi}(\theta^{(t+1)}) - \widetilde{\Phi}(\theta^{(t)})$ can be bounded by $\texttt{NE-gap}(\theta^{(t)})$ plus a $\lambda \max_i |\mathcal{A}_i| M$ bias term (Lemma 27 and 28), thus the proof is finished by standard telescoping technique and choosing an appropriate $\lambda$.

Compared with gradient play, natural gradient play manages to reduce the time complexity by a $\max_i |\mathcal{A}_i|$ factor. Further, gradient play only guarantees the minimal NE-gap smaller than $\epsilon$, while natural gradient play guarantees the average NE-gap along the trajectory smaller than $\epsilon$. To the best of our knowledge, this is the best time complexity bound for the softmax parameterization in a MPG.

# 6 An Illustrative example

This section aims to gain a better understanding of the four gradient play algorithms, (8), (10), (15), and (17). To better justify our theoretical results and provide additional insights, we choose a carefully designed simple two-player game so that our theoretical results can be easily revealed from the empirical observations. However the four algorithms also works for settings with more agents. [1] Due to space limits, we defer the simulation with more agents in Appendix A.

The reward table as well as the performance of the four algorithms are shown in Figure 1. Comparing the $\log$-barrier regularized algorithms to the unregularized counterparts, one can see that the regularized dynamics converge faster but with a bias induced by the regularizer. This finding corroborates

---

[1] Code can be found in https://github.com/DianYu420376/NeurIPS2022-softmax-MPG

the analyses given in Theorem 6 and 7. By contrast, the unregularized dynamics are able to find a policy with zero NE-gap asymptotically, but tend to get stuck in regions where $c(\theta^{(t)})$ is very close to zero, as illustrated in Fig 1(a)(b). Specifically unregularized natural gradient play gets stuck around iteration 100-400 in a region where the gradient norm and $c(\theta^{(t)})$ are both close to zero while the NE-gap is not. This corroborates the finding in Lemma 1. Similar behavior can be observed for gradient play if we keep running the algorithm. In comparing the natural gradient play to gradient play algorithms, natural gradient play generally converges faster, which matches with our complexity analysis. However, natural gradient play with $\log$-barrier regularization can suffer from numerical instability due to the $1/\pi_i^{(t)}(a_i|s)$ term in the exponential factor. In this case, the stepsize needs to be chosen carefully. To bypass the numerical instability, we truncate the update step of natural gradient play with $\log$-barrier regularization by a maximum absolute value of 1 for each entry.

# 7    Discussions and conclusions

We have established finite time iteration complexity bounds for gradient and natural gradient play under the softmax parameterization, considering both unregularized and $\log$-barrier regularized dynamics, in the Markov potential game setting. To our best knowledge, these are the first finite time global convergence results for softmax gradient play for MPGs. However, our work suffers from the following limitations: firstly, the paper mainly focuses on MPG settings, which limits its application to general-sum Markov games; secondly, convergence results for the unregularized case relies on an extra assumption that the stationary points are isolated; thirdly, for the regularized case, we consider $\log$-barrier regularization, which is admittedly a stronger regularization compared with entropy regularization which is more frequently used in practice. Some limitations are due to technical challenges, some might be caused by the fundamental difficulties of multi-agent learning. It remains interesting open questions to sharpen the analysis, derive similar or better bounds for other regularizations, and to develop more fundamental understandings of multi-agent learning.

## Acknowledgment

Runyu Zhang is supported by NSF AI institute: 2112085, ONR YIP: N00014-19-1-2217, NSF CNS: 2003111 and NSF CPS: 2038603.

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
