## Checklist

The checklist follows the references. Please read the checklist guidelines carefully for information on how to answer these questions. For each question, change the default **[TODO]** to [Yes] , [No] , or [N/A] . You are strongly encouraged to include a **justification to your answer**, either by referencing the appropriate section of your paper or providing a brief inline description. For example:

- Did you include the license to the code and datasets? [Yes] See Section **??**.
- Did you include the license to the code and datasets? [No] The code and the data are proprietary.
- Did you include the license to the code and datasets? [N/A]

Please do not modify the questions and only use the provided macros for your answers. Note that the Checklist section does not count towards the page limit. In your paper, please delete this instructions block and only keep the Checklist section heading above along with the questions/answers below.

1. For all authors...
    (a) Do the main claims made in the abstract and introduction accurately reflect the paper's contributions and scope? [Yes]
    (b) Did you describe the limitations of your work? [Yes] See the contribution paragraph in Section 1, assumptions in theorems, and Section 7: Discussions and Conclusions.
    (c) Did you discuss any potential negative societal impacts of your work? [Yes] As this work is primarily theoretical, we do not foresee any potentially negative societal impacts of this work.
    (d) Have you read the ethics review guidelines and ensured that your paper conforms to them? [Yes]

2. If you are including theoretical results...

    (a) Did you state the full set of assumptions of all theoretical results? [Yes]

    (b) Did you include complete proofs of all theoretical results? [Yes]

3. If you ran experiments...

    (a) Did you include the code, data, and instructions needed to reproduce the main experimental results (either in the supplemental material or as a URL)? [Yes] The URL is provided in Section 6

    (b) Did you specify all the training details (e.g., data splits, hyperparameters, how they were chosen)? [Yes]

    (c) Did you report error bars (e.g., with respect to the random seed after running experiments multiple times)? [N/A] Our experiments are deterministic

    (d) Did you include the total amount of compute and the type of resources used (e.g., type of GPUs, internal cluster, or cloud provider)? [Yes]

4. If you are using existing assets (e.g., code, data, models) or curating/releasing new assets...

    (a) If your work uses existing assets, did you cite the creators? [N/A]

    (b) Did you mention the license of the assets? [N/A]

    (c) Did you include any new assets either in the supplemental material or as a URL? [N/A]

    (d) Did you discuss whether and how consent was obtained from people whose data you're using/curating? [N/A]

    (e) Did you discuss whether the data you are using/curating contains personally identifiable information or offensive content? [N/A]

5. If you used crowdsourcing or conducted research with human subjects...

    (a) Did you include the full text of instructions given to participants and screenshots, if applicable? [N/A]

    (b) Did you describe any potential participant risks, with links to Institutional Review Board (IRB) approvals, if applicable? [N/A]

    (c) Did you include the estimated hourly wage paid to participants and the total amount spent on participant compensation? [N/A]

**Other notations:** We use the abbreviation $\pi_{\theta_{i,s}}$ to denote the probability distribution $\pi_{\theta_i}(\cdot|s)$ (as well as the corresponding $|\mathcal{A}_i|$ dimensional vector). We use $\|\cdot\|_1, \|\cdot\|_2, \|\cdot\|_\infty$ to denote the $\ell_1, \ell_2, \ell_\infty$ norm respectively. $\mathrm{KL}(\cdot\|\cdot)$ is used to denote the KL divergence of two probablity distributions. We also define the value function, Q-function, advantage function, averaged Q-function and averaged advantage function with respect to potential function $\phi$ as

$$V_\phi^\theta(s):=\mathbb{E}\left[\sum_{t=0}^\infty \gamma^t \phi(s(t),a(t))\,\middle|\,\pi_\theta, s(0)=s\right],$$

$$Q_\phi^\theta(s,a):=\mathbb{E}\left[\sum_{t=0}^\infty \gamma^t \phi(s(t),a(t))\,\middle|\,\pi_\theta, s(0)=s, a(0)=a\right],$$

$$A_\phi^\theta(s,a):=Q_\phi^\theta(s,a)-V_\phi^\theta(s), \quad \overline{Q_{i,\phi}^\theta}(s,a_i):=\sum_{a_{-i}}\pi_{\theta_{-i}}(a_{-i}|s)Q_\phi^\theta(s,a_i,a_{-i}),$$

$$\overline{A_{i,\phi}^\theta}(s,a_i):=\sum_{a_{-i}}\pi_{\theta_{-i}}(a_{-i}|s)A_\phi^\theta(s,a_i,a_{-i}).$$

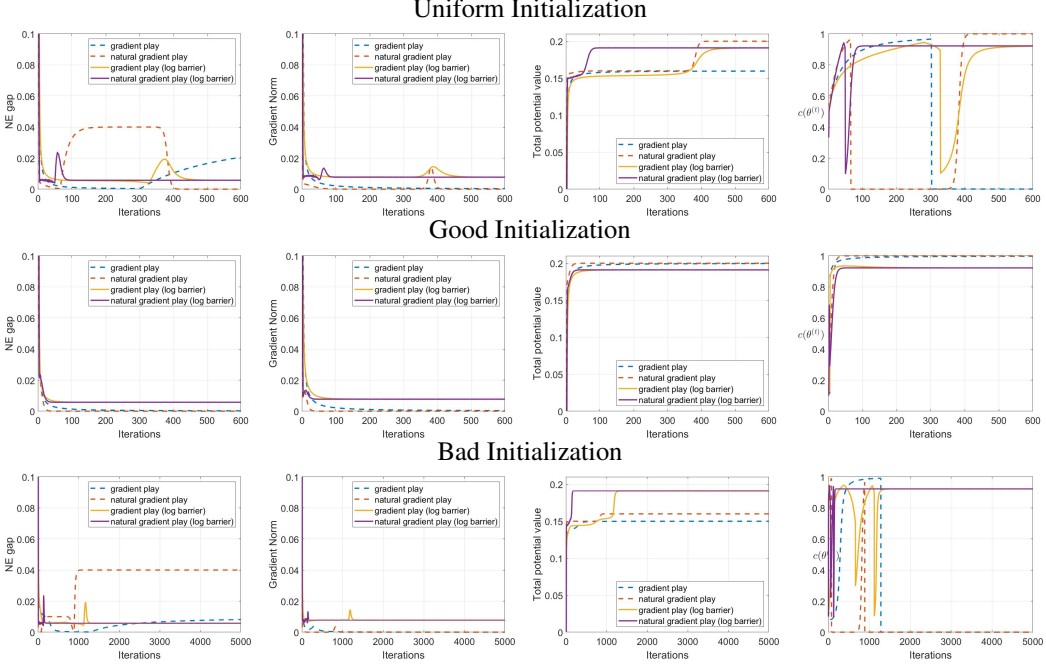

Figure 2: This set of figures shows how each algorithm performs for different initializations. Uniform initialization is the same as described in Figure 1; for good initialization, we choose the initial parameter as $\pi_1 = [0.1, 0.1, 0.8], \pi_2 = [0.5, 0.5]$; for good initialization, we choose the initial parameter as $\pi_1 = [0.8, 0.1, 0.1], \pi_2 = [0.5, 0.5]$. Figures from top row to bottom plot out $\texttt{NE-gap}(\theta^{(t)}), \|\nabla_\theta\Phi(\theta^{(t)})\|_2, \Phi(\theta^{(t)})$ and $c(\theta^{(t)})$ respectively. Here we choose $\eta = 5, \lambda = 0.003$.

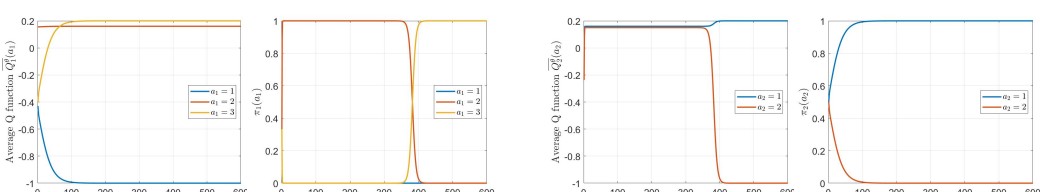

Figure 3: How $\overline{Q_i^{(t)}}(a_i), \pi_i^{(t)}(a_i)$ change with time when running unregularized natual gradient play. The left two figures plots out $\overline{Q_1^{(t)}}(a_1), \pi_1^{(t)}(a_1)$ and the right two figures plots out $\overline{Q_2^{(t)}}(a_2), \pi_2^{(t)}(a_2)$.

# A   Numerical Simulations

This section provides more material for the numerical example shown in Section 6. Figure 2 displays numerical performance for different initialization policies. All four algorithms perform well given a good initialization, i.e., initial policy close to a stable NE. However for bad initialization that is close to a non-NE stationary point, $\log$-barrier regularized algorithms can escape bad regions and converge to NE much faster than unregularized dynamics.

To examine why multi-agent learning suffers more from getting stuck at undesirable stationary points, we plot out the trajectory for $\overline{Q}_i^{(t)}(a_i), \pi_i^{(t)}(a_i)$ for both agents in Figure 3. We will mainly focus our attention on the two plots on the left. Note that for the first few steps, $\overline{Q}_1^{(t)}(a_1 = 2)$ is much larger than $\overline{Q}_1^{(t)}(a_1 = 3)$, thus the natural gradient play scheme (10) will drive $\pi_1^{(t)}(a_1 = 2)$ close to 1 and $\pi_1^{(t)}(a_1 = 3)$ close to 0 very quickly. However, at around iteration 70, $\overline{Q}_1^{(t)}(a_1 = 3)$ becomes slightly larger than $\overline{Q}_1^{(t)}(a_1 = 2)$. Unfortunately, at this stage, most of the probability is assigned to the suboptimal action $a_1 = 2$ and the optimal action receives $\pi_1^{(t)}(a_1 = 3)$ close to zero. Thus it will take more steps to bring $\pi_1^{(t)}(a_1 = 2)$ from 1 to 0 and $\pi_1^{(t)}(a_1 = 3)$ from 0 to 1, which reflects as the trajectory being stuck at the non-NE stationary policy with $\pi_1(a_1 = 3) = 1$ in numerical behavior. From this simulation, we may conclude that one important reason for natural gradient play to get stuck at undesirable stationary points is due to the fact that the value of averaged $Q$-functions $\overline{Q}_i^{(t)}$'s for different actions might switch order during the learning process. In contrast, for single agent bandit learning, the averaged $Q$-function as well as the $Q$-function itself is the same as the reward value of a certain action $r(a)$, and thus will not change order, which explains why it can achieve dimension free convergence in single agent learning.

Additionally, we would like to remark that the algorithms considered in this paper also generalizes to settings with more agents, and similar phenomenon will still be observed. See Figure 4 and 5 for numerical simulations on a 3-agent example and an 8-agent example.

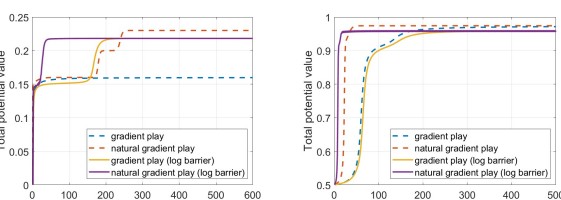

Figure 4: A 3-agent Example   Figure 5: An 8-agent Example

Running trajectories for all the four algorithms for one set of initializations takes approximately 2.04 seconds of CPU running time (Intel(R) Core(TM) i5-8250U CPU @ 1.60GHz 1.80 GHz).

# B   Derivation of Gradient and Performance Difference Lemma

*Proof.* (of Equation 4) According to policy gradient theorem [34]:

$$\frac{\partial J_i(\theta)}{\partial \theta_{s,a_i}} = \frac{1}{1 - \gamma} \sum_{s'} \sum_{a'} d_\theta(s') \pi_\theta(a'|s') \frac{\partial \log \pi_\theta(a'|s')}{\partial \theta_{s,a_i}} Q_i^\theta(s, a)$$

Since for softmax parameterization:

$$\frac{\partial \log \pi_\theta(a'|s')}{\partial \theta_{s,a_i}} = \frac{\partial \log \pi_{\theta_i}(a_i'|s')}{\partial \theta_{s,a_i}} = \mathbf{1}\{a_i' = a_i, s' = s\} - \mathbf{1}\{s' = s\} \pi_{\theta_i}(a_i|s)$$

Thus we have that:

$$\frac{\partial J_i(\theta)}{\partial \theta_{s,a_i}} = \frac{1}{1 - \gamma} \sum_{s'} \sum_{a'} d_\theta(s') \pi_\theta(a'|s') \left( \mathbf{1}\{a_i' = a_i, s' = s\} - \mathbf{1}\{s' = s\} \pi_{\theta_i}(a_i|s) \right) Q_i^\theta(s, a')$$

$$= \frac{1}{1 - \gamma} d_\theta(s) \pi_{\theta_i}(a_i|s) \sum_{a'_{-i}} \pi_{\theta_{-i}}(a'_{-i}|s) Q_i^\theta(s, a_i, a'_{-i}) - \frac{1}{1 - \gamma} d_\theta(s) \pi_{\theta_i}(a_i|s) \sum_{a'} \pi_\theta(a'|s') Q_i^\theta(s, a')$$

$$= \frac{1}{1 - \gamma} d_\theta(s) \pi_{\theta_i}(a_i|s) \overline{Q}_i^\theta(s, a_i, a'_{-i}) - \frac{1}{1 - \gamma} d_\theta(s) \pi_{\theta_i}(a_i|s) V_i^\theta(s)$$

$$= \frac{1}{1-\gamma} d_\theta(s) \pi_{\theta_i}(a_i|s) \overline{A_i^\theta}(s, a_i)$$

$\square$

We also introduce a useful lemma used throughout the proof which is derived from the performance difference lemma in MDP [15].

**Lemma 8.** *Let $\theta' = (\theta_i', \theta_{-i})$,*

$$J_i(\theta_i', \theta_{-i}) - J_i(\theta_i, \theta_{-i}) = \frac{1}{1-\gamma} \sum_{s, a_i} d_{\theta'}(s) \pi_{\theta_i'}(a_i|s) \overline{A_i^\theta}(s, a_i)$$

*Proof.* From performance difference lemma [15]

$$J_i(\theta_i', \theta_{-i}) - J_i(\theta_i, \theta_{-i}) = \frac{1}{1-\gamma} \sum_{s, a} d_{\theta'}(s) \pi_{\theta'}(a|s) A_i^\theta(s, a)$$

$$= \frac{1}{1-\gamma} \sum_{s, a_i} d_{\theta'}(s) \pi_{\theta_i'}(a_i|s) \sum_{a_{-i}} \pi_{\theta_{-i}}(a_{-i}|s) A_i^\theta(s, a_i, a_{-i})$$

$$= \frac{1}{1-\gamma} \sum_{s, a_i} d_{\theta'}(s) \pi_{\theta_i'}(a_i|s) \overline{A_i^\theta}(s, a_i).$$

$\square$

## C  Derivation of Natural Gradient Play

**Lemma 9.**

$$\mathbb{E}_{a \sim \pi_\theta(\cdot|s)} \left[ \nabla_{\theta_{i,s}} \log \pi_{\theta_i}(a_i|s) \nabla_{\theta_{i,s}} \log \pi_{\theta_i}(a_i|s)^\top \right] = \text{diag}\{\pi_{\theta_i,s}\} - \pi_{\theta_i,s} \pi_{\theta_i,s}^\top := F_{i,s}(\theta_{i,s}),$$

*where* $\text{diag}\{\cdot\}$ *denotes the diagonal matrix generated by the corresponding vector, and $\pi_{\theta_i,s} \in \mathbb{R}^{|\mathcal{A}_i|}$ is the vector that denotes $\pi_{\theta_i}(\cdot|s)$. Further, $F_{i,s}(\theta_{i,s})$ is a semi-positive definite matrix, where the eigenvalue $0$ has the eigenspace of dimension 1 that is the span of the all one-vector $\mathbf{1}$.*

*Proof.* Calculating the gradient using chain rule we have

$$\frac{\partial \log \pi_{\theta_i}(a_i|s)}{\partial \theta_{a_i', s}} = \mathbf{1}\{a_i' = a_i\} - \pi_{\theta_i}(a_i'|s).$$

Let $\mathbf{1}_{a_i} \in \mathbb{R}^{|\mathcal{A}_i|}$ denote the vector where the entry corresponds to $a_i$ is 1 and other entries are zero. Then

$$\nabla_{\theta_{i,s}} \log \pi_{\theta_i}(a|s) = \mathbf{1}_{a_i} - \pi_{\theta_i,s}$$

$$\implies \nabla_{\theta_{i,s}} \log \pi_{\theta_i}(a_i|s) \nabla_{\theta_{i,s}} \log \pi_{\theta_i}(a_i|s)^\top = \text{diag}\{\mathbf{1}_{a_i}\} - \pi_{\theta_i,s} \mathbf{1}_{a_i}^\top - \mathbf{1}_{a_i} \pi_{\theta_i,s}^\top + \pi_{\theta_i,s} \pi_{\theta_i,s}^\top$$

Taking the expectation $\mathbb{E}_{a \sim \pi_{\theta_i}(\cdot|s)}$ we have

$$\mathbb{E}_{a \sim \pi_{\theta_i}(\cdot|s)} \left[ \nabla_{\theta_{i,s}} \log \pi_{\theta_i}(a|s) \nabla_{\theta_{i,s}} \log \pi_{\theta_i}(a|s)^\top \right] = \text{diag}\{\pi_{\theta_i,s}\} - \pi_{\theta_i,s} \pi_{\theta_i,s}^\top - \pi_{\theta_i,s} \pi_{\theta_i,s}^\top + \pi_{\theta_i,s} \pi_{\theta_i,s}^\top$$

$$= \text{diag}\{\pi_{\theta_i,s}\} - \pi_{\theta_i,s} \pi_{\theta_i,s}^\top$$

Further, for softmax parameterization, $\pi_{\theta_i}(a_i|s) > 0$, $\forall a_i$. Thus $F_{i,s}(\theta_{i,s})$ is a (non-strict) diagonally dominant matrix with diagonal entries all being positive and off-diagonal entries all being negative, in which case the all-one vector $\mathbf{1}$ is the only eigenvector for eigenvalue $0$. $\square$

**Corollary 10.**

$$F_i(\theta) = \text{blkdiag}\{d_\theta(s) F_{i,s}(\theta_{i,s})\}_{s \in \mathcal{S}},$$

*where* $\text{blkdiag}\{\cdot\}$ *denotes the block-diagonal matrix generated by corresponding sub-matrices.*

*Proof.* This is a direct corollary of Lemma 9 , since

$$\frac{\partial \log \pi_{\theta_i}(a_i|s)}{\partial \theta_{a'_i,s'}} = 0, \quad \text{for } s' \neq s,$$

we have that

$$F_i(\theta) = \mathbb{E}_{s \sim d_\theta(\cdot)}\mathbb{E}_{a_i \sim \pi_{\theta_i}(\cdot|s)} \left[ \nabla_{\theta_i} \log \pi_{\theta_i}(a_i|s) \nabla_{\theta_i} \log \pi_\theta(a_i|s)^\top \right] = \text{blkdiag}\{d_\theta(s) F_{i,s}(\theta_{i,s})\}$$

$\square$

**Lemma 11.** *For vector $g : \mathcal{S} \times \mathcal{A}_i \to \mathbb{R}$, with $\sum_{a_i} g(s, a_i) = 0$, $\forall s \in \mathcal{S}$, we have that*

$$\left[F_i(\theta)^\dagger g\right]_{(s,a_i)} = \frac{1}{d_\theta(s)\pi_{\theta_i}(a_i|s)} g(s, a_i) + c(s),$$

*where $c(s)$ is a function that depend on state $s$ but not on $a_i$.*

*Proof.* Since $F_i(\theta)$ is a block diagonal matrix,

$$\left[F_i(\theta)^\dagger g\right]_{(s,\cdot)} = \frac{1}{d_\theta(s)} F_{i,s}(\theta_{i,s})^\dagger g(s,\cdot).$$

From Lemma 9, since $F_{i,s}$ only has a one-dimensional eigenspace for eigenvalue 0, and the eigenspace is the span of the all-one vector $\mathbf{1}$, we have that

$$F_{i,s}(\theta_{i,s})^\dagger F_{i,s}(\theta_{i,s}) = I - \frac{1}{|\mathcal{A}_i|}\mathbf{1}\mathbf{1}^\top.$$

Let $f(s, a_i) := \frac{1}{d_\theta(s)\pi_{\theta_i}(a_i|s)} g(s, a_i)$

$$d_\theta(s) \left[F_{i,s}(\theta_{i,s}) f(s,\cdot)\right]_{a_i} = d_\theta(s) \left( \pi_{\theta_i}(a_i|s) f(s, a_i) - \pi_{\theta_i}(a_i|s) \sum_{a'_i} \pi_{\theta_i}(a'_i|s) f(s, a'_i) \right)$$

$$= g(s, a_i) - \pi_{\theta_i}(a_i|s) \sum_{a'_i} g(s, a'_i) = g(s, a_i),$$

i.e.,

$$d_\theta(s) F_{i,s}(\theta_{i,s}) f(s,\cdot) = g(s,\cdot)$$

$$\implies \frac{1}{d_\theta(s)} F_{i,s}(\theta_{i,s})^\dagger g(s,\cdot) = F_{i,s}(\theta_{i,s})^\dagger F_{i,s}(\theta_{i,s}) f(s,\cdot)$$

$$= \left( I - \frac{1}{|\mathcal{A}_i|}\mathbf{1}\mathbf{1}^\top \right) f(s,\cdot)$$

$$= f(s,\cdot) - c(s)\mathbf{1},$$

i.e.,

$$\left[F_i(\theta)^\dagger g\right]_{(s,a_i)} = f(s, a_i) - c(s),$$

which completes the proof. $\square$

**Lemma 12.** *Scheme (9) and (10) are equivalent. Similarly, (16) and (17) are equivalent.*

*Proof.* It is not hard to check that $\nabla_{\theta_i} J_i(\theta), \nabla_{\theta_i} \widetilde{J}_i(\theta)$ satisfies

$$\sum_{a_i} [\nabla_{\theta_i} J_i(\theta)]_{(s,a_i)} = 0, \quad \sum_{a_i} \left[\nabla_{\theta_i} \widetilde{J}_i(\theta)\right]_{(s,a_i)} = 0,$$

thus we can apply Lemma 11 and conclude

$$F_i(\theta^{(t)})^\dagger \nabla_{\theta_i} J_i(\theta_i^{(t)}) = \frac{\overline{A_i^{(t)}}(s, a_i)}{1 - \gamma} + c(s)$$

$$F_i(\theta^{(t)})^\dagger \nabla_{\theta_i} \widetilde{J}_i(\theta_i^{(t)}) = \frac{\overline{A_i^{(t)}}(s, a_i)}{1 - \gamma} + \frac{\lambda}{d^{(t)}(s)\pi_i^{(t)}(a_i|s)} - \frac{\lambda|\mathcal{A}_i|}{d^{(t)}(s)} + c(s),$$

which completes the proof. $\qquad\square$

## D    Proof of Lemma 1 and Lemma 2

**Lemma 13.**

$$\texttt{NE-gap}_i(\theta) \leq \frac{1}{1 - \gamma} \max_{s, a_i} \overline{A_i^\theta}(s, a_i), \quad \texttt{NE-gap}(\theta) \leq \frac{1}{1 - \gamma} \max_i \max_{s, a_i} \overline{A_i^\theta}(s, a_i).$$

*Proof.* From performance difference lemma

$$J_i(\theta_i', \theta_{-i}) - J_i(\theta_i, \theta_{-i}) = \frac{1}{1 - \gamma} \sum_{s, a_i} d_{\theta'}(s) \pi_{\theta_i'}(a_i|s) \overline{A_i^\theta}(s, a_i) \quad \text{(Lemma 8)}$$

$$\leq \frac{1}{1 - \gamma} \sum_s d_{\theta'}(s) \max_{a_i} \overline{A_i^\theta}(s, a_i)$$

$$\leq \frac{1}{1 - \gamma} \sum_s d_{\theta'}(s) \max_{a_i} \overline{A_i^\theta}(s, a_i)$$

$$\leq \frac{1}{1 - \gamma} \max_{s, a_i} \overline{A_i^\theta}(s, a_i).$$

Thus we have that

$$\texttt{NE-gap}_i(\theta) \leq \frac{1}{1 - \gamma} \max_{s, a_i} \overline{A_i^\theta}(s, a_i), \quad \texttt{NE-gap}(\theta) \leq \frac{1}{1 - \gamma} \max_i \max_{s, a_i} \overline{A_i^\theta}(s, a_i).$$

$\square$

*Proof of Lemma 1.* From Lemma 13 we have that

$$\texttt{NE-gap}_i(\theta) \leq \frac{1}{1 - \gamma} \max_{s, a_i} \overline{A_i^\theta}(s, a_i).$$

Since

$$\max_{a_i} \overline{A_i^\theta}(s, a_i) \leq \frac{1}{\sum_{a_i^* \in \mathrm{argmax}_{a_i} \overline{Q_i^\theta}(s, a_i)} \pi_{\theta_i}(a_i^*|s)} \sum_{a_i} |\pi_{\theta_i}(a_i|s) \overline{A_i^\theta}(s, a_i)|$$

$$\leq \frac{\sqrt{|\mathcal{A}_i|}}{\sum_{a_i^* \in \mathrm{argmax}_{a_i} \overline{Q_i^\theta}(s, a_i)} \pi_{\theta_i}(a_i^*|s)} \sqrt{\sum_{a_i} \left( \pi_{\theta_i}(a_i|s) \overline{A_i^\theta}(s, a_i) \right)^2}$$

$$= \frac{\sqrt{|\mathcal{A}_i|}}{\sum_{a_i^* \in \mathrm{argmax}_{a_i} \overline{Q_i^\theta}(s, a_i)} \pi_{\theta_i}(a_i^*|s)} \frac{1 - \gamma}{d_\theta(s)} \sqrt{\sum_{a_i} \left( \frac{1}{1 - \gamma} d_\theta(s) \pi_{\theta_i}(a_i|s) \overline{A_i^\theta}(s, a_i) \right)^2}$$

$$\leq \frac{(1 - \gamma) M(\theta) \sqrt{|\mathcal{A}_i|}}{c(\theta)} \|\nabla_{\theta_i} J_i(\theta)\|_2.$$

Thus

$$\texttt{NE-gap}_i(\theta) \leq \frac{1}{1 - \gamma} \max_{s, a_i} \overline{A_i^\theta}(s, a_i)$$

$$\leq \frac{\sqrt{|\mathcal{A}_i|}M(\theta)}{c(\theta)}\|\nabla_{\theta_i}J_i(\theta)\|_2$$

$\square$

*Remark* 4. A similar bound to Lemma 1 can be obtained by leveraging equation (259) in [24]

$$\texttt{NE-gap}_i(\theta) \leq \frac{\sqrt{|\mathcal{S}||\mathcal{A}_i|}D_\infty}{c(\theta)}\|\nabla_{\theta_i}J_i(\theta)\|_2, \quad \text{where } D_\infty = \sup_{\theta,\theta'}\max_s \frac{d_{\theta'}(s)}{d_\theta(s)}. \tag{18}$$

Notice that there's an additional $\sqrt{S}$ dependency on the right hand side compared with Lemma 1, while replacing the term $M(\theta)$ by $D_\infty$. We remark that there's no fundamental difference between these two bounds. There's no significant difference in the proof techniques and it is hard to tell which one is better. One can also easily re-derive the set of analysis in the paper using (18), with bounds that depends on $D_\infty$ instead of $M$ and slightly differs in the dependency on $\mathcal{S}$ from our current result.

*Proof of Lemma 2.* Firstly, it is straightforward to see that if $c(\theta^*) \neq 0$, $\theta^*$ is a NE by applying Lemma 1. So we only need to focus on proving that if $\theta^*$ is a NE, then $c(\theta^*) = 1$.

From performance difference lemma, let $\theta' := (\theta'_i, \theta^*_{-i})$

$$J_i(\theta'_i, \theta^*_{-i}) - J_i(\theta^*_i, \theta^*_{-i}) = \frac{1}{1-\gamma}\sum_{s,a_i} d_{\theta'}(s)\pi_{\theta'_i}(a_i|s)\overline{A_i^{\theta^*}}(s,a_i)$$

Select $a_i^*(s) \in \operatorname{argmax}_{a_i}\overline{A_i^{\theta^*}}(s,a_i)$ and set:

$$\pi_{\theta'_i}(a_i|s) = \mathbf{1}\{a_i = a_i^*(s)\},$$

then

$$J_i(\theta'_i, \theta^*_{-i}) - J_i(\theta^*_i, \theta^*_{-i}) = \frac{1}{1-\gamma}\sum_{s,a_i} d_{\theta'}(s)\pi_{\theta'_i}(a_i|s)\overline{A_i^{\theta^*}}(s,a_i)$$

$$= \frac{1}{1-\gamma}\sum_s d_\theta(s)\max_{a_i}\overline{A_i^{\theta^*}}(s,a_i) \geq 0.$$

Since $\theta^*$ is a NE,

$$\implies \max \overline{A_i^{\theta^*}}(s,a_i) = 0, \ \forall s, \forall i.$$

Let $\Delta := \min_s \min_{a_i \notin \operatorname{argmax}_{a_i}\overline{A_i^{\theta^*}}(s,a_i)}|\overline{A_i^{\theta^*}}(s,a_i)|$. Since $\sum_{a_i}\pi_{\theta^*_i}(a_i|s)\overline{A_i^{\theta^*}}(s,a_i) = 0$

$$\implies 0 = \sum_{a_i \in \operatorname{argmax}_{a_i}\overline{A_i^{\theta^*}}(s,a_i)}\pi_{\theta^*_i}(a_i|s)\max_{a_i}\overline{A_i^{\theta^*}}(s,a_i) + \sum_{a_i \notin \operatorname{argmax}_{a_i}\overline{A_i^{\theta^*}}(s,a_i)}\pi_{\theta^*_i}(a_i|s)\overline{A_i^{\theta^*}}(s,a_i)$$

$$\leq -\Delta \sum_{a_i \notin \operatorname{argmax}_{a_i}\overline{A_i^{\theta^*}}(s,a_i)}\pi_{\theta^*_i}(a_i|s)$$

$$\implies \sum_{a_i \notin \operatorname{argmax}_{a_i}\overline{A_i^{\theta^*}}(s,a_i)}\pi_{\theta^*_i}(a_i|s) = 0$$

$$\implies \sum_{a_i \in \operatorname{argmax}_{a_i}\overline{A_i^{\theta^*}}(s,a_i)}\pi_{\theta^*_i}(a_i|s) = 1$$

$$\implies \sum_{a_i \in \operatorname{argmax}_{a_i}\overline{Q_i^{\theta^*}}(s,a_i)}\pi_{\theta^*_i}(a_i|s) = 1$$

$$\implies c(\theta^*) = 1$$

$\square$

# E   Proof of Theorem 4

### E.0.1   Asymptotic convergence for gradient play

**Lemma 14.** *For $\eta \leq \frac{(1-\gamma)^3}{6n}$, running scheme (8) will guarantee that $\lim_{t\to+\infty} \nabla\Phi(\theta^{(t)}) = 0$.*

*Proof.* Since $\Phi(\theta)$ is $\beta$-smooth w.r.t. $\theta$, where $\beta = \frac{6n}{(1-\gamma)^3}$

$$\Phi(\theta^{(t+1)}) - \Phi(\theta^{(t)}) \geq \left\langle \nabla\Phi(\theta^{(t)}), \theta^{(t+1)} - \theta^{(t)} \right\rangle - \frac{\beta}{2}\|\theta^{(t+1)} - \theta^{(t)}\|_2^2$$

$$\geq \frac{\eta}{2}\|\nabla\Phi(\theta^{(t)})\|_2^2 \geq 0$$

which proves the monotonicity of $\Phi(\theta^{(t)})$. Since $\phi$ is a bounded function, this gives:

$$\lim_{t\to+\infty} \|\nabla\Phi(\theta^{(t)})\|_2 = 0. \qquad \square$$

From Lemma 14 and that the stationary points are isolated, we know that the limit for $\theta^{(t)}$ exists, i.e., it is valid to define

$$\theta^{(\infty)} := \lim_{t\to+\infty} \theta^{(t)}.$$

We abbreviate the related functions with respect to $\theta^{(\infty)}$ as follows:

$$Q_i^{(\infty)}(s,a) := Q_i^{\theta^{(\infty)}}(s,a), \qquad V_i^{(\infty)}(s) := V_i^{\theta^{(\infty)}}(s), \qquad A_i^{(\infty)}(s,a) := Q_i^{(\infty)}(s,a) - V_i^{(\infty)}(s)$$

$$\overline{Q_i^{(\infty)}}(s,a_i) := \sum_{a_{-i}} \pi_{-i}^{(\infty)}(a_{-i}|s)Q^{(\infty)}(s,a_i,a_{-i}), \qquad \overline{A_i^{(\infty)}}(s,a_i) := \sum_{a_{-i}} \pi_{-i}^{(\infty)}(a_{-i}|s)A^{(\infty)}(s,a_i,a_{-i})$$

Since $\theta^{(\infty)}$ is the limit of $\theta^{(t)}$, we have that:

$$\lim_{t\to+\infty} \overline{Q_i^{(t)}}(s,a_i) = \overline{Q_i^{(\infty)}}(s,a_i), \qquad \lim_{t\to+\infty} \overline{A_i^{(t)}}(s,a_i) = \overline{A_i^{(\infty)}}(s,a_i) \tag{19}$$

Define:

$$I_0^{i,s} := \{a_i | \overline{Q_i^{(\infty)}}(s,a_i) = V^{(\infty)}(s)\} = \{a_i | \overline{A_i^{(\infty)}}(s,a_i) = 0\}$$

$$I_+^{i,s} := \{a_i | \overline{Q_i^{(\infty)}}(s,a_i) > V^{(\infty)}(s)\} = \{a_i | \overline{A_i^{(\infty)}}(s,a_i) > 0\}$$

$$I_-^{i,s} := \{a_i | \overline{Q_i^{(\infty)}}(s,a_i) < V^{(\infty)}(s)\} == \{a_i | \overline{A_i^{(\infty)}}(s,a_i) < 0\}$$

Let

$$\Delta := \min_i \min_{\{s,a_i | A_i^{(\infty)}(s,a_i) \neq 0\}} |A_i^{(\infty)}(s,a_i)| \tag{20}$$

From Lemma 13, it is sufficient to show that $I_+^{i,s} = \emptyset$, $\forall i, s$.

From the Lemma 14 and the above definitions we have the following corollaries:

**Corollary 15.** *There exists $T_1$, such that $\forall t > T_1$, $\forall s \in \mathcal{S}$, $\forall i \in \{1, 2, \ldots, n\}$,*

$$A_i^{(t)}(s,a_i) < -\frac{\Delta}{4}, \quad \forall a_i \in I_-^{i,s}$$

$$A_i^{(t)}(s,a_i) > \frac{\Delta}{4}, \quad \forall a_i \in I_+^{i,s}$$

$$|A_i^{(t)}(s,a_i)| < \frac{\Delta}{4}, \quad \forall a_i \in I_0^{i,s}$$

*Proof.* This is a direct corollary from (19) and (20). $\qquad \square$

**Corollary 16.**

$$\lim_{t \to +\infty} \sum_{a_i \in I_0^{i,s}} \pi_i^{(t)}(a_i|s) = 1$$

$$\lim_{t \to +\infty} \sum_{a_i \in I_+^{i,s} \cup I_-^{i,s}} \pi_i^{(t)}(a_i|s) = 0$$

*Proof.* This is a direct corollary from Lemma 14,

$$\lim_{t \to +\infty} \nabla \Phi(\theta^{(t)}) = 0$$

$$\implies \lim_{t \to +\infty} \frac{\partial \Phi(\theta^{(t)})}{\partial \theta_{s,a_i}} = \lim_{t \to +\infty} \frac{1}{1-\gamma} d^{(t)}(s) \pi_i^{(t)}(a_i|s) \overline{A_i^{(t)}}(s, a_i) = 0$$

$$\implies \lim_{t \to +\infty} \pi_i^{(t)}(a_i|s) \lim_{t \to +\infty} \overline{A_i^{(t)}}(s, a_i) = 0$$

$$\implies \lim_{t \to +\infty} \pi_i^{(t)}(a_i|s) = 0, \ \forall a_i \notin I_0^{i,s}$$

$$\implies \lim_{t \to +\infty} \sum_{a_i \in I_+^{i,s} \cup I_-^{i,s}} \pi_i^{(t)}(a_i|s) = 0$$

$$\implies \lim_{t \to +\infty} \sum_{a_i \in I_0^{i,s}} \pi_i^{(t)}(a_i|s) = 1 - \lim_{t \to +\infty} \sum_{a_i \in I_+^{i,s} \cup I_-^{i,s}} \pi_i^{(t)}(a_i|s) = 1 \qquad \square$$

**Lemma 17.** $\forall a_i \in I_+^{i,s}, \theta_{s,a_i}^{(t)}$ *is bounded from below.* $\forall a_i \in I_-^{i,s}, \lim_{t \to +\infty} \theta_{s,a_i}^{(t)} = -\infty.$

*Proof.* The first statement, $\forall a_i \in I_+^{i,s}, \theta_{s,a_i}^{(t)}$ is bounded from below, is trivial from Corollary 15. We only need to prove the second statement. The key observation is that:

$$\sum_{a_i} \frac{\partial \Phi(\theta^{(t)})}{\partial \theta_{s,a_i}} = \frac{1}{1-\gamma} d^{(t)}(s) \sum_{a_i} \pi_i^{(t)}(a_i|s) \overline{A_i^{(t)}}(s, a_i) = 0$$

Thus

$$\sum_{a_i} \theta_{s,a_i}^{(t)} = \sum_{a_i} \theta_{s,a_i}^{(0)}.$$

From Corollary 16, we have that

$$\lim_{t \to +\infty} \sum_{a_i \in I_+^{i,s} \cup I_-^{i,s}} \pi_i^{(t)}(a_i|s) = 0$$

$$\implies \exists \, a_i \in I_0^{i,s}, \ s.t. \ \limsup_{t \to +\infty} \theta_{s,a_i}^{(t)} = +\infty$$

And since all $\theta_{s,a_i}^{(t)}$ sum up to a constant and that $\forall a_i \in I_+^{i,s}, \theta_{s,a_i}^{(t)}$ is bounded from below, we have that:

$$\exists \, \overline{a_i} \in I_0^{i,s} \cup I_-^{i,s}, \ s.t. \ \liminf_{t \to +\infty} \theta_{s,\overline{a_i}}^{(t)} = -\infty. \tag{21}$$

From Corollary 15, for $a_i \in I_-^{i,s}, \theta_{s,a_i}^{(t)}$ is monotonically decreasing for $t > T_1$, thus

$$\lim_{t \to +\infty} \theta_{s,a_i}^{(t)} := \theta_{s,a_i}^{(\infty)},$$

where $\theta_{s,a_i}^{(\infty)}$ is either a constant or $-\infty$. We'll prove by contradiction. Suppose $\theta_{s,a_i}^{(\infty)}$ is a constant, then for any $\delta > 0$ there exists $T_1' \geq T_1$ such that $\forall \, t \geq T_1', |\theta_{s,a_i}^{(t)} - \theta_{s,a_i}^{(\infty)}| \leq \delta.$

Let $\overline{a_i} \in \mathcal{A}_i$ be defined as in (21), define:

$$\tau(t) := \begin{cases} t+1, & \text{if } \theta_{s,\overline{a_i}}^{(t)} > \theta_{s,a_i}^{(\infty)} - \delta \\ \min_{t'}\{T_1' \leq t' \leq t | \theta_{s,\overline{a_i}}^{(\tau)} \leq \theta_{s,a_i}^{(\infty)} - \delta, \ \forall t' \leq \tau \leq t\}, & \text{otherwise} \end{cases}$$

We will focus on the set where $\{t | \tau(t) \leq t\}$. Since $\liminf_{t \to +\infty} \theta_{s,\overline{a_i}}^{(t)} = -\infty$, there are infinitely many elements in this set.

For all $\tau(t) \leq \tau \leq t$, we have that:

$$\left| \frac{\frac{\partial \Phi(\theta^{(\tau)})}{\partial \theta_{s,a_i}}}{\frac{\partial \Phi(\theta^{(\tau)})}{\partial \theta_{s,\overline{a_i}}}} \right| = \left| \frac{\pi_i^{(\tau)}(a_i|s)\overline{A_i^{(\tau)}}(s,a_i)}{\pi_i^{(\tau)}(\overline{a_i}|s)\overline{A_i^{(\tau)}}(s,\overline{a_i})} \right| = \exp\left(\theta_{s,a_i}^{(\tau)} - \theta_{s,\overline{a_i}}^{(\tau)}\right) \left| \frac{\overline{A_i^{(\tau)}}(s,a_i)}{\overline{A_i^{(\tau)}}(s,\overline{a_i})} \right|$$

$$\geq \left| \frac{\overline{A_i^{(\tau)}}(s,a_i)}{\overline{A_i^{(\tau)}}(s,\overline{a_i})} \right| \geq \frac{\Delta(1-\gamma)}{4}$$

Thus

$$\frac{\partial \Phi(\theta^{(\tau)})}{\partial \theta_{s,a_i}} \leq \frac{\Delta(1-\gamma)}{4} \frac{\partial \Phi(\theta^{(\tau)})}{\partial \theta_{s,\overline{a_i}}}, \quad \tau(t) \leq \tau \leq t$$

$$\implies \frac{1}{\eta}\left(\theta_{s,a_i}^{(t+1)} - \theta_{s,a_i}^{(\tau(t))}\right) = \sum_{\tau(t)}^{t} \frac{\partial \Phi(\theta^{(\tau)})}{\partial \theta_{s,a_i}} \leq \frac{\Delta(1-\gamma)}{4} \sum_{\tau(t)}^{t} \frac{\partial \Phi(\theta^{(\tau)})}{\partial \theta_{s,\overline{a_i}}} = \frac{\Delta(1-\gamma)}{4\eta}\left(\theta_{s,\overline{a_i}}^{(t+1)} - \theta_{s,\overline{a_i}}^{(\tau(t))}\right)$$

(22)

Since:

$$\theta_{s,\overline{a_i}}^{(\tau(t))} \geq \theta_{s,\overline{a_i}}^{(\tau(t)-1)} - \eta\frac{1}{(1-\gamma)^2} \geq \theta_{s,a_i}^{(\infty)} - \delta - \eta\frac{1}{(1-\gamma)^2}$$

is bounded from below, and that $\theta_{s,a_i}^{\tau(t)}$ is also bounded from above by $\theta_{s,a_i}^{(T_1)}$, thus taking $\liminf_{t \to +\infty}$ on both sides of eq (22) will give

$$\liminf_{t \to +\infty} \theta_{s,a_i}^{(t+1)} - \theta_{s,a_i}^{(\tau(t))} \leq \frac{\Delta(1-\gamma)}{4}\left(\liminf_{t \to +\infty} \theta_{s,\overline{a_i}}^{(t+1)} - \theta_{s,a_i}^{(\infty)} + \delta + \eta\frac{1}{(1-\gamma)^2}\right) = -\infty$$

$$\implies \liminf_{t \to +\infty} \theta_{s,a_i}^{(t)} = -\infty$$

which contradicts the assumption that $\theta_{s,a_i}^{(\infty)}$ is a constant, and thus we can conclude that

$$\lim_{t \to +\infty} \theta_{s,a_i}^{(t)} = -\infty, \quad \forall a_i \in I_-^{i,s}. \qquad \square$$

**Lemma 18.** $\forall a_i^+ \in I_+^{i,s}$, for any $a \in I_0^{i,s}$, if there exists $t \geq T_1$ such that $\pi_i^{(t)}(a_i|s) \leq \pi_i^{(t)}(a_i^+|s)$, then for all $\tau \geq t$, $\pi_i^{(\tau)}(a_i|s) \leq \pi_i^{(\tau)}(a_i^+|s)$

*Proof.* We will prove by induction. Suppose for a certain $\tau \geq t$, it holds that $\pi_i^{(\tau)}(a_i|s) \leq \pi_i^{(\tau)}(a_i^+|s)$, then:

$$\frac{\partial \Phi(\theta^{(\tau)})}{\partial \theta_{s,a_i^+}} = \frac{1}{1-\gamma} d^{(\tau)}(s)\pi_i^{(\tau)}(a_i^+|s)\overline{A_i^{(\tau)}}(s,a_i^+)$$

$$\geq \frac{1}{1-\gamma} d^{(\tau)}(s)\pi_i^{(\tau)}(a_i|s)\overline{A_i^{(\tau)}}(s,a_i^+)$$

$$\geq \frac{1}{1-\gamma} d^{(\tau)}(s)\pi_i^{(\tau)}(a_i|s)\overline{A_i^{(\tau)}}(s,a_i)$$

$$= \frac{\partial \Phi(\theta^{(\tau)})}{\partial \theta_{s,a_i}}$$

Since $\pi_i^{(\tau)}(a_i|s) \leq \pi_i^{(\tau)}(a_i^+|s) \implies \theta_{s,a_i}^{(\tau)} \leq \theta_{s,a_i^+}^{(\tau)}$, we have:

$$\theta_{s,a_i^+}^{(\tau+1)} = \theta_{s,a_i^+}^{(\tau)} + \eta \frac{\partial \Phi(\theta^{(\tau)})}{\partial \theta_{s,a_i^+}} \geq \theta_{s,a_i}^{(\tau)} + \eta \frac{\partial \Phi(\theta^{(\tau)})}{\partial \theta_{s,a_i}} = \theta_{s,a_i}^{(\tau+1)}$$

Thus $\pi_i^{(\tau+1)}(a_i|s) \leq \pi_i^{(\tau+1)}(a_i^+|s)$ also holds, which completes the proof. $\square$

**Lemma 19.** $I_+^{i,s} = \emptyset$.

*Proof.* We will prove by contradiction. If $I_+^{i,s} \neq \emptyset$, select an arbitrary $a_i^+ \in I_+^{i,s}$ and define

$$B_0^{i,s}(a_i^+) := \{a_i \in I_0^{i,s} \mid \pi_i^{(t)}(a_i|s) \leq \pi_i^{(t)}(a_i^+|s), \forall t \geq T_1\}.$$

From Lemma 17, we have that for any $a_i \in I_-^{i,s}$ $\lim_{t\to+\infty} \frac{\pi_i^{(t)}(a_i|t)}{\pi_i^{(t)}(a_i^+|t)} = 0$, thus there exists $T_2 > T_1$ such that for any $t \geq T_2$,

$$\frac{\pi_i^{(t)}(a_i|t)}{\pi_i^{(t)}(a_i^+|t)} \leq \frac{(1-\gamma)\Delta}{16|\mathcal{A}_i|}, \quad \forall a_i \in I_-^{i,s}.$$

Additionally, since for any $a_i \in I_0^{i,s}$, $\lim_{t\to+\infty} \overline{A_i^{(t)}}(s,a_i) = 0$, there exists $T_3 > T_1$ such that for any $t \geq T_3$,

$$\overline{A_i^{(t)}}(s,a_i) \geq \frac{-\Delta}{16|\mathcal{A}_i|}, \quad \forall a_i \in I_0^{i,s}.$$

Thus, for $t \geq \max\{T_2, T_3\}$, from the fact that $\sum_{a_i} \pi_i^{(t)}(a_i|s)\overline{A_i^{(t)}}(s,a_i) = 0$, we have:

$$0 = \sum_{a_i \in I_0^{i,s}} \pi_i^{(t)}(a_i|s)\overline{A_i^{(t)}}(s,a_i) + \sum_{a_i \in I_+^{i,s}} \pi_i^{(t)}(a_i|s)\overline{A_i^{(t)}}(s,a_i) + \sum_{a_i \in I_-^{i,s}} \pi_i^{(t)}(a_i|s)\overline{A_i^{(t)}}(s,a_i)$$

$$\geq \sum_{a_i \in I_0^{i,s} \setminus B_0^{i,s}(a_i^+)} \pi_i^{(t)}(a_i|s)\overline{A_i^{(t)}}(s,a_i) + \sum_{a_i \in B_0^{i,s}(a_i^+)} \pi_i^{(t)}(a_i|s)\overline{A_i^{(t)}}(s,a_i)$$

$$+ \pi_i^{(t)}(a_i^+|s)\overline{A_i^{(t)}}(s,a_i^+) + \sum_{a_i \in I_-^{i,s}} \pi_i^{(t)}(a_i|s)\overline{A_i^{(t)}}(s,a_i)$$

$$\geq \sum_{a_i \in I_0^{i,s} \setminus B_0^{i,s}(a_i^+)} \pi_i^{(t)}(a_i|s)\overline{A_i^{(t)}}(s,a_i) + \sum_{a_i \in B_0^{i,s}(a_i^+)} \pi_i^{(t)}(a_i|s)\frac{-\Delta}{16|\mathcal{A}_i|}$$

$$+ \pi_i^{(t)}(a_i^+|s)\overline{A_i^{(t)}}(s,a_i^+) + \sum_{a_i \in I_-^{i,s}} \frac{(1-\gamma)\Delta}{16|\mathcal{A}_i|}\pi_i^{(t)}(a_i^+|s)\overline{A_i^{(t)}}(s,a_i)$$

$$\geq \sum_{a_i \in I_0^{i,s} \setminus B_0^{i,s}(a_i^+)} \pi_i^{(t)}(a_i|s)\overline{A_i^{(t)}}(s,a_i) + |\mathcal{A}_i|\pi_i^{(t)}(a_i|s)\frac{-\Delta}{16|\mathcal{A}_i|}$$

$$+ \pi_i^{(t)}(a_i^+|s)\frac{\Delta}{4} + |\mathcal{A}_i|\frac{(1-\gamma)\Delta}{16|\mathcal{A}_i|}\pi_i^{(t)}(a_i^+|s)\frac{-1}{1-\gamma}$$

$$\geq \sum_{a_i \in I_0^{i,s} \setminus B_0^{i,s}(a_i^+)} \pi_i^{(t)}(a_i|s)\overline{A_i^{(t)}}(s,a_i) + \pi_i^{(t)}(a_i^+|s)\frac{\Delta}{8}$$

$$\implies \sum_{a_i \in I_0^{i,s} \setminus B_0^{i,s}(a_i^+)} \pi_i^{(t)}(a_i|s)\overline{A_i^{(t)}}(s,a_i) < 0.$$

Thus for $t \geq \max\{T_2, T_3\}$,

$$\sum_{a_i \in I_0^{i,s} \setminus B_0^{i,s}(a_i^+)} \theta_{s,a_i}^{(t+1)} = \sum_{a_i \in I_0^{i,s} \setminus B_0^{i,s}(a_i^+)} \theta_{s,a_i}^{(t)} + \eta \frac{1}{1-\gamma} d^{(t)}(s) \sum_{a_i \in I_0^{i,s} \setminus B_0^{i,s}(a_i^+)} \pi_i^{(t)}(a_i|s) \overline{A_i^{(t)}}(s, a_i)$$
$$< \sum_{a_i \in I_0^{i,s} \setminus B_0^{i,s}(a_i^+)} \theta_{s,a_i}^{(t)},$$

which leads to the fact that $\sum_{a_i \in I_0^{i,s} \setminus B_0^{i,s}(a_i^+)} \theta_{s,a_i}^{(t)}$ is bounded from above. Further, from Lemma 17, $\theta_{s,a_i^+}^{(t)}$ is bounded from below, thus the value

$$\frac{\sum_{a_i \in I_0^{i,s} \setminus B_0^{i,s}(a_i^+)} \pi_i^{(t)}(a_i|s)}{\pi_i^{(t)}(a_i^+|s)}$$

is bounded from above. However from Corollary 16,

$$\lim_{t \to +\infty} \frac{\sum_{a_i \in I_0^{i,s}} \pi_i^{(t)}(a_i|s)}{\pi_i^{(t)}(a_i^+|s)} = +\infty.$$

Thus

$$\lim_{t \to +\infty} \frac{\sum_{a_i \in B_0^{i,s}(a_i^+)} \pi_i^{(t)}(a_i|s)}{\pi_i^{(t)}(a_i^+|s)} = +\infty,$$

which contradicts the fact that

$$\pi_i^{(t)}(a_i|s) \leq \pi_i^{(t)}(a_i^+|s), \quad \forall a_i \in B_0^{i,s}(a_i^+)$$

and finishes the proof by contradiction. $\qquad\square$

Lemma 19 directly implies asymptotic convergence for gradient play as state in Theorem 4.

*Remark* 5. **(Discussion on the isolated stationary points assumption)** The proof of Theorem 4 resembles the technique used in [2] for the single agent case, which relies heavily on the fact that the sequence of $Q$-functions $Q^{(t)}(s, a)$ obtains a limit $Q^{(\infty)}(s, a)$. The existence of such a limit in the single agent case follows from the monotonicity of the $Q$-functions. However, generalizing this proof to the multi-agent case requires the assumption that the sequence of averaged $Q$-functions $\overline{Q_i^{(t)}}(s, a_i)$ (which can be non-monotonic, see, e.g., Figure 3 in Appendix) has a limit $\overline{Q_i^{(\infty)}}(s, a_i)$, which is not necessarily true in general. For instance, if the set of stationary policies $\mathcal{SP} := \{\pi : \pi_i(a_i|s)\overline{A_i^\pi}(s, a_i) = 0, \forall s \in \mathcal{S}, a_i \in \mathcal{A}, i = 1, 2, \ldots, n\}$ is not isolated, one cannot rule out the possibility that (natural) gradient play will not converge to a fixed point $\pi^{(\infty)}$ (see e.g. [1] for counterexamples). Consequently, $\overline{Q_i^{(t)}}(s, a_i)$ might not converge to a single value. For the above reasons, we assume the stationary policies are isolated to ensure that $\pi^{(t)}$ converges to a fixed stationary policy $\pi^{(\infty)}$ and thus $\overline{Q_i^{(t)}}(s, a_i)$ obtains a limit. We believe that this assumption is a conservative condition that is sufficient to imply asymptotic convergence. It remains an interesting open question to establish convergence without this assumption.

### E.0.2 Asymptotic convergence for natural gradient play

The asymptotic convergence for natural gradient play is easier to establish compared with gradient play.

From Lemma 20 and the assumption that $\phi(s, a)$ is upper-bounded, we know

$$\lim_{t \to +\infty} \sum_{a_i} \pi_i^{(t)}(a_i|s) \exp\left(\frac{\eta \overline{A_i^{(t)}}(s, a_i)}{1-\gamma}\right) = 1, \; \forall s, i = 1, 2, \ldots, n.$$

Since

$$\sum_{a_i} \pi_i^{(t)}(a_i|s) \exp\left(\frac{\eta \overline{A_i^{(t)}}(s,a_i)}{1-\gamma}\right) \geq \sum_{a_i} \pi_i^{(t)}(a_i|s) \left(1 + \left(\frac{\eta \overline{A_i^{(t)}}(s,a_i)}{1-\gamma}\right) + \frac{1}{4}\left(\frac{\eta \overline{A_i^{(t)}}(s,a_i)}{1-\gamma}\right)^2\right)$$

$$(e^x \geq 1 + x + \frac{x^2}{4} \text{ for } |x| \leq 1)$$

$$= 1 + \frac{\eta^2}{4(1-\gamma)^2} \sum_{a_i} \pi_i^{(t)}(a_i|s) \overline{A_i^{(t)}}(s,a_i)^2$$

$$\implies \lim_{t\to+\infty} \sum_{a_i} \pi_i^{(t)}(a_i|s) \overline{A_i^{(t)}}(s,a_i)^2 = 0$$

$$\implies \lim_{t\to+\infty} \pi_i^{(t)}(a_i|s) \overline{A_i^{(t)}}(s,a_i) = 0, \ \forall s, a_i, i = 1,2,\ldots,n$$

$$\implies \lim_{t\to+\infty} \|\nabla_\theta \Phi(\theta^{(t)})\|_2 = 0$$

Similar to the proof for gradient play, from the assumption that stationary points are isolated, we can conclude that $\pi^{(t)}$ converges to some stationary policy $\pi^{(\infty)}$, and we can define $\overline{Q_i^{(\infty)}}(s,a_i), \overline{A_i^{(\infty)}}(s,a_i)$ accordingly. Asymptotic convergence is equivalent to

$$I_+^{i,s} := \left\{a_i : \overline{A_i^{(\infty)}}(s,a_i) > 0\right\} = \emptyset, \quad \forall s, \ i = 1,2,\ldots,n$$

We prove by contradiction. Suppose there exists $a_i^+$ such that $\overline{A_i^{(\infty)}}(s,a_i^+) > 0$. From $\lim_{t\to+\infty} \pi_i^{(t)}(a_i|s)\overline{A_i^{(t)}}(s,a_i) = 0$, we have that $\lim_{t\to+\infty} \pi_i^{(t)}(a_i^+|s) = 0$.

Select $a_i^0$ such that $\lim_{t\to+\infty} \pi_i^{(t)}(a_i^0|s) > 0$. From $\lim_{t\to+\infty} \pi_i^{(t)}(a_i|s)\overline{A_i^{(t)}}(s,a_i) = 0$, we have that $\lim_{t\to+\infty} \overline{A_i^{(t)}}(s,a_i^0) = 0$. Thus there exists $\Delta > 0$ and $T$ such that for $t > T$,

$$\overline{A_i^{(t)}}(s,a_i^+) > \Delta, \overline{A_i^{(t)}}(s,a_i^0) < \frac{\Delta}{2}$$

Thus from natural gradient play scheme (10)

$$\frac{\pi_i^{(t)}(a_i^+|s)}{\pi_i^{(t)}(a_i^0|s)} = \frac{\pi_i^{(T)}(a_i^+|s)}{\pi_i^{(T)}(a_i^0|s)} \exp\left(\frac{\eta}{1-\gamma}\sum_{\tau=T}^{t-1} \overline{A_i^{(\tau)}}(s,a_i^+) - \overline{A_i^{(\tau)}}(s,a_i^0)\right) \geq \frac{\pi_i^{(T)}(a_i^+|s)}{\pi_i^{(T)}(a_i^0|s)},$$

which contradict the fact that $\lim_{t\to+\infty} \frac{\pi_i^{(t)}(a_i^+|s)}{\pi_i^{(t)}(a_i^0|s)} = 0$, and thus completes the proof.

# F  Proof of Theorem 5

## F.1  Proof of Theorem 5 (Gradient play part)

*Proof of Theorem 5, gradient play.* From Lemma 29, $\Phi$ is $\beta$-smooth with $\beta = \frac{6n}{(1-\gamma)^3}$, we have that:

$$\Phi(\theta^{(t+1)}) - \Phi(\theta^{(t)}) \geq \left\langle \nabla\Phi(\theta^{(t)}), \theta^{(t+1)} - \theta^{(t)} \right\rangle - \frac{\beta}{2}\|\theta^{(t+1)} - \theta^{(t)}\|^2$$

$$= (\eta - \frac{\beta\eta^2}{2})\|\nabla\Phi(\theta^{(t)})\|^2$$

$$\geq \frac{\eta}{2}\|\nabla\Phi(\theta^{(t)})\|^2$$

Summing over $t$ we get:

$$\frac{\phi_{\max} - \phi_{\min}}{1 - \gamma} \geq \Phi(\theta^{(T)}) - \Phi(\theta^{(0)}) \geq \frac{\eta}{2} \sum_{t=0}^{T-1} \|\nabla\Phi(\theta^{(t)})\|^2$$

From Theorem 1 we have that

$$\|\nabla\Phi(\theta^{(t)})\| \geq \frac{c}{M\sqrt{\max_i |\mathcal{A}_i|}} \mathtt{NE\text{-}gap}(\theta^{(t)})$$

Thus

$$\frac{1}{T} \sum_{t=0}^{T-1} \mathtt{NE\text{-}gap}(\theta^{(t)})^2 \leq \frac{2\max_i |\mathcal{A}_i| M^2 (\phi_{\max} - \phi_{\min})}{(1-\gamma)c^2 \eta T}$$

which completes the proof. $\qquad\square$

## F.2 Proof of Theorem 5 (Natural gradient play part)

**Lemma 20.** *For $\eta \leq \frac{(1-\gamma)^2}{2n(\phi_{\max} - \phi_{\min})}$, running scheme (10) will guarantee that*

$$\Phi(\theta^{(t+1)}) - \Phi(\theta^{(t)}) \geq \frac{1}{\eta} \sum_{i=1}^{n} \sum_{s} d^{(t+1)}(s) \log Z_t^{i,s},$$

*where $Z_t^{i,s}$ is defined by*

$$Z_t^{i,s} := \sum_{a_i} \pi_i^{(t)}(a_i|s) \exp\left(\frac{\eta \overline{A_{i,\phi}^{(t)}}(s, a_i)}{1-\gamma}\right).$$

*Proof.* From performance difference lemma we have that

$$\Phi(\theta^{(t+1)}) - \Phi(\theta^{(t)}) = \frac{1}{1-\gamma} \sum_{s} d^{(t+1)}(s) \sum_{a} \left(\pi^{(t+1)}(a|s) - \pi^{(t)}(a|s)\right) A_\phi^{(t)}(s, a).$$

We define

$$\widetilde{A_{i,\phi}^{(t)}}(s, a_i) := \sum_{a_{-i}} \prod_{j=1}^{i-1} \pi_j^{(t+1)}(a_j|s) \prod_{j=i+1}^{n} \pi_j^{(t)}(a_j|s) A_\phi^{(t)}(s, a_i, a_{-i}). \tag{23}$$

Then

$$\Phi(\theta^{(t+1)}) - \Phi(\theta^{(t)}) = \frac{1}{1-\gamma} \sum_{s} d^{(t+1)}(s) \sum_{a} \left(\pi^{(t+1)}(a|s) - \pi^{(t)}(a|s)\right) A_\phi^{(t)}(s, a)$$

$$= \frac{1}{1-\gamma} \sum_{s} d^{(t+1)}(s) \sum_{a} \sum_{i=1}^{n} \left(\prod_{j=1}^{i} \pi_j^{(t+1)}(a_j|s) \prod_{j=i+1}^{n} \pi_j^{(t)}(a_j|s) - \prod_{j=1}^{i-1} \pi_j^{(t+1)}(a_j|s) \prod_{j=i}^{n} \pi_j^{(t)}(a_j|s)\right) A_\phi^{(t)}(s, a)$$

$$= \frac{1}{1-\gamma} \sum_{s} d^{(t+1)}(s) \sum_{i=1}^{n} \sum_{a_i} \left(\pi_i^{(t+1)}(a_i|s) - \pi_i^{(t)}(a_i|s)\right) \widetilde{A_{i,\phi}^{(t)}}(s, a_i)$$

$$= \frac{1}{1-\gamma} \sum_{s} d^{(t+1)}(s) \sum_{i=1}^{n} \sum_{a_i} \left(\pi_i^{(t+1)}(a_i|s) - \pi_i^{(t)}(a_i|s)\right) \overline{A_{i,\phi}^{(t)}}(s, a_i)$$

$$+ \frac{1}{1-\gamma} \sum_{s} d^{(t+1)}(s) \sum_{i=1}^{n} \sum_{a_i} \left(\pi_i^{(t+1)}(a_i|s) - \pi_i^{(t)}(a_i|s)\right) \left(\widetilde{A_{i,\phi}^{(t)}}(s, a_i) - \overline{A_{i,\phi}^{(t)}}(s, a_i)\right)$$

$$= \underbrace{\frac{1}{1-\gamma} \sum_{s} d^{(t+1)}(s) \sum_{i=1}^{n} \sum_{a_i} \pi_i^{(t+1)}(a_i|s) \overline{A_{i,\phi}^{(t)}}(s, a_i)}_{\text{Part A}}$$

$$+ \frac{1}{1-\gamma} \sum_s d^{(t+1)}(s) \sum_{i=1}^n \sum_{a_i} \left( \pi_i^{(t+1)}(a_i|s) - \pi_i^{(t)}(a_i|s) \right) \left( \widetilde{A_{i,\phi}^{(t)}}(s,a_i) - \overline{A_{i,\phi}^{(t)}}(s,a_i) \right).$$

$$\underbrace{\phantom{+ \frac{1}{1-\gamma} \sum_s d^{(t+1)}(s) \sum_{i=1}^n \sum_{a_i} \left( \pi_i^{(t+1)}(a_i|s) - \pi_i^{(t)}(a_i|s) \right) \left( \widetilde{A_{i,\phi}^{(t)}}(s,a_i) - \overline{A_{i,\phi}^{(t)}}(s,a_i) \right)}}_{\text{Part B}}$$

From scheme (10),

$$\overline{A_{i,\phi}^{(t)}}(s,a_i) = \frac{1-\gamma}{\eta} \left( \log \left( \frac{\pi_i^{(t+1)}(a_i|s)}{\pi_i^{(t)}(a_i|s)} \right) + \log \left( Z_t^{i,s} \right) \right)$$

Substitute this into Part A, we have

$$\text{Part A} = \frac{1}{\eta} \sum_s d^{(t+1)}(s) \sum_{i=1}^n \sum_{a_i} \pi_i^{(t+1)}(a_i|s) \overline{A_{i,\phi}^{(t)}}(s,a_i)$$

$$= \frac{1}{\eta} \sum_s d^{(t+1)}(s) \sum_{i=1}^n \sum_{a_i} \pi_i^{(t+1)}(a_i|s) \left( \log \left( \frac{\pi_i^{(t+1)}(a_i|s)}{\pi_i^{(t)}(a_i|s)} \right) + \log \left( Z_t^{i,s} \right) \right)$$

$$= \frac{1}{\eta} \sum_s \sum_{i=1}^n d^{(t+1)}(s) \text{KL}(\pi_{i,s}^{(t+1)}||\pi_{i,s}^{(t)}) + \frac{1}{\eta} \sum_s \sum_{i=1}^n d^{(t+1)}(s) \log \left( Z_t^{i,s} \right).$$

Further, we have that

$$\left| \widetilde{A_{i,\phi}^{(t)}}(s,a_i) - \overline{A_{i,\phi}^{(t)}}(s,a_i) \right|$$

$$= \left| \sum_{a_{-i}} \left( \prod_{j=1}^{i-1} \pi_j^{(t+1)}(a_j|s) - \prod_{j=1}^{i-1} \pi_j^{(t)}(a_j|s) \right) \prod_{j=i+1}^n \pi_j^{(t)}(a_j|s) A_\phi^{(t)}(s,a_i,a_{-i}) \right|$$

$$\leq \frac{\phi_{\max} - \phi_{\min}}{1-\gamma} \sum_{j=1}^{i-1} \|\pi_{j,s}^{(t+1)} - \pi_{j,s}^{(t)}\|_1$$

$$\leq \frac{\phi_{\max} - \phi_{\min}}{1-\gamma} \sum_{j=1}^n \|\pi_{j,s}^{(t+1)} - \pi_{j,s}^{(t)}\|_1.$$

Thus

$$|\text{Part B}| \leq \frac{1}{1-\gamma} \sum_s d^{(t+1)}(s) \sum_{i=1}^n \sum_{a_i} \left| \pi_i^{(t+1)}(a_i|s) - \pi_i^{(t)}(a_i|s) \right| \left| \widetilde{A_{i,\phi}^{(t)}}(s,a_i) - \overline{A_{i,\phi}^{(t)}}(s,a_i) \right|$$

$$\leq \frac{\phi_{\max} - \phi_{\min}}{(1-\gamma)^2} \sum_{i=1}^n \sum_s d^{(t+1)}(s) \sum_{a_i} \left| \pi_i^{(t+1)}(a_i|s) - \pi_i^{(t)}(a_i|s) \right| \sum_{j=1}^n \|\pi_{j,s}^{(t+1)} - \pi_{j,s}^{(t)}\|_1$$

$$\leq \frac{\phi_{\max} - \phi_{\min}}{(1-\gamma)^2} \sum_s d^{(t+1)}(s) \left( \sum_{i=1}^n \|\pi_{i,s}^{(t+1)} - \pi_{i,s}^{(t)}\|_1 \right)^2$$

$$\leq \frac{n(\phi_{\max} - \phi_{\min})}{(1-\gamma)^2} \sum_s d^{(t+1)}(s) \sum_{i=1}^n \|\pi_{i,s}^{(t+1)} - \pi_{i,s}^{(t)}\|_1^2$$

$$\leq \frac{2n(\phi_{\max} - \phi_{\min})}{(1-\gamma)^2} \sum_s d^{(t+1)}(s) \sum_{i=1}^n \text{KL}(\pi_{i,s}^{(t+1)}||\pi_{i,s}^{(t)}) \quad \text{(Pinsker's inequality)}$$

Thus, when $\eta \leq \frac{(1-\gamma)^2}{2n(\phi_{\max}-\phi_{\min})}$, we have that

$$\Phi(\theta^{(t+1)}) - \Phi(\theta^{(t)}) = \text{Part A} + \text{Part B}$$

$$\geq \left( \frac{1}{\eta} - \frac{2n(\phi_{\max} - \phi_{\min})}{(1-\gamma)^2} \right) \sum_s \sum_{i=1}^n d^{(t+1)}(s) \text{KL}(\pi_{i,s}^{(t+1)}||\pi_{i,s}^{(t)}) + \frac{1}{\eta} \sum_s \sum_{i=1}^n d^{(t+1)}(s) \log \left( Z_t^{i,s} \right)$$

$$\geq \frac{1}{\eta} \sum_s \sum_{i=1}^n d^{(t+1)}(s) \log\left(Z_t^{i,s}\right),$$

which completes the proof. $\square$

**Lemma 21.** *For* $\eta \leq (1-\gamma)^2$

$$\sum_{i=1}^n \sum_s d^{(t+1)}(s) \log Z_t^{i,s} \geq \frac{c\eta^2}{3M}\texttt{NE-gap}(\theta^{(t)})^2$$

*Proof.* From Lemma 13 we have that $\texttt{NE-gap}(\theta) \leq \frac{1}{1-\gamma} \max_i \max_{s,a_i} \overline{A_i^\theta}(s, a_i)$. On the other hand,

$$Z_t^{i,s} = \sum_{a_i} \pi_i^{(t)}(a_i|s) \exp\left(\frac{\eta \overline{A_i^{(t)}}(s, a_i)}{1-\gamma}\right)$$

$$= \sum_{a_i \notin \operatorname{argmax}_{a_i} \overline{Q_i^{(t)}}(s,a_i)} \pi_i^{(t)}(a_i|s) \exp\left(\frac{\eta \overline{A_i^{(t)}}(s, a_i)}{1-\gamma}\right) + \sum_{a_i \in \operatorname{argmax}_{a_i} \overline{Q_i^{(t)}}(s,a_i)} \pi_i^{(t)}(a_i|s) \exp\left(\frac{\eta \max_{a_i} \overline{A_i^{(t)}}(s, a_i)}{1-\gamma}\right)$$

$$\geq \sum_{a_i \notin \operatorname{argmax}_{a_i} \overline{Q_i^{(t)}}(s,a_i)} \pi_i^{(t)}(a_i|s)\left(1 + \frac{\eta \overline{A_i^{(t)}}(s, a_i)}{1-\gamma}\right)$$

$$+ \sum_{a_i \in \operatorname{argmax}_{a_i} \overline{Q_i^{(t)}}(s,a_i)} \pi_i^{(t)}(a_i|s)\left(1 + \frac{\eta \max_{a_i} \overline{A_i^{(t)}}(s, a_i)}{1-\gamma} + \frac{1}{2}\left(\frac{\eta \max_{a_i} \overline{A_i^{(t)}}(s, a_i)}{1-\gamma}\right)^2\right)$$

$$= \sum_{a_i} \pi_i^{(t)}(a_i|s) + \frac{\eta}{1-\gamma}\sum_{a_i} \pi_i^{(t)}(a_i|s)\overline{A_i^{(t)}}(s, a_i) + \frac{1}{2}\sum_{a_i \in \operatorname{argmax}_{a_i} \overline{Q_i^{(t)}}(s,a_i)} \pi_i^{(t)}(a_i|s)\left(\frac{\eta \max_{a_i} \overline{A_i^{(t)}}(s, a_i)}{1-\gamma}\right)^2$$

$$= 1 + \frac{1}{2}\sum_{a_i \in \operatorname{argmax}_{a_i} \overline{Q_i^{(t)}}(s,a_i)} \pi_i^{(t)}(a_i|s)\left(\frac{\eta \max_{a_i} \overline{A_i^{(t)}}(s, a_i)}{1-\gamma}\right)^2$$

$$\geq 1 + \frac{c}{2}\left(\frac{\eta \max_{a_i} \overline{A_i^{(t)}}(s, a_i)}{1-\gamma}\right)^2 .$$

Thus

$$\log(Z_t^{i,s}) \geq \log\left(1 + \frac{c}{2}\left(\frac{\eta \max_{a_i} \overline{A_i^{(t)}}(s, a_i)}{1-\gamma}\right)^2\right).$$

Because when $\eta \leq (1-\gamma)^2$, we have $\frac{c}{2}\left(\frac{\eta \max_{a_i} \overline{A_i^{(t)}}(s,a_i)}{1-\gamma}\right)^2 \leq \frac{1}{2}$, and that

$$\log(1+x) \geq \frac{2}{3}x, \quad \text{for } 0 \leq x \leq \frac{1}{2},$$

thus

$$\log(Z_t^{i,s}) \geq \log\left(1 + \frac{c}{2}\left(\frac{\eta \max_{a_i} \overline{A_i^{(t)}}(s, a_i)}{1-\gamma}\right)^2\right)$$

$$\geq \frac{c}{3}\left(\frac{\eta \max_{a_i}\overline{A_i^{(t)}}(s,a_i)}{1-\gamma}\right)^2.$$

Thus

$$\sum_{i=1}^n \sum_s d^{(t+1)}(s)\log Z_t^{i,s} \geq \frac{c}{3}\sum_{i=1}^n \sum_s d^{(t+1)}(s)\left(\frac{\eta \max_{a_i}\overline{A_i^{(t)}}(s,a_i)}{1-\gamma}\right)^2$$

$$\geq \frac{c\eta^2}{3M(1-\gamma)^2}\max_i \max_s \max_{a_i}\overline{A_i^{(t)}}(s,a_i)^2$$

$$\geq \frac{c\eta^2}{3M}\texttt{NE-gap}(\theta^{(t)})^2. \qquad \square$$

We are now ready to prove the bound for natural gradient play in Theorem 5.

*Proof of Theorem 5, natural gradient play.* Combining Lemma 20 and 21 we have

$$\Phi(\theta^{(t+1)}) - \Phi(\theta^{(t)}) \geq \frac{1}{\eta}\sum_{i=1}^n \sum_s d^{(t+1)}(s)\log Z_t^{i,s}$$

$$\geq \frac{c\eta}{3M}\texttt{NE-gap}(\theta^{(t)})^2$$

Summing over $t$ we have

$$\frac{\phi_{\max} - \phi_{\min}}{1-\gamma} \geq \Phi(\theta^{(T)}) - \Phi(\theta^{(0)}) \geq \frac{c\eta}{3M}\sum_{t=0}^{T-1}\texttt{NE-gap}(\theta^{(t)})^2,$$

thus

$$\frac{\sum_{t=0}^{T-1}\texttt{NE-gap}(\theta^{(t)})^2}{T} \leq \frac{3M(\phi_{\max} - \phi_{\min})}{(1-\gamma)c\eta T},$$

which completes the proof. $\qquad \square$

# G   Proof for $\log$-barrier regularization

## G.1   Proof of Theorem 6

We start with the following lemma:

**Lemma 22.** *Suppose $\theta$ is such that $\|\nabla_{\theta_i}\widetilde{J}_i(\theta)\|_2 \leq \lambda$, then $\texttt{NE-gap}_i(\theta) \leq \lambda M|\mathcal{A}_i|$, where $M$ is defined as in Assumption 1.*

*Proof.* From $\|\nabla_{\theta_i}\widetilde{J}_i(\theta)\|_2 \leq \frac{\lambda}{2}$ we have that

$$\frac{\partial \widetilde{J}_i(\theta)}{\partial \theta_{s,a_i}} = \frac{1}{1-\gamma}d_\theta(s)\pi_{\theta_i}(a_i|s)\overline{A_i^\theta}(s,a_i) + \lambda - \lambda|\mathcal{A}_i|\pi_{\theta_i}(a_i|s)$$

$$= \pi_{\theta_i}(a_i|s)\left(\frac{1}{1-\gamma}d_\theta(s)\overline{A_i^\theta}(s,a_i) - \lambda|\mathcal{A}_i|\right) + \lambda \leq \lambda$$

$$\implies \pi_{\theta_i}(a_i|s)\left(\frac{1}{1-\gamma}d_\theta(s)\overline{A_i^\theta}(s,a_i) - \lambda|\mathcal{A}_i|\right) \leq 0$$

$$\implies \frac{1}{1-\gamma}d_\theta(s)\overline{A_i^\theta}(s,a_i) - \lambda|\mathcal{A}_i| \leq 0$$

$$\implies \overline{A_i^\theta}(s,a_i) \leq \frac{\lambda|\mathcal{A}_i|(1-\gamma)}{d_\theta(s)} \leq \lambda|\mathcal{A}_i|(1-\gamma)M$$

Thus,

$$\texttt{NE-gap}_i(\theta) = \sup_{\theta_i^*} J_i(\theta_i^*, \theta_{-i}) - J_i(\theta_i, \theta_{-i}) = \frac{1}{1-\gamma} \sum_{s,a_i} d_{\theta^*}(s) \pi_{\theta_i^*}(a_i|s) \overline{A_i^\theta}(s, a_i)$$

$$\leq \frac{1}{1-\gamma} \sum_{s,a_i} d_{\theta^*}(s) \max_{s,a_i} \overline{A_i^\theta}(s, a_i)$$

$$\leq \frac{1}{1-\gamma} \sum_{s,a_i} d_{\theta^*}(s) \lambda |\mathcal{A}_i|(1-\gamma)M$$

$$\leq \lambda |\mathcal{A}_i| M. \qquad \square$$

Lemma 22 implies that any policy with gradient norm smaller than $\lambda$ is also a $\lambda M \max_i |\mathcal{A}_i|$-NE. Thus by properly choosing $\lambda$, agents can find a $\epsilon$-NE by running gradient play.

We now prove Theorem 6.

*Proof of Theorem 6.* From Lemma 30, $\widetilde{\Phi}$ is $\beta$-smooth with $\beta = \frac{6n}{(1-\gamma)^3} + 2\lambda \max_i |\mathcal{A}_i|$, we have that:

$$\widetilde{\Phi}(\theta^{(t+1)}) - \widetilde{\Phi}(\theta^{(t)}) \geq \left\langle \nabla \widetilde{\Phi}(\theta^{(t)}), \theta^{(t+1)} - \theta^{(t)} \right\rangle - \frac{\beta}{2} \|\theta^{(t+1)} - \theta^{(t)}\|^2$$

$$= (\eta - \frac{\beta \eta^2}{2}) \|\nabla \Phi(\theta^{(t)})\|^2$$

$$\geq \frac{\eta}{2} \|\nabla \Phi(\theta^{(t)})\|^2$$

For $\theta^{(0)} = \mathbf{0}$, summing over $t$ we get:

$$\frac{\phi_{\max} - \phi_{\min}}{1-\gamma} \geq \widetilde{\Phi}(\theta^{(T)}) - \widetilde{\Phi}(\theta^{(0)}) \geq \frac{\eta}{2} \sum_{t=0}^{T-1} \|\nabla \widetilde{\Phi}(\theta^{(t)})\|^2$$

Thus,

$$\min_{0 \leq t \leq T-1} \|\nabla \widetilde{\Phi}(\theta^{(t)})\| \leq \frac{2(\phi_{\max} - \phi_{\min})}{(1-\gamma)\eta T}.$$

Thus for

$$T \geq \frac{2(\phi_{\max} - \phi_{\min})}{(1-\gamma)\eta \lambda^2}$$

$$= \frac{2 \max_i |\mathcal{A}_i|^2 (\phi_{\max} - \phi_{\min}) M^2}{(1-\gamma)\eta \epsilon^2},$$

it can be guaranteed that

$$\min_{0 \leq t \leq T-1} \|\nabla \widetilde{\Phi}(\theta^{(t)})\| \leq \lambda = \frac{\epsilon}{\max_i |\mathcal{A}_i| M}.$$

Then applying Lemma 22 completes the proof. $\qquad \square$

### G.2 Proof of Theorem 7

For notational simplicity, we define the following variables:

$$f_i^{(t)}(s, a_i) := \frac{1}{1-\gamma} \overline{A_i^{(t)}}(s, a_i) + \frac{\lambda}{d^{(t)}(s)\pi_i^{(t)}(a_i|s)} - \frac{\lambda |\mathcal{A}_i|}{d^{(t)}(s)}$$

$$Z_t^{i,s} := \sum_{a_i} \pi_i^{(t)}(a_i|s) \exp\left( \eta f^{(t)}(s, a_i) \right)$$

$$\Delta_i^{(t)}(s, a_i) := \frac{\pi_i^{(t+1)}(a_i|s)}{\pi_i^{(t)}(a_i|s)} - 1$$

**Lemma 23.**

$$\sum_{a_i} \pi_i^{(t)}(a_i|s) f_i^{(t)}(s, a_i) = 0$$

$$\sum_{a_i} \pi_i^{(t)}(a_i|s) \Delta_i^{(t)}(s, a_i) = 0$$

$$Z_t^{i,s} \geq 1$$

*Proof.* From the definition of $f_i^{(t)}(s, a_i), \Delta_i^{(t)}(s, a_i)$,

$$\sum_{a_i} \pi_i^{(t)}(a_i|s) f_i^{(t)}(s, a_i)$$

$$= \frac{1}{1-\gamma} \sum_{a_i} \pi_i^{(t)}(a_i|s) \overline{A_i^{(t)}}(s, a_i) + \lambda \sum_{a_i} \pi_i^{(t)}(a_i|s) \frac{1}{d^{(t)}(s)\pi^{(t)}(a_i|s)} - \frac{\lambda|\mathcal{A}_i|}{d^{(t)}(s)} \sum_{a_i} \pi_i^{(t)}(a_i|s)$$

$$= \frac{1}{d^{(t)}(s)} (\lambda|\mathcal{A}_i| - \lambda|\mathcal{A}_i|) = 0$$

$$\sum_{a_i} \pi_i^{(t)}(a_i|s) \Delta_i^{(t)}(s, a_i) = \sum_{a_i} \pi_i^{(t)}(a_i|s) \left( \frac{\pi_i^{(t+1)}(a_i|s)}{\pi_i^{(t)}(a_i|s)} - 1 \right)$$

$$= \sum_{a_i} \pi_i^{(t+1)}(a_i|s) - \sum_{a_i} \pi_i^{(t)}(a_i|s) = 1 - 1 = 0$$

Using the fact that $e^x \geq 1 + x$,

$$Z_t^{i,s} = \sum_{a_i} \pi_i^{(t)}(a_i|s) \exp\left( \eta f_i^{(t)}(s, a_i) \right)$$

$$\geq \sum_{a_i} \pi_i^{(t)}(a_i|s) \left( 1 + \eta f_i^{(t)}(s, a_i) \right)$$

$$\geq \sum_{a_i} \pi_i^{(t)}(a_i|s) + \eta \sum_{a_i} \pi_i^{(t)}(a_i|s) f_i^{(t)}(s, a_i) \geq 1. \qquad \square$$

**Lemma 24.** *For* $\eta \leq \frac{1}{15\left( \frac{1}{(1-\gamma)^2} + \lambda|\mathcal{A}_i|M \right)}, \theta^{(0)} = \mathbf{0}$, *running scheme* (17) *will guarantee that*

$$\pi_i^{(t)}(a_i|s) \geq \frac{\lambda}{4\left( \lambda|\mathcal{A}_i|M + \frac{1}{(1-\gamma)^2} \right)}.$$

*Proof.* We will prove by induction. For $\theta^0 = \mathbf{0}$ apparently $\pi^{(0)}$ satisfies the lower bound. Suppose that

$$\pi_i^{(t)}(a_i|s) \geq \frac{\lambda}{4\left( \lambda|\mathcal{A}_i|M + \frac{1}{(1-\gamma)^2} \right)},$$

then from the definition of $f_i^{(t)}(s, a)$ we have that

$$-\frac{1}{(1-\gamma)^2} - \lambda|\mathcal{A}_i|M \leq f_i^{(t)}(s, a_i) \leq 5\left( \frac{1}{(1-\gamma)^2} + \lambda|\mathcal{A}_i|M \right)$$

Thus

$$-\frac{1}{15} \leq \eta f_i^{(t)}(s, a_i) \leq \frac{1}{3},$$

which leads to the fact that

$$Z_t^{i,s} = \sum_{a_i} \pi_i^{(t)}(a_i|s) \exp\left( \eta f_i^{(t)}(s, a_i) \right)$$

$$\leq \sum_{a_i} \pi_i^{(t)}(a_i|s) \left(1 + \left(\eta f_i^{(t)}(s, a_i)\right) + \left(\eta f_i^{(t)}(s, a_i)\right)^2\right) \tag{24}$$

$$\left(e^x \leq 1 + x + x^2, \text{for} -\frac{1}{15} \leq x \leq \frac{1}{3}\right)$$

$$= 1 + \sum_{a_i} \pi_i^{(t)}(a_i|s) \left(\eta f_i^{(t)}(s, a_i)\right)^2 \tag{25}$$

$$\leq 1 + \frac{1}{3^2} = \frac{10}{9}.$$

Thus we have that

$$\frac{\pi_i^{(t+1)}(a_i|s)}{\pi_i^{(t)}(a_i|s)} = \frac{\exp\left(\eta f_i^{(t)}(s, a_i)\right)}{Z_t^{i,s}} \geq \frac{1 + \eta f^{(t)}(s, a_i)}{Z_t^{i,s}} \geq \frac{1 - \frac{1}{15}}{\frac{10}{9}} = \frac{21}{25}.$$

Thus, for $a_i$ such that $\pi_i^{(t)}(a_i|s) \geq \frac{\lambda}{3\left(\lambda|\mathcal{A}_i|M + \frac{1}{(1-\gamma)^2}\right)}$, we have

$$\pi_i^{(t+1)}(a_i|s) \geq \frac{21}{25} \frac{\lambda}{3\left(\lambda|\mathcal{A}_i|M + \frac{1}{(1-\gamma)^2}\right)} \geq \frac{\lambda}{4\left(\lambda|\mathcal{A}_i|M + \frac{1}{(1-\gamma)^2}\right)}.$$

On the other hand, for $a_i$ such that $\pi_i^{(t)}(a_i|s) < \frac{\lambda}{3\left(\lambda|\mathcal{A}_i|M + \frac{1}{(1-\gamma)^2}\right)}$, we have

$$f_i^{(t)}(s, a_i) \geq -\frac{1}{(1-\gamma)^2} - \lambda|\mathcal{A}_i|M + 3\left(\frac{1}{(1-\gamma)^2} + \lambda|\mathcal{A}_i|M\right) = 2\left(\frac{1}{(1-\gamma)^2} + \lambda|\mathcal{A}_i|M\right),$$

From inequality (25) we have that

$$Z_{i,t} \leq 1 + \eta^2 \sum_{a_i} \pi_i^{(t)}(a_i|s) f_i^{(t)}(s, a_i)^2$$

$$\leq 1 + 25\eta^2 \left(\frac{1}{(1-\gamma)^2} + \lambda|\mathcal{A}_i|M\right)^2$$

$$\leq 1 + \frac{5}{3}\eta \left(\frac{1}{(1-\gamma)^2} + \lambda|\mathcal{A}_i|M\right)$$

Thus

$$\frac{\pi_i^{(t+1)}(a_i|s)}{\pi_i^{(t)}(a_i|s)} = \frac{\exp\left(\eta f_i^{(t)}(s, a_i)\right)}{Z_t^{i,s}} \geq \frac{1 + \left(\eta f_i^{(t)}(s, a_i)\right)}{Z_t^{i,s}} \geq \frac{1 + 2\eta\left(\frac{1}{(1-\gamma)^2} + \lambda|\mathcal{A}_i|M\right)}{1 + \frac{5}{3}\eta\left(\frac{1}{(1-\gamma)^2} + \lambda|\mathcal{A}_i|M\right)} \geq 1,$$

then according to the induction assumption, we have

$$\pi_i^{(t+1)}(a_i|s) \geq \frac{\lambda}{4\left(\lambda|\mathcal{A}_i|M + \frac{1}{(1-\gamma)^2}\right)},$$

which completes the proof of the lemma. $\square$

**Corollary 25.** *Under the condition of Lemma 24, running* (17) *will guarantee that*

$$-\frac{1}{15} \leq \eta f_i^{(t)}(s, a_i) \leq \frac{1}{3}, \quad Z_t^{i,s} \leq \frac{10}{9}, \quad -\frac{1}{5} \leq \Delta_i^{(t)}(s, a_i) \leq \frac{1}{2},$$

*Proof.* The first two inequalities are proved in the proof of Lemma 24, we only need to show $-\frac{1}{5} \leq \Delta_i^{(t)}(s, a_i) \leq \frac{1}{2}$. In the proof of Lemma 24, we have already shown that

$$\frac{\pi_i^{(t+1)}(a_i|s)}{\pi_i^{(t)}(a_i|s)} = \frac{\exp\left(\eta f_i^{(t)}(s, a_i)\right)}{Z_t^{i,s}} \geq \frac{1 + \eta f^{(t)}(s, a_i)}{Z_t^{i,s}} \geq \frac{1 - \frac{1}{15}}{\frac{10}{9}} = \frac{21}{25} \geq \frac{4}{5},$$

thus
$$\Delta_i^{(t)}(s, a_i) \geq \frac{4}{5} - 1 = -\frac{1}{5}.$$

On the other hand,
$$\frac{\pi_i^{(t+1)}(a_i|s)}{\pi_i^{(t)}(a_i|s)} = \frac{\exp\left(\eta f_i^{(t)}(s, a_i)\right)}{Z_t^{i,s}} \leq \exp\left(\eta f_i^{(t)}(s, a_i)\right) \leq \exp\left(\frac{1}{3}\right) \leq \frac{3}{2},$$

thus
$$\Delta_i^{(t)}(s, a_i) \leq \frac{3}{2} - 1 = \frac{1}{2},$$

which completes the proof of the corollary. $\qquad\square$

**Lemma 26.**
$$\widetilde{\Phi}(\theta^{(t+1)}) - \widetilde{\Phi}(\theta^{(t)}) \geq \left(\frac{1}{2\eta} - 4\lambda \max_i |\mathcal{A}_i| M^2 - \frac{4M}{(1-\gamma)^2} - \frac{3nM}{(1-\gamma)^3}\right) \sum_{i=1}^{n} \sum_{s, a_i} d^{(t)}(s) \pi_i^{(t)}(a_i|s) \Delta_i^{(t)}(s, a_i)^2$$

*Proof.* Let $\widetilde{\theta}^{i,(t)}$ be defined as:
$$\widetilde{\theta}^{i,(t)} := \left(\theta_1^{(t)}, \dots, \theta_{i-1}^{(t)}, \theta_i^{(t+1)}, \dots, \theta_n^{(t+1)}\right).$$

Then we have that
$$
\begin{aligned}
\Phi(\theta^{(t+1)}) - \Phi(\theta^{(t)}) &= \sum_{i=1}^{n} \Phi(\widetilde{\theta}^{i,(t)}) - \Phi(\widetilde{\theta}^{i+1,(t)}) \\
&= \sum_{i=1}^{n} J_i(\widetilde{\theta}^{i,(t)}) - J_i(\widetilde{\theta}^{i+1,(t)}) \\
&= \frac{1}{1-\gamma} \sum_{i=1}^{n} \sum_s d_{\widetilde{\theta}^{i,(t)}}(s) \sum_{a_i} \pi_i^{(t+1)}(a_i|s) \overline{A_i^{\widetilde{\theta}^{i+1,(t)}}}(s, a_i) \quad \text{(Lemma 8)} \\
&= \frac{1}{1-\gamma} \sum_{i=1}^{n} \sum_s d_{\widetilde{\theta}^{i,(t)}}(s) \sum_{a_i} \left(\pi_i^{(t+1)}(a_i|s) - \pi_i^{(t)}(a_i|s)\right) \overline{Q_i^{\widetilde{\theta}^{i+1,(t)}}}(s, a_i).
\end{aligned}
$$

Thus
$$
\begin{aligned}
&\widetilde{\Phi}(\theta^{(t+1)}) - \widetilde{\Phi}(\theta^{(t)}) = \Phi(\theta^{(t+1)}) - \Phi(\theta^{(t)}) + \lambda \sum_i \sum_{s, a_i} \log\left(\frac{\pi_i^{(t+1)}(a_i|s)}{\pi_i^{(t)}(a_i|s)}\right) \\
&= \frac{1}{1-\gamma} \sum_{i=1}^{n} \sum_s d_{\widetilde{\theta}^{i,(t)}}(s) \sum_{a_i} \left(\pi_i^{(t+1)}(a_i|s) - \pi_i^{(t)}(a_i|s)\right) \overline{Q_i^{\widetilde{\theta}^{i+1,(t)}}}(s, a_i) + \lambda \sum_i \sum_{s, a_i} \log\left(1 + \Delta^{(t)}(s, a_i)\right) \\
&= \frac{1}{1-\gamma} \sum_{i=1}^{n} \sum_s d^{(t)}(s) \sum_{a_i} \left(\pi_i^{(t+1)}(a_i|s) - \pi_i^{(t)}(a_i|s)\right) \overline{Q_i^{(t)}}(s, a_i) \\
&\quad + \frac{1}{1-\gamma} \sum_{i=1}^{n} \sum_s d^{(t)}(s) \sum_{a_i} \left(\pi_i^{(t+1)}(a_i|s) - \pi_i^{(t)}(a_i|s)\right) \left(\overline{Q_i^{\widetilde{\theta}^{i+1,(t)}}}(s, a_i) - \overline{Q_i^{(t)}}(s, a_i)\right) \\
&\quad + \frac{1}{1-\gamma} \sum_{i=1}^{n} \sum_s \left(d_{\widetilde{\theta}^{i,(t)}}(s) - d^{(t)}(s)\right) \sum_{a_i} \left(\pi_i^{(t+1)}(a_i|s) - \pi_i^{(t)}(a_i|s)\right) \overline{Q_i^{\widetilde{\theta}^{i+1,(t)}}}(s, a_i) \\
&\quad + \lambda \sum_i \sum_{s, a_i} \log\left(1 + \Delta^{(t)}(s, a_i)\right)
\end{aligned}
$$

$$
= \underbrace{\frac{1}{1-\gamma} \sum_{i=1}^{n} \sum_{s} d^{(t)}(s) \sum_{a_i} \left( \pi_i^{(t+1)}(a_i|s) - \pi_i^{(t)}(a_i|s) \right) \overline{Q_i^{(t)}}(s, a_i) + \lambda \Delta_i^{(t)}(s, a_i)}_{\text{Part A}}
$$

$$
+ \underbrace{\lambda \sum_{i} \sum_{s,a_i} \log \left( 1 + \Delta^{(t)}(s, a_i) \right) - \Delta_i^{(t)}(s, a_i)}_{\text{Part B}}
$$

$$
+ \underbrace{\frac{1}{1-\gamma} \sum_{i=1}^{n} \sum_{s} d^{(t)}(s) \sum_{a_i} \left( \pi_i^{(t+1)}(a_i|s) - \pi_i^{(t)}(a_i|s) \right) \left( \overline{Q_i^{\tilde{\theta}^{i+1,(t)}}}(s, a_i) - \overline{Q_i^{(t)}}(s, a_i) \right)}_{\text{Part C}}
$$

$$
+ \underbrace{\frac{1}{1-\gamma} \sum_{i=1}^{n} \sum_{s} \left( d_{\tilde{\theta}^{i,(t)}}(s) - d^{(t)}(s) \right) \sum_{a_i} \left( \pi_i^{(t+1)}(a_i|s) - \pi_i^{(t)}(a_i|s) \right) \overline{Q_i^{\tilde{\theta}^{i+1,(t)}}}(s, a_i)}_{\text{Part D}}.
$$

We will now bound each part separately. We first get a lower bound for part A:

$$
\text{Part A} = \sum_{i=1}^{n} \sum_{s,a_i} d^{(t)}(s) \left[ \left( \pi_i^{(t+1)}(a_i|s) - \pi_i^{(t)}(a_i|s) \right) \left( \frac{1}{1-\gamma} \overline{A_i^{(t)}}(s, a_i) \right) + \frac{\lambda}{d^{(t)}(s)} \Delta_i^{(t)}(s, a_i) \right]
$$

$$
= \sum_{i=1}^{n} \sum_{s,a_i} d^{(t)}(s) \left[ \pi_i^{(t)}(a_i|s) \Delta_i^{(t)}(s, a_i) \left( \frac{1}{1-\gamma} \overline{A_i^{(t)}}(s, a_i) \right) + \frac{\lambda}{d^{(t)}(s)} \Delta_i^{(t)}(s, a_i) \right]
$$

$$
= \sum_{i=1}^{n} \sum_{s,a_i} d^{(t)}(s) \left[ \pi_i^{(t)}(a_i|s) \Delta_i^{(t)}(s, a_i) \left( \frac{1}{1-\gamma} \overline{A_i^{(t)}}(s, a_i) - \frac{\lambda|\mathcal{A}_i|}{d^{(t)}(s)} \right) + \frac{\lambda}{d^{(t)}(s)} \Delta_i^{(t)}(s, a_i) \right]
$$

$$
= \sum_{i=1}^{n} \sum_{s,a_i} d^{(t)}(s) \pi_i^{(t)}(a_i|s) \Delta_i^{(t)}(s, a_i) \left( \frac{1}{1-\gamma} \overline{A_i^{(t)}}(s, a_i) + \frac{\lambda}{d^{(t)}(s) \pi_i^{(t)}(a_i|s)} - \frac{\lambda|\mathcal{A}_i|}{d^{(t)}(s)} \right)
$$

$$
= \sum_{i=1}^{n} \sum_{s,a_i} d^{(t)}(s) \pi_i^{(t)}(a_i|s) \Delta_i^{(t)}(s, a_i) f_i^{(t)}(s, a_i)
$$

$$
= \sum_{i=1}^{n} \sum_{s,a_i} d^{(t)}(s) \pi_i^{(t)}(a_i|s) \Delta_i^{(t)}(s, a_i) \frac{1}{\eta} \left( \log \left( \frac{\pi_i^{(t+1)}(a_i|s)}{\pi_i^{(t)}(a_i|s)} \right) + \log(Z_t^{i,s}) \right)
$$

$$
= \frac{1}{\eta} \sum_{i=1}^{n} \sum_{s,a_i} d^{(t)}(s) \pi_i^{(t)}(a_i|s) \Delta_i^{(t)}(s, a_i) \log \left( 1 + \Delta_i^{(t)}(s, a_i) \right)
$$

$$
+ \frac{1}{\eta} \sum_{i=1}^{n} \sum_{s} d^{(t)}(s) \log(Z_t^{i,s}) \sum_{a_i} \pi_i^{(t)}(a_i|s) \Delta_i^{(t)}(s, a_i)
$$

$$
= \frac{1}{\eta} \sum_{i=1}^{n} \sum_{s,a_i} d^{(t)}(s) \pi_i^{(t)}(a_i|s) \Delta_i^{(t)}(s, a_i) \log \left( 1 + \Delta_i^{(t)}(s, a_i) \right)
$$

$$
= \frac{1}{\eta} \sum_{i=1}^{n} \sum_{s,a_i} d^{(t)}(s) \pi_i^{(t)}(a_i|s) \left| \Delta_i^{(t)}(s, a_i) \right| \left| \log \left( 1 + \Delta_i^{(t)}(s, a_i) \right) \right|
$$

From the boundedness of $\Delta_i^{(t)}(s, a_i)$ in Corollary 25, we have that

$$
\left| \log \left( 1 + \Delta_i^{(t)}(s, a_i) \right) \right| \geq \frac{1}{2} \left| \Delta_i^{(t)}(s, a_i) \right|
$$

Substitute this into the above inequalities, we get

$$\text{Part A} \geq \frac{1}{2\eta} \sum_{i=1}^{n} \sum_{s,a_i} d^{(t)}(s) \pi_i^{(t)}(a_i|s) \Delta_i^{(t)}(s,a_i)^2$$

Now we will give a lower bound for part B. Similarly, from the boundedness of $\Delta_i^{(t)}(s,a_i)$ in Corollary 25, we have that

$$\log\left(1 + \Delta_i^{(t)}(s,a_i)\right) - \Delta_i^{(t)}(s,a_i) \geq -\Delta_i^{(t)}(s,a_i)^2.$$

Thus

$$\text{Part B} = \lambda \sum_{i} \sum_{s,a_i} \log\left(1 + \Delta^{(t)}(s,a_i)\right) - \Delta_i^{(t)}(s,a_i) \geq -\lambda \sum_{i} \sum_{s,a_i} \Delta_i^{(t)}(s,a_i)^2.$$

Additionally, using Lemma 24,

$$\text{Part B} \geq -\lambda \sum_{i} \sum_{s,a_i} \Delta_i^{(t)}(s,a_i)^2$$

$$\geq -4\left(\lambda \max_i |\mathcal{A}_i| M + \frac{1}{(1-\gamma)^2}\right) \sum_{i} \sum_{s,a_i} \pi_i^{(t)}(a_i|s) \Delta_i^{(t)}(s,a_i)^2$$

$$\geq -4M\left(\lambda \max_i |\mathcal{A}_i| M + \frac{1}{(1-\gamma)^2}\right) \sum_{i} \sum_{s,a_i} d^{(t)}(s) \pi_i^{(t)}(a_i|s) \Delta_i^{(t)}(s,a_i)^2.$$

We will now move on to bound the absolute value of Part C.

$$|\text{Part C}| \leq \frac{1}{1-\gamma} \sum_{i=1}^{n} \sum_{s} d^{(t)}(s) \sum_{a_i} \left|\pi_i^{(t+1)}(a_i|s) - \pi_i^{(t)}(a_i|s)\right| \left|\overline{Q_i^{\widetilde{\theta}^{i+1,(t)}}}(s,a_i) - \overline{Q_i^{(t)}}(s,a_i)\right|.$$

Since

$$\left|\overline{Q_i^{\widetilde{\theta}^{i+1,(t)}}}(s,a_i) - \overline{Q_i^{(t)}}(s,a_i)\right| \leq \sum_{a_{-i}} \pi_{\widetilde{\theta}_{-i}^{i+1,(t)}}(a_{-i}|s) \left|Q^{\widetilde{\theta}^{i+1,(t)}}(s,a_i,a_{-i}) - Q^{(t)}(s,a_i,a_{-i})\right|$$

$$+ \sum_{a_{-i}} \left|\pi_{\widetilde{\theta}_{-i}^{i+1,(t)}}(a_{-i}|s) - \pi_{-i}^{(t)}(a_{-i}|s)\right| \left|Q^{(t)}(s,a_i)\right|$$

$$\leq \max_a \left|Q^{\widetilde{\theta}^{i+1,(t)}}(s,a) - Q^{(t)}(s,a)\right| + \frac{1}{1-\gamma} \sum_{j=1}^{n} \|\pi_{j,s}^{(t+1)} - \pi_{j,s}^{(t)}\|_1.$$

From Lemma 32,

$$\max_a \left|Q^{\widetilde{\theta}^{i+1,(t)}}(s,a) - Q^{(t)}(s,a)\right| \leq \frac{1}{(1-\gamma)^2} \max_s \|\pi_{\widetilde{\theta}_s^{i+1,(t)}} - \pi_{\theta_s^{(t)}}\|_1$$

$$\leq \frac{1}{(1-\gamma)^2} \max_s \sum_{j=1}^{n} \|\pi_{j,s}^{(t+1)} - \pi_{j,s}^{(t)}\|_1$$

Thus we have that

$$\left|\overline{Q_i^{\widetilde{\theta}^{i+1,(t)}}}(s,a_i) - \overline{Q_i^{(t)}}(s,a_i)\right| \leq \max_a \left|Q^{\widetilde{\theta}^{i+1,(t)}}(s,a) - Q^{(t)}(s,a)\right| + \frac{1}{1-\gamma} \sum_{j=1}^{n} \|\pi_{j,s}^{(t+1)} - \pi_{j,s}^{(t)}\|_1$$

$$\leq \frac{2}{(1-\gamma)^2} \max_s \sum_{j=1}^{n} \|\pi_{j,s}^{(t+1)} - \pi_{j,s}^{(t)}\|_1.$$

Thus

$$
|\text{Part C}| \leq \frac{1}{1-\gamma} \sum_{i=1}^{n} \sum_{s} d^{(t)}(s) \sum_{a_i} \left| \pi_i^{(t+1)}(a_i|s) - \pi_i^{(t)}(a_i|s) \right| \frac{2}{(1-\gamma)^2} \max_{s} \sum_{j=1}^{n} \|\pi_{j,s}^{(t+1)} - \pi_{j,s}^{(t)}\|_1
$$

$$
\leq \frac{2}{(1-\gamma)^3} \left( \max_{s} \sum_{j=1}^{n} \|\pi_{j,s}^{(t+1)} - \pi_{j,s}^{(t)}\|_1 \right) \cdot \sum_{i=1}^{n} \sum_{s} d^{(t)}(s) \sum_{a_i} \left| \pi_i^{(t+1)}(a_i|s) - \pi_i^{(t)}(a_i|s) \right|
$$

$$
\leq \frac{2}{(1-\gamma)^3} \left( \max_{s} \sum_{j=1}^{n} \|\pi_{j,s}^{(t+1)} - \pi_{j,s}^{(t)}\|_1 \right) \cdot \sum_{s} d^{(t)}(s) \sum_{j=1}^{n} \|\pi_{j,s}^{(t+1)} - \pi_{j,s}^{(t)}\|_1
$$

$$
\leq \frac{2}{(1-\gamma)^3} \left( \max_{s} \sum_{j=1}^{n} \|\pi_{j,s}^{(t+1)} - \pi_{j,s}^{(t)}\|_1 \right)^2
$$

From Cauchy-Schwarz inequality,

$$
\left( \sum_{j=1}^{n} \|\pi_{j,s}^{(t+1)} - \pi_{j,s}^{(t)}\|_1 \right)^2 = \left( \sum_{j=1}^{n} \sum_{a_j} \pi_i^{(t)}(a_j|s) \left| \Delta_j^{(t)}(s,a_j) \right| \right)^2
$$

$$
\leq \left( \sum_{j=1}^{n} \sum_{a_j} \pi_i^{(t)}(a_j|s) \right) \left( \sum_{j=1}^{n} \sum_{a_j} \pi_j^{(t)}(a_j|s) \Delta_j^{(t)}(s,a_j)^2 \right)
$$

$$
= n \sum_{j=1}^{n} \sum_{a_j} \pi_j^{(t)}(a_j|s) \Delta_j^{(t)}(s,a_j)^2
$$

Thus

$$
|\text{Part C}| \leq \frac{2n}{(1-\gamma)^3} \sum_{i=1}^{n} \sum_{a_i} \pi_i^{(t)}(a_i|s) \Delta_i^{(t)}(s,a_i)^2
$$

$$
\leq \frac{2nM}{(1-\gamma)^3} \sum_{i=1}^{n} \sum_{a_i} d^{(t)}(s) \pi_i^{(t)}(a_i|s) \Delta_i^{(t)}(s,a_i)^2
$$

Lastly, we will bound the absolute value of Part D.

$$
|\text{Part D}| = \left| \frac{1}{1-\gamma} \sum_{i=1}^{n} \sum_{s} \left( d_{\widetilde{\theta}^{i,(t)}}(s) - d^{(t)}(s) \right) \sum_{a_i} \left( \pi^{(t+1)}(a_i|s) - \pi_i^{(t)}(a_i|s) \right) \overline{Q_i^{\widetilde{\theta}^{i+1,(t)}}}(s,a_i) \right|
$$

$$
\leq \frac{1}{(1-\gamma)^2} \sum_{i=1}^{n} \sum_{s} \left| d_{\widetilde{\theta}^{i,(t)}}(s) - d^{(t)}(s) \right| \sum_{a_i} \left| \pi^{(t+1)}(a_i|s) - \pi_i^{(t)}(a_i|s) \right|
$$

$$
\leq \frac{1}{(1-\gamma)^2} \sum_{i=1}^{n} \max_{s} \|\pi_{i,s}^{(t+1)} - \pi_{i,s}^{(t)}\|_1 \sum_{s} \left| d_{\widetilde{\theta}^{i,(t)}}(s) - d^{(t)}(s) \right|.
$$

From Corollary 34

$$
\sum_{s} \left| d_{\widetilde{\theta}^{i,(t)}}(s) - d^{(t)}(s) \right| \leq \frac{1}{1-\gamma} \max_{s} \left\| \pi_{\widetilde{\theta}^{i,(t)}}(a|s) - \pi^{(t)}(a|s) \right\|_1
$$

$$
\leq \frac{1}{1-\gamma} \max_{s} \sum_{i=1}^{n} \|\pi_{i,s}^{(t+1)} - \pi_{i,s}^{(t)}\|_1.
$$

Thus

$$|\text{Part D}| \le \frac{1}{(1-\gamma)^2} \sum_{i=1}^{n} \max_s \|\pi_{i,s}^{(t+1)} - \pi_{i,s}^{(t)}\|_1 \sum_s \left| d_{\widetilde{\theta}^{i,(t)}}(s) - d^{(t)}(s) \right|$$

$$\le \frac{1}{(1-\gamma)^3} \left( \sum_{i=1}^{n} \max_s \|\pi_{i,s}^{(t+1)} - \pi_{i,s}^{(t)}\|_1 \right) \left( \max_s \sum_{i=1}^{n} \|\pi_{i,s}^{(t+1)} - \pi_{i,s}^{(t)}\|_1 \right)$$

$$\le \frac{1}{(1-\gamma)^3} \left( \sum_{i=1}^{n} \max_s \|\pi_{i,s}^{(t+1)} - \pi_{i,s}^{(t)}\|_1 \right)^2$$

From Cauchy-Schwarz inequality

$$\left( \sum_{i=1}^{n} \max_s \|\pi_{i,s}^{(t+1)} - \pi_{i,s}^{(t)}\|_1 \right)^2 \le n \sum_{i=1}^{n} \max_s \left( \sum_{a_i} \left| \pi_i^{(t+1)}(a_i|s) - \pi_i^{(t)}(a_i|s) \right| \right)^2$$

$$= n \sum_{i=1}^{n} \max_s \left( \sum_{a_i} \pi_i^{(t)}(a_i|s) \left| \Delta_i^{(t)}(a_i|s) \right| \right)^2$$

$$\le n \sum_{i=1}^{n} \max_s \left( \sum_{a_i} \pi_i^{(t)}(a_i|s) \right) \left( \sum_{a_i} \pi_i^{(t)}(a_i|s) \Delta_i^{(t)}(a_i|s)^2 \right)$$

$$\le n \sum_{i=1}^{n} \max_s \sum_{a_i} \pi_i^{(t)}(a_i|s) \Delta_i^{(t)}(a_i|s)^2$$

$$\le n \sum_{i=1}^{n} \sum_{s,a_i} \pi_i^{(t)}(a_i|s) \Delta_i^{(t)}(a_i|s)^2.$$

Thus

$$|\text{Part D}| \le \frac{n}{(1-\gamma)^3} \sum_{i=1}^{n} \sum_{s,a_i} \pi_i^{(t)}(a_i|s) \Delta_i^{(t)}(a_i|s)^2$$

$$\le \frac{nM}{(1-\gamma)^3} \sum_{i=1}^{n} \sum_{s,a_i} d^{(t)}(s) \pi_i^{(t)}(a_i|s) \Delta_i^{(t)}(a_i|s)^2,$$

Combining the bounds on Part A,B,C,D we get

$$\widetilde{\Phi}(\theta^{(t+1)}) - \widetilde{\Phi}(\theta^{(t)}) = \text{Part A} + \text{Part B} + \text{Part C} + \text{Part D}$$

$$\ge \left( \frac{1}{2\eta} - 4\lambda \max_i |\mathcal{A}_i| M^2 - \frac{4M}{(1-\gamma)^2} - \frac{3nM}{(1-\gamma)^3} \right) \sum_{i=1}^{n} \sum_{s,a_i} d^{(t)}(s) \pi_i^{(t)}(a_i|s) \Delta_i^{(t)}(s, a_i)^2,$$

which completes the proof. □

**Lemma 27.** *Under the condition as in Lemma 24,*

$$\sum_{i=1}^{n} \sum_{s,a_i} d^{(t)}(s) \pi_i^{(t)}(a_i|s) \Delta_i^{(t)}(s, a_i)^2 \ge \frac{\eta^2}{9} \sum_{i=1}^{n} \sum_{s,a_i} d^{(t)}(s) \pi_i^{(t)}(a_i|s) f_i^{(t)}(s, a_i)^2$$

*Proof.* Recall from the definition of $\Delta_i^{(t)}(s, a_i)$:

$$\Delta_i^{(t)}(s, a_i) = \frac{\exp\left( \eta f_i^{(t)}(s, a_i) \right)}{Z_t^{i,s}} - 1.$$

Thus

$$\sum_{a_i} \pi_i^{(t)}(a_i|s)\Delta_i^{(t)}(s,a_i)^2 = \frac{1}{\left(Z_t^{i,s}\right)^2}\sum_{a_i}\pi_i^{(t)}(a_i|s)\left(\exp\left(\eta f_i^{(t)}(s,a_i)\right) - Z_t^{i,s}\right)^2$$

$$= \frac{1}{\left(Z_t^{i,s}\right)^2}\left[\sum_{a_i}\pi_i^{(t)}(a_i|s)\left(\exp\left(\eta f_i^{(t)}(s,a_i)\right) - 1\right)^2\right.$$

$$\left. - 2\sum_{a_i}\pi_i^{(t)}(a_i|s)\left(\exp\left(\eta f_i^{(t)}(s,a_i)\right) - 1\right)\left(Z_t^{i,s} - 1\right) + \sum_{a_i}\pi_i^{(t)}(a_i|s)\left(Z_t^{i,s} - 1\right)^2\right]$$

$$= \frac{1}{\left(Z_t^{i,s}\right)^2}\left[\sum_{a_i}\pi_i^{(t)}(a_i|s)\left(\exp\left(\eta f_i^{(t)}(s,a_i)\right) - 1\right)^2 - \left(Z_t^{i,s} - 1\right)^2\right]$$

Since $|e^x - 1| \geq \frac{|x|}{2}$ for $x \geq -1$, we have that

$$\sum_{a_i}\pi_i^{(t)}(a_i|s)\left(\exp\left(\eta f_i^{(t)}(s,a_i) - 1\right)\right)^2 \geq \sum_{a_i}\pi_i^{(t)}(a_i|s)\left(\frac{\eta}{2}f_i^{(t)}(s,a_i)\right)^2$$

$$\geq \frac{\eta^2}{4}\sum_{a_i}\pi_i^{(t)}(a_i|s)f_i^{(t)}(s,a_i)^2$$

Additionally, as is proved in Lemma 24,

$$Z_t^{i,s} = \sum_{a_i}\pi_i^{(t)}(a_i|s)\exp\left(\eta f^{(t)}(s,a_i)\right)$$

$$\leq \sum_{a_i}\pi_i^{(t)}(a_i|s)\left(1 + \left(\eta f^{(t)}(s,a_i)\right) + \left(\eta f^{(t)}(s,a_i)\right)^2\right)$$

$$\leq 1 + \eta^2\sum_{a_i}\pi_i^{(t)}(a_i|s)f_i^{(t)}(s,a_i)^2.$$

Thus

$$\sum_{a_i}\pi_i^{(t)}(a_i|s)\Delta_i^{(t)}(s,a_i)^2 = \frac{1}{\left(Z_t^{i,s}\right)^2}\left[\sum_{a_i}\pi_i^{(t)}(a_i|s)\left(\exp\left(\eta f_i^{(t)}(s,a_i)\right) - 1\right)^2 - \left(Z_t^{i,s} - 1\right)^2\right]$$

$$\geq \frac{1}{\left(Z_t^{i,s}\right)^2}\left[\frac{\eta^2}{4}\sum_{a_i}\pi_i^{(t)}(a_i|s)f_i^{(t)}(s,a_i)^2 - \left(\eta^2\sum_{a_i}\pi_i^{(t)}(a_i|s)f_i^{(t)}(s,a_i)^2\right)^2\right]$$

$$= \frac{1}{\left(Z_t^{i,s}\right)^2}\left(\eta^2\sum_{a_i}\pi_i^{(t)}(a_i|s)f_i^{(t)}(s,a_i)^2\right)\left(\frac{1}{4} - \eta^2\sum_{a_i}\pi_i^{(t)}(a_i|s)f_i^{(t)}(s,a_i)^2\right).$$

From Corollary 25

$$-\frac{1}{15} \leq \eta f_i^{(t)}(s,a_i) \leq \frac{1}{3}$$

$$\implies \eta^2\sum_{a_i}\pi_i^{(t)}(s,a_i)f_i^{(t)}(s,a_i)^2 \leq \frac{1}{9}.$$

Thus

$$\sum_{a_i}\pi_i^{(t)}(a_i|s)\Delta_i^{(t)}(s,a_i)^2\frac{1}{\left(Z_t^{i,s}\right)^2}\left(\eta^2\sum_{a_i}\pi_i^{(t)}(a_i|s)f_i^{(t)}(s,a_i)^2\right)\left(\frac{1}{4} - \eta^2\sum_{a_i}\pi_i^{(t)}(a_i|s)f_i^{(t)}(s,a_i)^2\right)$$

$$\geq \frac{1}{\left(Z_t^{i,s}\right)^2} \left(\eta^2 \sum_{a_i} \pi_i^{(t)}(a_i|s) f_i^{(t)}(s,a_i)^2\right) \left(\frac{1}{4} - \frac{1}{9}\right)$$

$$\geq \left(\frac{9}{10}\right)^2 \left(\frac{1}{4} - \frac{1}{9}\right) \left(\eta^2 \sum_{a_i} \pi_i^{(t)}(a_i|s) f_i^{(t)}(s,a_i)^2\right)$$

$$\geq \frac{\eta^2}{9} \sum_{a_i} \pi_i^{(t)}(a_i|s) f_i^{(t)}(s,a_i)^2,$$

Thus

$$\sum_{i=1}^n \sum_{s,a_i} d^{(t)}(s)\pi_i^{(t)}(a_i|s)\Delta_i^{(t)}(s,a_i)^2 \geq \frac{\eta^2}{9} \sum_{i=1}^n \sum_{s,a_i} d^{(t)}(s)\pi_i^{(t)}(a_i|s) f_i^{(t)}(s,a_i)^2$$

which completes the proof. $\qquad\square$

**Lemma 28.**

$$\mathtt{NE\text{-}gap}(\theta^{(t)}) \leq \frac{\sum_{i=1}^n \sum_{s,a_i} d^{(t)}(s)\pi_i^{(t)}(a_i|s) f_i^{(t)}(s,a_i)^2}{4\lambda} + \lambda \max_i |\mathcal{A}_i| M,$$

where $M = \sup_\theta \max_s \frac{1}{d_\theta(s)}$.

*Proof.* We will now prove the lemma.

$$d^{(t)}(s)\pi_i^{(t)} f_i^{(t)}(s,a_i)^2 = d^{(t)}(s)\pi_i^{(t)}(a_i|s) \left(\frac{1}{1-\gamma}\overline{A_i^{(t)}}(s,a_i) + \lambda\frac{1}{d^{(t)}(s)\pi_i^{(t)}(a_i|s)} - \frac{\lambda|\mathcal{A}_i|}{d^{(t)}(s)}\right)^2$$

$$= d^{(t)}(s)\pi_i^{(t)}(a_i|s) \left(\frac{1}{1-\gamma}\overline{A_i^{(t)}}(s,a_i) - \frac{\lambda|\mathcal{A}_i|}{d^{(t)}(s)}\right)^2 + \frac{\lambda^2}{d^{(t)}(s)\pi_i^{(t)}(a_i|s)} + 2\lambda\left(\frac{1}{1-\gamma}\overline{A_i^{(t)}}(s,a_i) - \frac{\lambda|\mathcal{A}_i|}{d^{(t)}(s)}\right)$$

$$\geq 4\lambda\left(\frac{1}{1-\gamma}\overline{A_i^{(t)}}(s,a_i) - \frac{\lambda|\mathcal{A}_i|}{d^{(t)}(s)}\right).$$

$$\implies \quad \frac{1}{1-\gamma}\overline{A_i^{(t)}}(s,a_i) \leq \frac{d^{(t)}(s)\pi_i^{(t)}(a_i|s) f_i^{(t)}(s,a_i)^2}{4\lambda} + \frac{\lambda|\mathcal{A}_i|}{d^{(t)}(s)}$$

$$\leq \frac{\sum_i \sum_{s,a_i} d^{(t)}(s)\pi_i^{(t)}(a_i|s) f_i^{(t)}(s,a_i)^2}{4\lambda} + \lambda \max_i |\mathcal{A}_i| M.$$

Thus from Lemma 13,

$$\mathtt{NE\text{-}gap}_i(\theta^{(t)}) \leq \frac{1}{1-\gamma}\max_{s,a_i}\overline{A_i^{(t)}}(s,a_i) \leq \frac{\sum_i \sum_{s,a_i} d^{(t)}(s)\pi_i^{(t)}(a_i|s) f_i^{(t)}(s,a_i)^2}{4\lambda} + \lambda \max_i |\mathcal{A}_i| M,$$

which completes the proof. $\qquad\square$

We are now ready to prove Theorem 7.

*Proof of Theorem 7.* From Lemma 26 we have that for

$$\eta \leq \min\left\{\frac{1}{15\left(\frac{1}{(1-\gamma)^2} + \lambda|\mathcal{A}_i|M\right)}, \frac{1}{4\left(4\lambda\max_i |\mathcal{A}_i|M^2 + \frac{4M}{(1-\gamma)^2} + \frac{3nM}{(1-\gamma)^3}\right)}\right\},$$

$$\widetilde{\Phi}(\theta^{(t+1)}) - \widetilde{\Phi}(\theta^{(t)}) \geq \frac{1}{4\eta}\sum_{i=1}^n \sum_{s,a_i} d^{(t)}(s)\pi_i^{(t)}(a_i|s)\Delta_i^{(t)}(s,a_i)^2.$$

From Lemma 27,

$$\widetilde{\Phi}(\theta^{(t+1)}) - \widetilde{\Phi}(\theta^{(t)}) \geq \frac{1}{4\eta} \sum_{i=1}^{n} \sum_{s,a_i} d^{(t)}(s)\pi_i^{(t)}(a_i|s)\Delta_i^{(t)}(s,a_i)^2$$

$$\geq \frac{\eta}{36} \sum_{i=1}^{n} \sum_{s,a_i} d^{(t)}(s)\pi_i^{(t)}(a_i|s)f_i^{(t)}(s,a_i)^2$$

Thus by telescoping we have

$$\frac{\sum_{t=0}^{T-1} \sum_{i=1}^{n} \sum_{s,a_i} d^{(t)}(s)\pi_i^{(t)}(a_i|s)f_i^{(t)}(s,a_i)^2}{T} \leq \frac{36\left(\widetilde{\Phi}(\theta^{(T)}) - \widetilde{\Phi}(\theta^{(0)})\right)}{\eta T}.$$

From Lemma 28,

$$\frac{\sum_{t=0}^{T-1} \texttt{NE-gap}(\theta^{(t)})}{T} \leq \frac{1}{4\lambda} \frac{\sum_{t=0}^{T-1} \sum_{i=1}^{n} \sum_{s,a_i} d^{(t)}(s)\pi_i^{(t)}(a_i|s)f_i^{(t)}(s,a_i)^2}{T} + \lambda \max_i |\mathcal{A}_i| M$$

$$\leq \frac{9\left(\widetilde{\Phi}(\theta^{(T)}) - \widetilde{\Phi}(\theta^{(0)})\right)}{\eta\lambda T} + \lambda \max_i |\mathcal{A}_i| M.$$

Specifically, set $\lambda = \frac{\epsilon}{2\max_i |\mathcal{A}_i| M}$ and $\theta^{(0)} = \mathbf{0}$, then for any

$$T \geq \frac{18\left(\widetilde{\Phi}(\theta^{(T)}) - \widetilde{\Phi}(\theta^{(0)})\right)}{(1-\gamma)\eta\lambda\epsilon} = \frac{36\max_i |\mathcal{A}_i|(\phi_{\max} - \phi_{\min})M}{(1-\gamma)\eta\epsilon^2}$$

$$\geq O\left(\frac{n\max_i |\mathcal{A}_i|(\phi_{\max} - \phi_{\min})M^2}{(1-\gamma)^4\epsilon^2}\right),$$

we have

$$\frac{\sum_{t=0}^{T-1} \texttt{NE-gap}(\theta^{(t)})}{T} \leq \frac{\epsilon}{2} + \frac{\epsilon}{2} = \epsilon,$$

which completes the proof. $\qquad\qquad\square$

## H   Smoothness Proofs

This section mainly focuses on the smoothness of $\Phi$ and $\widetilde{\Phi}$. We first state the smoothness results in Lemma 29 and Lemma 30. The auxiliary lemmas used during proof of the above two lemmas are stated in Lemma 31 and Appendix I.

**Lemma 29** (Smoothness of $\Phi(\theta)$)**.**

$$\|\nabla_\theta \Phi(\theta') - \nabla_\theta \Phi(\theta)\|_2 \leq \frac{6n}{(1-\gamma)^3}\|\theta' - \theta\|_2$$

*Proof.* From Lemma 31 we have that

$$\|\nabla_\theta \Phi(\theta') - \nabla_\theta \Phi(\theta)\|_2^2 = \sum_{i=1}^{n} \|\nabla_{\theta_i} \Phi(\theta') - \nabla_{\theta_i} \Phi(\theta)\|_2^2$$

$$\leq \sum_{i=1}^{n} \|\nabla_{\theta_i} \Phi(\theta') - \nabla_{\theta_i} \Phi(\theta)\|_1^2$$

$$\leq \sum_{i=1}^{n} \left(\frac{6}{(1-\gamma)^3} \sum_{i=1}^{n} \|\theta_i' - \theta_i\|_2\right)^2$$

$$= \frac{36n}{(1-\gamma)^6} \left(\sum_{i=1}^{n} \|\theta_i' - \theta_i\|_2\right)^2$$

$$\leq \frac{36n^2}{(1-\gamma)^6} \sum_{i=1}^{n} \|\theta_i' - \theta_i\|_2^2$$

$$= \frac{36n^2}{(1-\gamma)^6} \|\theta' - \theta\|_2^2,$$

thus

$$\|\nabla_\theta \Phi(\theta') - \nabla_\theta \Phi(\theta)\|_2 \leq \frac{6n}{(1-\gamma)^3} \|\theta' - \theta\|_2$$

$\square$

**Lemma 30** (Smoothness of $\widetilde{\Phi}(\theta)$)**.**

$$\left\| \nabla_\theta \widetilde{\Phi}(\theta') - \nabla_\theta \widetilde{\Phi}(\theta) \right\|_2 \leq \left( \frac{6n}{(1-\gamma)^3} + 2\lambda \max_i |\mathcal{A}_i| \right) \|\theta' - \theta\|_2$$

*Proof.* Since

$$\frac{\partial \sum_{i=1}^{n} \sum_{s,a_i} \log \pi_{\theta_i}(a_i|s)}{\partial \theta_{s,a_i}} = 1 - |\mathcal{A}_i| \pi_{\theta_i}(a_i|s)$$

we have that

$$\left\| \nabla_\theta \left( \sum_{i=1}^{n} \sum_{s,a_i} \log \pi_{\theta_i'}(a_i|s) - \sum_{i=1}^{n} \sum_{s,a_i} \log \pi_{\theta_i}(a_i|s) \right) \right\|_2^2 = \sum_{i=1}^{n} |\mathcal{A}_i|^2 \sum_s \sum_{a_i} \left( \pi_{\theta_i'}(a_i|s) - \pi_{\theta_i}(a_i|s) \right)^2$$

$$\leq \sum_{i=1}^{n} |\mathcal{A}_i|^2 \sum_s \|\pi_{\theta_{i,s}'} - \pi_{\theta_{i,s}}\|_1^2 \overset{\text{(Corollary 37)}}{\leq} 4 \sum_{i=1}^{n} |\mathcal{A}_i|^2 \sum_s \|\theta_{i,s}' - \theta_{i,s}\|_2^2 \leq 4 \max_i |\mathcal{A}_i|^2 \|\theta' - \theta\|_2^2$$

Thus

$$\left\| \nabla_\theta \widetilde{\Phi}(\theta') - \nabla_\theta \widetilde{\Phi}(\theta) \right\|_2$$

$$\leq \|\nabla_\theta \Phi(\theta') - \nabla_\theta \Phi(\theta)\|_2 + \lambda \left\| \nabla_\theta \left( \sum_{i=1}^{n} \sum_{s,a_i} \log \pi_{\theta_i'}(a_i|s) - \sum_{i=1}^{n} \sum_{s,a_i} \log \pi_{\theta_i}(a_i|s) \right) \right\|_2$$

$$\leq \left( \frac{6n}{(1-\gamma)^3} + 2\lambda \max_i |\mathcal{A}_i| \right) \|\theta' - \theta\|_2.$$

$\square$

**Lemma 31.**

$$\|\nabla_{\theta_i} \Phi(\theta') - \nabla_{\theta_i} \Phi(\theta)\|_1 \leq \frac{6}{(1-\gamma)^3} \sum_{i=1}^{n} \|\theta_i' - \theta_i\|_2$$

*Proof.*

$$\|\nabla_{\theta_i} \Phi(\theta') - \nabla_{\theta_i} \Phi(\theta)\|_1 = \frac{1}{1-\gamma} \sum_{s,a_i} \left| d_{\theta'}(s) \pi_{\theta_i'}(a_i|s) \overline{A_i^{\theta'}}(s,a_i) - d_\theta(s) \pi_{\theta_i}(a_i|s) \overline{A_i^\theta}(s,a_i) \right|$$

$$= \frac{1}{1-\gamma} \sum_{s,a_i} \left| d_{\theta'}(s) \pi_{\theta_i'}(a_i|s) \sum_{a_{-i}} \pi_{\theta_{-i}'}(a_{-i}|s) A_i^{\theta'}(s,a_i,a_{-i}) - d_\theta(s) \pi_{\theta_i}(a_i|s) \sum_{a_{-i}} \pi_{\theta_{-i}}(a_{-i}|s) A_i^\theta(s,a_i) \right|$$

$$\leq \frac{1}{1-\gamma} \sum_{s,a} \left| d_{\theta'}(s) \pi_{\theta'}(a|s) A_i^{\theta'}(s,a) - d_\theta(s) \pi_\theta(a|s) A_i^\theta(s,a) \right|$$

$$\leq \frac{1}{1-\gamma} \left( \sum_{s,a} |d_{\theta'}(s) \pi_{\theta'}(a|s) - d_\theta(s) \pi_\theta(a|s)| \left| A_i^{\theta'}(s,a) \right| + \sum_{s,a} d_\theta(s) \pi_\theta(a|s) \left| A_i^{\theta'}(s,a) - A_i^\theta(s,a) \right| \right)$$

$$\leq \frac{1}{1-\gamma} \left( \frac{1}{1-\gamma} \sum_{s,a} |d_{\theta'}(s) \pi_{\theta'}(a|s) - d_\theta(s) \pi_\theta(a|s)| + \max_{s,a_i} \left| A_i^{\theta'}(s,a) - A_i^\theta(s,a) \right| \right)$$

From Lemma 32 and Corollary 35, we have that

$$\|\nabla_{\theta_i}\Phi(\theta') - \nabla_{\theta_i}\Phi(\theta)\|_1 \leq \frac{3}{(1-\gamma)^3} \max_s \|\pi_{\theta'_s} - \pi_{\theta_s}\|_1$$

$$\leq \frac{3}{(1-\gamma)^3} \max_s \sum \|\pi_{\theta'_{i,s}} - \pi_{\theta_{i,s}}\|_1$$

From Corollary 37 we have that

$$\|\nabla_{\theta_i}\Phi(\theta') - \nabla_{\theta_i}\Phi(\theta)\|_1 \leq \frac{3}{(1-\gamma)^3} \max_s \sum \|\pi_{\theta'_{i,s}} - \pi_{\theta_{i,s}}\|_1$$

$$\leq \frac{6}{(1-\gamma)^3} \sum_{i=1}^{n} \|\theta_i - \theta_i\|_2. \qquad \square$$

# I   Some Useful Lemmas

**Lemma 32.**

$$\left| Q^{\theta'}(s,a) - Q^{\theta}(s,a) \right| \leq \frac{1}{(1-\gamma)^2} \max_s \|\pi_{\theta'_s} - \pi_{\theta_s}\|_1$$

$$\left| V^{\theta'}(s) - V^{\theta}(s) \right| \leq \frac{1}{(1-\gamma)^2} \max_s \|\pi_{\theta'_s} - \pi_{\theta_s}\|_1,$$

*and thus*

$$\left| A^{\theta'}(s,a) - A^{\theta}(s,a) \right| \leq \frac{2}{(1-\gamma)^2} \max_s \|\pi_{\theta'_s} - \pi_{\theta_s}\|_1$$

*Proof.* From performance difference lemma we have that

$$\left| Q^{\theta'}(s,a) - Q^{\theta}(s,a) \right|$$

$$= \left| \sum_{t=1}^{+\infty} \gamma^t \sum_{s'} \mathrm{Pr}^{\theta'}(s(t) = s'|s(0) = s, a(0) = a) \sum_{a'} (\pi_{\theta'}(a'|s') - \pi_{\theta}(a'|s')) Q^{\theta}(s',a') \right|$$

$$\leq \left| \sum_{t=1}^{+\infty} \gamma^t \sum_{s'} \mathrm{Pr}^{\theta'}(s(t) = s'|s(0) = s, a(0) = a) \sum_{a'} |\pi_{\theta'}(a'|s') - \pi_{\theta}(a'|s')| \left| Q^{\theta}(s',a') \right| \right|$$

$$\leq \left| \sum_{t=1}^{+\infty} \gamma^t \max_{s'} \sum_{a'} |\pi_{\theta'}(a'|s') - \pi_{\theta}(a'|s')| \frac{1}{1-\gamma} \right|$$

$$= \frac{1}{(1-\gamma)^2} \max_s \sum_a |\pi_{\theta'}(a|s) - \pi_{\theta}(a|s)|$$

$$= \frac{1}{(1-\gamma)^2} \max_s \|\pi_{\theta'_s} - \pi_{\theta_s}\|_1$$

Same argument also holds for $\left| V^{\theta'}(s) - V^{\theta}(s) \right|$, and thus

$$\left| A^{\theta'}(s,a) - A^{\theta}(s,a) \right| \leq \left| Q^{\theta'}(s,a) - Q^{\theta}(s,a) \right| + \left| V^{\theta'}(s) - V^{\theta}(s) \right| \leq \frac{2}{(1-\gamma)^2} \max_s \|\pi_{\theta'_s} - \pi_{\theta_s}\|_1.$$

$$\square$$

**Lemma 33.**

$$\frac{1}{1-\gamma} \sum_{s,a} (d_{\theta'}(s)\pi_{\theta'}(a|s) - d_{\theta}(s)\pi_{\theta}(a|s)) r(s,a) \leq \frac{1}{(1-\gamma)^2} \|r\|_\infty \max_s \|\pi_{\theta'_s} - \pi_{\theta_s}\|_1,$$

*where* $\|r\|_\infty = \max_{s,a} |r(s,a)|$.

*Proof.* For any reward function $r(s,a)$, we can define its value function $V^\theta(s)$ and $Q^\theta(s,a)$ correspondingly. Using performance difference lemma we have that

$$\frac{1}{1-\gamma} \sum_{s,a} \left( d_{\theta'}(s)\pi_{\theta'}(a|s) - d_\theta(s)\pi_\theta(a|s) \right) r(s,a)$$

$$= \sum_s \rho(s)(V^{\theta'}(s) - V^\theta(s))$$

$$= \frac{1}{1-\gamma} \sum_s d_{\theta'}(s) \sum_a \left( \pi_{\theta'}(a|s) - \pi_\theta(a|s) \right) Q^\theta(s,a)$$

$$\leq \frac{1}{(1-\gamma)^2} \|r\|_\infty \sum_s d_{\theta'}(s) \|\pi_{\theta'_s} - \pi_{\theta_s}\|_1$$

$$\leq \frac{1}{(1-\gamma)^2} \|r\|_\infty \max_s \|\pi_{\theta'_s} - \pi_{\theta_s}\|_1. \qquad \square$$

We have the following two corollaries for Lemma 33.

**Corollary 34.**

$$\frac{1}{1-\gamma} \sum_s |d_{\theta'}(s) - d_\theta(s)| \leq \frac{1}{(1-\gamma)^2} \max_s \|\pi_{\theta'_s} - \pi_{\theta_s}\|_1$$

*Proof.*

$$\frac{1}{1-\gamma} \sum_s |d_{\theta'}(s) - d_\theta(s)| = \max_{-1 \leq r(s) \leq 1} \frac{1}{1-\gamma} \sum_{s,a} (d_{\theta'}(s)\pi_{\theta'}(a|s) - d_\theta(s)\pi_\theta(a|s))r(s)$$

$$\leq \frac{1}{(1-\gamma)^2} \max_s \|\pi_{\theta'_s} - \pi_{\theta_s}\|_1. \qquad \square$$

**Corollary 35.**

$$\frac{1}{1-\gamma} \sum_s |d_{\theta'}(s)\pi_{\theta'}(a|s) - d_\theta(s)\pi_\theta(a|s)| \leq \frac{1}{(1-\gamma)^2} \max_s \|\pi_{\theta'_s} - \pi_{\theta_s}\|_1.$$

*Proof.*

$$\frac{1}{1-\gamma} \sum_s |d_{\theta'}(s)\pi_{\theta'}(a|s) - d_\theta(s)\pi_\theta(a|s)|$$

$$= \max_{-1 \leq r(s,a) \leq 1} \frac{1}{1-\gamma} \sum_{s,a} (d_{\theta'}(s)\pi_{\theta'}(a|s) - d_\theta(s)\pi_\theta(a|s))r(s,a)$$

$$\leq \frac{1}{(1-\gamma)^2} \max_s \|\pi_{\theta'_s} - \pi_{\theta_s}\|_1. \qquad \square$$

**Lemma 36.**

$$\sum_{a_i} \left( \pi_{\theta'_i}(a_i|s) - \pi_{\theta_i}(a_i|s) \right) f(a_i) \leq 2\|f\|_\infty \|\theta'_{i,s} - \theta_{i,s}\|_2$$

*Proof.* It suffices to show that

$$\left\| \nabla_{\theta_{i,s}} \sum_{a_i} \pi_{\theta_i}(a_i|s)f(a_i) \right\|_2 \leq 2\|f\|_\infty, \ \forall\theta,$$

then by Lagrange mean value theorem,

$$\sum_{a_i} \left( \pi_{\theta'_i}(a_i|s) - \pi_{\theta_i}(a_i|s) \right) f(a_i)$$

$$\leq \max_{t, \bar{\theta}=t\theta+(1-t)\theta'} \left\| \nabla_{\theta_{i,s}} \sum_{a_i} \pi_{\bar{\theta}_i}(a_i|s) f(a_i) \right\|_2 \|\theta'_{i,s} - \theta_{i,s}\|_2 \leq 2\|f\|_\infty \|\theta'_{i,s} - \theta_{i,s}\|_2.$$

Since

$$\frac{\partial \sum_{a_i} \pi_{\theta_i}(a_i|s) f(a_i)}{\partial \theta_{a_i,s}} = \pi_{\theta_i}(a_i|s)(f(a_i) - \bar{f}), \text{ where } \bar{f} = \sum_{a_i} \pi_{\theta_i}(a_i|s) f(a_i),$$

we have

$$\left\| \nabla_{\theta_{i,s}} \sum_{a_i} \pi_{\theta_i}(a_i|s) f(a_i) \right\|_2^2 = \sum_{a_i} \pi_{\theta_i}(a_i|s)^2 (f(a_i) - \bar{f})^2 \leq \sum_{a_i} \pi_{\theta_i}(a_i|s)^2 (2\|f\|_\infty)^2 \leq 4\|f\|_\infty^2,$$

which completes the proof. $\qquad\square$

**Corollary 37.** *(of Lemma 36)*

$$\|\pi_{\theta'_{i,s}} - \pi_{\theta_{i,s}}\|_1 \leq 2\|\theta'_{i,s} - \theta_{i,s}\|_2 \leq 2\|\theta'_i - \theta_i\|_2$$

*Proof.*

$$\|\pi_{\theta'_{i,s}} - \pi_{\theta_{i,s}}\|_1 = \max_{f: \|f\|_\infty \leq 1} \sum_{a_i} \left( \pi_{\theta'_i}(a_i|s) - \pi_{\theta_i}(a_i|s) \right) f(a_i) \leq 2\|\theta'_{i,s} - \theta_{i,s}\|_2. \qquad\square$$