# OpenReview forum: "On the Global Convergence Rates of Decentralized Softmax Gradient Play in Markov Potential Games"
_NeurIPS.cc/2022/Conference — NeurIPS 2022 Accept_

### Official Review · Reviewer_onZS · 2022-07-11

**Rating:** 5
**Confidence:** 5
**Soundness:** 3 good
**Presentation:** 3 good
**Contribution:** 3 good

**Summary:**

This paper mainly analyzes the global convergence rates of decentralized Softmax policy play in MPG in detail, including both policy gradient and natural policy gradient, with and without log-barrier regularization, which numerically validates that the log-barrier regularized algorithms are indeed more robust against becoming trapped near undesirable non-NE stationary points. The theoretical results illustrates the complexity bounds of MPG, and the natural gradient play outperforms gradient play counterparts.

**Questions:**

1) What is the connection between Softmax policy and Boltzmann policy, log-barrier and entropy regularization, and how are they related to some exploration methods?
2) The log-barrier method is introduced to avoid undesirable regions of policy space. What is the difference between this and the Trust region policy optimization based methods, e.g., MATRPO, HATRPO? How to illustrate the advantages of this regularization method?

**Ethics Review Area:**

["I don’t know"]

**Limitations:**

Yes, the authors detail the limitations of this work and possible future research works. It could be better if the fundamental reasons of these limitations can be explained.

**Strengths And Weaknesses:**

Strengths:
1) One major concern for this work is the innovation of the proof sketch, which is mainly based on the properties of smooth functions and inequality transformations.
The authors should highlight the difficulties and solutions in the proof process.
2) The authors give a comprehensive analysis of the convergence rate of the policy gradient methods in MPG. The theoretical results seems correct and fills the gap of Softmax policy play in MPG.
3) The authors provide detailed descriptions of each term in the theoretical result, which are helpful for understanding.

Weaknesses:
1) Compared to entropy regularization methods, and even some trust region based methods, the authors should emphasize the necessity for log-barrier regularization.
2) The authors should add some citations to illustrate the impact of Fisher information in line 57.
3) The theoretical analysis contains a large number of inequality derivations, but the author omits some sources of inequality derivation, such as Lemma 29, Lemma 31, etc., which are difficult to understand.
4) This article contains some notation errors and typos, such as the table in Introduction, the title of 3.2, etc.

---

> ### Author Response · Authors · 2022-08-02
> **Response to Reviewer onZS (Part 1)**
>
> We sincerely thank the reviewer for providing suggestions and critical feedback! In the revised version, we have fixed the typos and notation errors accordingly. We have also added the citation to fisher information as suggested by the reviewer. We address the reviewer's other comments as follows:
>
> ---*"One major concern for this work is the innovation of the proof sketch, which is mainly based on the properties of smooth functions and inequality transformations. The authors should highlight the difficulties and solutions in the proof process"*
>
> Thanks for the comment and suggestion. As the reviewer has pointed out, some of the analyses are indeed similar to the work by Agarwal et al, such as the asymptotic convergence proof as well as the convergence rate for $\log$-barrier gradient play. However, there are many results that use completely different analysis tools. For example, *the analysis of natural gradient play are very different both for the regularized case and the unregularized case, especially for the log-barrier regularization case, the result is highly non-trivial*. In Agarwal et al., the analysis on NPG leverages the unique existence of optimal value function $V^*$ so that similar analysis for mirror-descent can also carry over to NPG analysis, and thus obtain dimension-free convergence. However, in the multi-agent setting, there's no well-defined $V^*$ as NEs can be non-unique, thus, we need to further deploy the structure of the total potential function $\Phi$ to prove corresponding sufficient descent lemmas for NPG schemes, which to our knowledge is new and can serve as a contribution on its own merit. We apologize for not making this contribution clear in our current paper. We have revised our writing and highlighted this contribution. In the revised paper, Remark 2 and 3 are added to explain the technical novelty in the analysis. Reviewer gv2p also has similar concerns and we would like to refer the reviewer to "Response to Reviewer gv2p (Part 1)" for more details.
>
> ---*"Compared to entropy regularization methods, and even some trust region based methods, the authors should emphasize the necessity for log-barrier regularization"* and *"What is the difference between this and the Trust region policy optimization based methods, e.g., MATRPO, HATRPO? How to illustrate the advantages of this regularization method?"*
>
> Relevant discussion on the motivation of the choice of regularizer can be found in the paragraph *"Discussion on the choice of the regularizer"* in Section 5 of the revised paper. Here we quote part of the paragraph to explain why entropy regularization is not considered in this paper: "we adopt the $\log$-barrier instead of entropy regularization due to technical rather than practical considerations. Although entropy regularization achieves fast exponential convergence in single-agent learning, for multi-agent learning, we haven't been able to obtain results as strong as the $\log$-barrier regularization. Intuitively, the $\log$-barrier regularized gradient field repels the trajectory from regions with small $\pi_i(a_i|s)$ values (where the geometry becomes close to singular) more strongly, which enables us to obtain our current analysis. However, we emphasize that our result does not imply that log-barrier is better than entropy regularization in practice. It remains future work to determine whether entropy regularization, or other methods such as trust region-based methods, can achieve the same, or even better convergence rates." To further add to this discussion, some recent work by Cen et al. proves positive results for entropy regularization in *matrix potential game*, yet the generalization to MPG still remains open.
>
> Similarly, we are not claiming that $\log$-barrier is better than the TRPO-type of methods for handling undesirable regions of policy space in MPGs. We believe that TRPO algorithms (probably with some regularization that encourage exploration) should also work for MPGs, but they might require more complicated analysis. Indeed, it would be interesting and meaningful to develop TRPO-based methods for multiagent MPGs and other SGs. We apologize for leaving it as future work. We hope our work can attract more research interest from the RL community to develop various policy optimization-based methods for multi-agent RL, which will provide a deeper understanding of the pros and cons of different methods.

---

> > ### Author Response · Authors · 2022-08-02
> > **Response to Reviewer onZS (Part 2)**
> >
> > ---*"What is the connection between Softmax policy and Boltzmann policy, log-barrier and entropy regularization, and how are they related to some exploration methods?"*
> >
> > Softmax policy is different from Boltzmann policy because Boltzman implements the policy which follows the softmax of the Q function, whereas our softmax policy is the softmax of the parameter $\theta$ instead. We remark that our result does not imply that $\log$-barrier is better than implementing the Boltzmann policy or entropy regularization or other regularization approaches. Our results provide a starting point for analyzing softmax gradient methods for MPGs. Studying those different approaches and comparing their performance are interesting future directions.
> >
> > ---*"The theoretical analysis contains a large number of inequality derivations, but the author omits some sources of inequality derivation, such as Lemma 29, Lemma 31, etc., which are difficult to understand."*
> >
> > We thank the reviewer for reading carefully into the supplementary material. The proofs of Lemma 29 and Lemma 31 are provided directly after the statement of the lemmas. We apologize if the cross-referencing of lemmas leaves a misimpression that the proof is omitted. We have added some explanations in Appendix H (highlighted in blue) in the revised paper to make the proof structure clearer. If there is other confusion in the proofs, we are happy to further address them.
> >
> > ---*"This article contains some notation errors and typos, such as the table in Introduction, the title of 3.2, etc."*
> >
> > We thank the reviewer again for reading the paper carefully. We have fixed the notation errors and typos that we found in the revised paper. For the table in the Introduction, some of the convergence rates are not exactly the same as their original paper because, in order to make all the results more comparable, we re-derived the results in these papers using variables defined in this paper. This is stated in the caption of Table 1, which we quote here: "Note that the definition of $M$ is slightly different from the ''distribution mismatch coefficient'' $D_\infty$ defined in Agarwal et al (see more details in descriptions that follow Assumption 1). To make the complexity results more comparable, we slightly modify and re-derive the results in Agarwal et al., Mei et al, Leonardos et al and Zhang et al." We apologize for not making this highlighted enough. In the revised version, we have emphasized these sentences in italic font.

---

### Official Review · Reviewer_gv2p · 2022-07-11

**Rating:** 5
**Confidence:** 4
**Soundness:** 4 excellent
**Presentation:** 4 excellent
**Contribution:** 1 poor

**Summary:**

This paper analyzes the convergence of gradient play in Markov potential games where the players have softmax policies. Relying on recent advances in the analysis of policy gradients for softmax policies in single-agent RL, the authors first give asymptotic guarantees of convergence for both gradient play and natural gradient play and then subsequently provide finite-time convergence rates for both algorithms. These rates suffer due to a dependence on a non-uniform gradient-domination constant which may get arbitrarily large (as in the single agent case). To address this, the authors again continue as in the single-agent case and suggest doing gradient-play on a regularized version of the problem. When the amount of regularization is well chosen, the authors show that gradient-play from a fully random policy achieves fast convergence that is independent on the curvature constant 1/c but albeit with a worse dependence on the number of actions. They conclude with numerical experiments.


**Questions:**

1. Because of the Łojasiewicz Inequality it seems that all critical points of gradient-play are Nash equilibria of the game, but can there be saddle points in the potential function like those pointed out in [3] above?





**Limitations:**

Overall the paper is almost too straightforward an extension of prior works [1] and [3], and there does not seem to be anything particularly interesting about the combination. Making the exact contributions of this work clear would be helpful. Indeed the asymptotic global convergence guarantees and finite time rates are immediate from the potential structure and the Non-uniform Łojasiewicz Inequality. The idea of regularizing the problem seems to be a simple generalization of the idea in [1] to deal with the issue that 1/c might be arbitrarily large, but it is unclear if the multi-agent structure makes the derivation of the 1/c-less rate any more complex than in [1].

**Strengths And Weaknesses:**

This paper is very well written and easy to follow, though the results are not very surprising and the proof techniques and paper overall seem to follow from [1] in a very straightforward way. While results do not in general need to be surprising to be interesting, it is unclear what the contributions of this paper are beyond the explicit rates which follow from simple non-convex optimization arguments combined with the main property of the problem: namely the Łojasiewicz Inequality. Such analyses have been done before in [2],[3] albeit without the softmax structure, and the softmax structure is handled in exactly the same way as in [1].

Furthermore, the problem setting of Markov potential games is not very well motivated. Indeed, practical examples of when these games arise and why one might formulate the problem as a MPG as opposed to a single agent problem would greatly help.

[1] Agarwal et al. On the theory of policy gradient methods: Optimality, approximation, and distribution shift

[2] Leonardo et al. Global convergence of multi-agent policy gradient in markov potential games

[3] Zhang et al. Gradient play in multi-agent markov stochastic games: stationary points, convergence, and sample complexity

---

> ### Author Response · Authors · 2022-08-02
> **Response to Reviewer gv2p (Part 1)**
>
> We sincerely thank the reviewer for the critical feedback! We address the concerns raised by the reviewer as follows:
>
> ---*"...proof techniques and paper overall seem to follow from [1] in a very straightforward way"* and *"Overall the paper is almost too straightforward an extension of prior works [1] and [3],...Making the exact contributions of this work clear would be helpful."*
>
> We acknowledge that our work is greatly motivated and inspired by these exciting previous works (as well as others, e.g., Mei et al. 2020), however, we note that our work is different from them, not only in terms of the problem setting (softmax parameterization, MPG), but also in terms of the analysis techniques as well as the interpretation of the results. We apologize for not making them clear in the initial submission. We have revised our writing and highlighted the contribution and technical novelty in the submitted revision. Please also see our comment *"A Summary of Changes in the Paper Revision"* in the OpenReview page for the summary of the revision. We also provide more discussion below.
>
> Technical-wise, the analysis for the asymptotic convergence of unregularized policy gradient (PG) and the convergence rate for log-barrier  PG are indeed similar to the work by Agarwal et al. However, other results use very different analysis tools. In particular, *the analysis of natural policy gradient (NPG) are very different for both the unregularized and $\log$-barrier regularized cases, especially for the $\log$-barrier regularized case, the result is highly non-trivial*. In Agarwal et al., the analysis on NPG leverages the unique existence of optimal value function $V^*$ so that similar analysis for mirror-descent can also carry over to NPG analysis, and thus obtain dimension-free convergence. However, in the multi-agent setting, there's no well-defined $V^*$ as NEs can be non-unique with different potential values, thus, we need to further deploy the structure of the total potential function $\Phi$ to prove corresponding sufficient descent lemmas for NPG schemes. To our knowledge, this is new and can serve as a contribution on its own merit. In the revised paper, Remark 2 and Remark 3 are added to explain the technical novelty in the analysis.
>
> This difference in the analysis also results in the different convergence rates and their interpretation. For single-agent learning, unregularized NPG scheme can already enjoy dimension-free convergence, yet for the multi-agent learning case, unregularized NPG still suffers from slow convergence because of the dependence on $1/c$. We also corroborate our theoretical results with the numerical example (e.g., Figure 1) showing that unregularized NPG can get stuck at sub-optimal points for a very long time. This sharp contrast of the performance of unregularized NPG itself is also an interesting message provided by this paper.
>
> As for the difference with Leonardos et al., Zhang et al., although we all study MPG, we consider completely different policy parameterizations which lead to very different optimization landscapes and thus very different analysis techniques. The most significant difference is that, for direct parameterization considered by the two papers, there is an exact equivalence between NE and first-order stationary point, whereas in the softmax setting there exists undesirable stationary points that are non-NE: for instance, any deterministic policies are stationary points, regardless of whether they are NEs or not. This difference causes additional difficulties to our analysis, e.g., even asymptotic convergence to NE is not obvious. To clear up the confusion, we have added a new section *"Relationship between first-order stationary points and NEs"* (Section 3) in the revised paper.  Additionally, Leonardos et al. and Zhang et al. only considered gradient play whereas natural gradient play schemes were not considered in their work.

---

> > ### Author Response · Authors · 2022-08-02
> > **Response to Reviewer gv2p (Part 2)**
> >
> > ---*"...the problem setting of Markov potential games is not very well motivated. Indeed, practical examples of when these games arise and why one might formulate the problem as a MPG as opposed to a single agent problem would greatly help."*
> >
> > Indeed, MPG is only a special subset of the general stochastic games. Due to the space limit, in this paper we could not zoom into discussions on when a game could be formulated as a MPG. We refer the reviewer to Appendix B in Zhang et al [3] for more discussion on the necessary or sufficient conditions, as well as applications for MPG. One major reason for studying MPG is that the existence of potential function in MPG enables the possibility of learning NEs efficiently. But note that MPG is still substantially different from single-agent/centralized learning even with the existence of the potential function, which is more thoroughly discussed in Remark 1 *"Differences between MPG and single-agent/centralized MDP"* in the paper. As discussed in Remark 1, even for identical interest games, the decentralized independent policy structure prevents formulating the problem as a single-agent problem. When formulating the problem as a single agent problem using centralized policies, policy gradient methods require iterations that scale *exponentially* with the number of agents to converge; while for MPG, agents take decentralized policies and policy gradient methods scale *linearly* with the number of agents to converge. This is one major advantage of formulating it as a MPG rather than a single agent problem--to save iteration complexity (but as a trade-off optimality might be sacrificed). Nonetheless, we agree with the reviewer on the limitations of MPG. It is indeed an interesting and important future work to identify other special structures for SGs that enable the possibility of NE-learning and enjoy broad real-world applications in the meantime.
> >
> > ---*"Because of the Łojasiewicz Inequality it seems that all critical points of gradient-play are Nash equilibria of the game, but can there be saddle points in the potential function like those pointed out in [3] above?"*
> >
> > Thanks for asking the question. Firstly, we would like to clarify that not all critical points (first-order stationary points) are NEs (see explanation in Section 3 of the revised paper), thus the asymptotic convergence is nontrivial because there is no obvious guarantee why gradient methods do not converge to the critical points that are non-NEs. Sorry for causing the confusion, we have dedicated Section 3 to make this point clear. And yes, the local stability can vary between different types of NEs and we conjecture that the local stability results should carry over from Zhang et al. However, the main focus of the paper is to characterize the finite-time global convergence rate. Characterizing the local stability is left as one of the future works.

---

### Official Review · Reviewer_4yF8 · 2022-07-22

**Rating:** 6
**Confidence:** 4
**Soundness:** 4 excellent
**Presentation:** 3 good
**Contribution:** 3 good

**Summary:**

The authors focus on the analysis of various algorithms/dynamics for Markov Potential Games, using softmax parametrization. Softmax parametrization is challenging as it creates more fixed points than Nash equilibria. Also Markov Potential Games include multiple Nash equilibria as well so this adds an extra challenge. The authors provide rates for policy gradient with softmax parametrization (note that only asymptotical was known prior to this work from Agarwal et al), natural policy gradient with softmax parametrization (here only results for single agent were known and the authors use a nice trick from Mei et al) and finally policy and natural policy gradient with log-barrier functions. The log-barrier function allows the authors to get rid of some parameter $c$ that can be arbitrarily large and typically slows down policy gradient. Mathematically speaking, this constant $c$ is because of the new fixed points that are created in the boundary due to softmax.

**Questions:**

It feels that you cannot get rid of the mismatch coefficient in your rates. There are some other works though that use quantal response Nash idea and get rid of the mismatch coefficient. Is the existence of mismatch coefficient in your rates necessary even when you use log-barrier functions or you believe your analysis can be improved?

**Limitations:**

This is theoretical work

**Strengths And Weaknesses:**

Pros
The results are important, the authors add an important contribution related to softmax parametrization. Note that softmax parametrization is important for function approximation.

Cons
The techniques were more or less established before this work, the authors had to do a careful merge of the existing techniques and generalize the results to Markov Potential Games.

Typos: Section 3.2 title: Convergence rate, not fate.

---

> ### Author Response · Authors · 2022-08-02
> **Response to Reviewer 4yF8**
>
> Thank you for reading the paper carefully and appreciating our work! We have added more explanation on the contribution and innovation of our work in the paper revision compared with previous works, especially Agarwal et al., Leonardos et al., Zhang et al, (also see the discussion with Reviewer 99n7 and gv2p for more details). We address the review's other concerns and questions as follows:
>
> ---*"There are some other works though that use quantal response Nash idea and get rid of the mismatch coefficient. Is the existence of mismatch coefficient in your rates necessary even when you use log-barrier functions or you believe your analysis can be improved?"*
>
> The work by Cen et al. 2021 considers QRE and their result does not contain the distributional mismatch coefficient. However, they consider the two-player-zero-sum Markov game setting, whose problem structures and properties (e.g. existence of unique Nash value functions) are very different from the MPG setting considered in this paper. As for the MPG setting, all the current results, e.g. Leonardos et al, Zhang et al,  are not able to get rid of this constant to the best of our knowledge. It is indeed an interesting open question whether this distributional mismatch coefficient a proof artifact or something fundamental for learning in MPG.

---

> > ### Comment · Reviewer_4yF8 · 2022-08-07
> > **Thank you for the response**
> >
> > I would like to thank the authors for their response. I feel the paper should get in, I cannot increase my score though because I feel it should get in as a poster (not spotlight). I will continue the discussion with fellow reviewers and AC. If needed later, I will increase my score.

---

### Official Review · Reviewer_99n7 · 2022-07-26

**Rating:** 5
**Confidence:** 5
**Soundness:** 3 good
**Presentation:** 3 good
**Contribution:** 2 fair

**Summary:**

This paper studies the behavior of the gradient play when the policies are modeled via softmax surrogate.
Gradient play can be either first-order (policy gradient) or second-order (natural policy-gradient) method.
In contrast to the previous results that aim to establish results for the direct parametrization $p(a|s)=\theta_{s,a}$, this work builds upon a series of results for Markov Potential Games and tries to establish rates of convergence to the corresponding Markov Nash equilibria.

For this setting, the authors provide three kind of results for the solution concept of Markov Nash equilibria in the generalization of identical interest RL game, known as Markov Potential Games.

1) The asymptotic convergence of gradient play with softmax regularization
2) Convergence rates which include some ``trajectory''-dependent constants
3) The impact of log-barrier regularization in gradient play in the rates

**Questions:**

See the weakness section.

Additionally, I would like a clarification: Why in the single agent MDP the last row/first column does not copy automatically the result of the multi-agent case?





**Ethics Review Area:**

["I don’t know"]

**Limitations:**

Non-applicable

**Strengths And Weaknesses:**

Summary of contributions.
In this general context, the main contributions of the manuscript under review can be stated as follows:
(1a) In principle this is the first work that offers convergence rates for softmax policy gradient method.
(1b) It should be mentioned that as far as the asymptotic convergence results have been already established last two years. Therefore, for the standard gradient play via softmax, I will encounter as novelty the convergence rate part

(2a) The authors, via a manipulation of gradient dominance/ Polyak-Ljovasiewitz armamentum, guarantee fast convergence rates for the case of MPGs with Isolated equilibria.
(2b) However, their convergence rate include a trajectory dependent constant which can be exponential worse.  It would be much more interesting if authors gives us examples where the worst case is achieved and how in practice this constants fluctuate.

(3a) Finally, log-barrier is used for softmax regularization to get independent constant-style results.
(3b) From applicability perspective, it is unclear how much harder is the inclusion of log-barrier to achieve the necessary result.
(3c) Again I would like to notice that it is not the first work that applies log-barrier regularization in RL problems.







Evaluation and recommendation.

I believe that the core of the authors’ results is interesting: in particular, the derivation of convergence rates for softmax has been an interesting question recent years. However the positioning of the main draft, does not explain at all which are the important non-trivial observations which meshes well with existing results in the field, and which can provide further insights in the behavior of policy gradient methods to extract the promised results.

I would expect from the authors a much more extensive analysis of the challenges & the burdens that the current submission tackles and what was the main intuition that leads authors to the usage of log-barrier regularization. At the end of the day, the usage of such strong regularization was a natural solution due to some collection of issues or just a matter of lack?

At the same time, the paper’s writing, presentation and positioning leaves a lot to be desired:
a fair number of results are presented in a confusing or misleading way (especially in
regard to the state of the art), the writing style obfuscates the analysis and statements, and
the authors’ prose is often inconsistent with the results actually being proven.

Ultimately, I believe that a thoroughly rewritten version of this paper could be ultimately publishable
in NeurIPS proceedings, but the submitted manuscript is not yet there. For this reason, I recommend a
majorly enlightening revision with the understanding that the paper should be rewritten from the ground
up with emphasis on clarity, readability and ease of comparison to the existing literature on
the topic.

Indeed, the line between asking for a border line acceptance and borderline rejecting the paper outright while encouraging resubmission at a later point is a blurry one.
I am recommending a Border-Acceptance  because I find sufficient merit in the paper’s contributions; however, these should be stated in a way that is consistent with the Conference standards, an endeavor which will require a careful rewriting

---

> ### Author Response · Authors · 2022-08-02
> **Response to Reviewer 99n7 (Part 1)**
>
> We sincerely thank the reviewer for appreciating the contribution of our work! The reviewer's summary of contribution is indeed consistent with the contribution of this work. Thank you for your effort in reviewing and understanding our work. We also appreciate the reviewer's suggestion on improving the clarity of the paper and have incorporated the comments into the paper revision. The summary of the changes is listed in our Comment *"A Summary of Changes in the Paper Revision"*. Since the reviewer explicitly suggested a revision, we repeat the summary of the changes here in case it is not easy for reviewers to identify the specific comment. Note that the revised parts are highlighted in blue in the submitted revision.
>
> * Clarify the technical contribution and differences compared with Agarwal et al., in the introduction and also expand the explanation on the technical contribution and differences in the main text; See *Remark 2* in Section 4.1 and *Remark 3* in Section 5.2 of the revised paper.
> * Restructure the original Section 3 *"Unregularized gradient play"* to be the new Section 3 *"Relationship between first-order stationary point and Nash equilibrium"*  and Section 4 *"Unregularized gradient play"*. The new section 3 focuses on presenting the non-uniform Łojasiewicz condition and its implication on the relationship between the stationary points and Nash equilibria. The new structure clears confusion on the relationship between stationary points and Nash equilibria and also helps us present the necessity of our asymptotic convergence results as well as the technical challenges and differences compared to the literature on the single agent setting and the MPG gradient play with direct parameterized policies.
> * Emphasize the technical motivation and discussion of the choice of our $\log$-barrier regularizer; see the paragraph *"Discussion on the choice of the regularizer"* in Section 5 of the revised paper.
>
> Due to the limited time for the rebuttal, we could not do a major revision, so we mainly focus on clarifying a few main confusions raised by all reviews. We will continue to revise the writing for the final version of the paper.

---

> > ### Author Response · Authors · 2022-08-02
> > **Response to Reviewer 99n7 (Part 2)**
> >
> > Below, we provide a detailed response to the reviewer's comments.
> >
> > ---*"...the positioning of the main draft, does not explain at all which are the important non-trivial observations which meshes well with existing results in the field, and which can provide further insights in the behavior of policy gradient methods to extract the promised results."*
> >
> > We thank the reviewer again for appreciating the novelty of establishing the convergence rate of softmax policy gradient methods. Indeed, establishing the convergence rate and understanding the limitation of the softmax policy gradient through these rates are the major goal and novel contributions of this work. We apologize if our initial submission did not highlight clearly enough the non-trivial observations and technical differences. We hope our revised paper has clarified these confusions. As the revised paper still needs to follow the page limit, below we provide a more detailed explanation of our analysis of the softmax policy gradient methods in 4 main steps. For each step, we explain the technical derivation of the results by focusing on illustrating which results follow the standard derivation in literature and which are non-trivial results developed by this paper. If we miss any point or there is any part that is still unclear, we are happy to further address.
> >
> > 1. We first study the relationship between stationary points and NEs, and show that there are 'good' stationary points that are NEs and 'bad' stationary points that are not. The differentiation of the good and bad ones is established by the non-uniform Łojasiewicz condition (See Section 3 in the revised version), whose proof is inspired by the single agent counterparts in Mei et al., yet the result is slightly different due to the problem setting (see Line 171-176 of the revised paper).
> > 2. It is not directly obvious whether gradient methods will converge to the 'good' stationary points, thus the natural next step is to first prove asymptotic convergence to NE, where we apply analysis similar to Agarwal et al. to give an affirmative answer.
> > 3. Further, the convergence rate for unregularized algorithm is built upon the asymptotic convergence as we need the property of $c(\theta^*)=1$ for any NE $\theta^*$ to show $c:=\inf_t c(\theta^{(t)})>0$. For gradient play, the proof follows relatively straightforward from non-uniform Łojasiewicz condition and standard non-convex smooth optimization result on gradient norm. Because the Łojasiewicz condition is non-uniform, a trajectory dependent constant $c$ appears in the denominator of the bound. For *natural gradient play*, existing proof techniques are *not directly* applicable thus *new* analysis tools are introduced, which we give a more detailed description in *Remark 2 "proof sketch and novelty"* in Section 4.1 in the revised paper.
> > 4. As the term $c$ can be very close to zero, which slows down the convergence, it motivates us to use regularization approaches to eliminate the dependence on $c$. Here we use $\log$-barrier regularization (reasons explained in the next paragraph). For gradient play, our proof resembles the proof in Agarwal et al. For *natural gradient play*, the proof is *novel* and *different* from single-agent analyses for NPG, and a detailed proof sketch is given in *Remark 3* in Section 5.2 of the revised paper.

---

> > > ### Author Response · Authors · 2022-08-02
> > > **Response to Reviewer 99n7 (Part 3)**
> > >
> > > --- *"...what was the main intuition that leads authors to the usage of log-barrier regularization...the usage of such strong regularization was a natural solution due to some collection of issues or just a matter of lack?"*
> > >
> > > Relevant discussion on the motivation of the choice of regularizer can be found in the paragraph *"Discussion on the choice of the regularizer"* in Section 5 of the revised paper. Here we quote part of the paragraph to explain why entropy regularization is not considered in this paper: "we adopt the $\log$-barrier instead of entropy regularization due to technical rather than practical considerations. Although entropy regularization achieves fast exponential convergence in single-agent learning, for multi-agent learning, we haven't been able to obtain results as strong as the $\log$-barrier regularization. Intuitively, the $\log$-barrier regularized gradient field repels the trajectory from regions with small $\pi_i(a_i|s)$ values (where the geometry becomes close to singular) more strongly, which enables us to obtain our current analysis. However, we emphasize that our result does not imply that log-barrier is better than entropy regularization in practice. It remains future work to determine whether entropy regularization, or other methods such as trust region-based methods, can achieve the same, or even better convergence rates." To further add to the remark, our attempt for gradient play with entropy regularization cannot get rid of the $1/c$ dependence, but the assumption of isolated stationary points can be removed. We also discussed with the other researchers interested in similar topics about natural gradient play under entropy regularization. Based on our discussion, we only obtained a suboptimal rate with $O(1/\epsilon^6)$. Thus the major reason we do not study entropy regularization in this paper is that we failed to obtain results that are comparable to $\log$-barrier regularization. However, this might be a proof artifact and does not imply that $\log$-barrier is better than entropy regularization. In fact, some recent work by Cen et al. *"Independent Natural Policy Gradient Methods for Potential Games: Finite-time Global Convergence with Entropy Regularization"* proves positive results for entropy regularization in *matrix potential game*, yet the generalization to MPG still remains open.
> > >
> > >
> > > ---*"...their convergence rate includes a trajectory dependent constant which can be exponential worse. It would be much more interesting if authors gives us examples where the worst case is achieved and how in practice this constants fluctuate."*
> > >
> > > There are more systematic studies on the worst case of the constant in literature, e.g. Li et al. as cited in our paper, which is why we didn't spend too much space on this topic. However, our numerical example does corroborate the theoretical findings, where we can see that unregularized algorithms can get stuck at non-NE stationary points (e.g., Figure 1(a), 1(d)), resulting in plateaus in the learning curve.
> > >
> > > ---*"Why in the single agent MDP the last row/first column does not copy automatically the result of the multi-agent case?"*
> > >
> > > Thanks for the question. Yes, it can be directly copied, however, the special structure of single-agent learning might enable better rates, thus our derivation might be suboptimal for single-agent setting, which is why we put 'unknown' for that entry.

---

### Author Response · Authors · 2022-08-02
**A Summary of Changes in the Paper Revision**

We sincerely thank all the reviewers for providing constructive feedback on our work and thank the associate chair for handling our paper. The reviewers' comments provide us with good ideas on improving the clarity of the paper and we have been starting to incorporate the comments into the paper revision. Due to the limited time for the rebuttal, we could not do a thorough revision in the submitted revision, so we mainly focus on clarifying a few main confusions raised by all reviews. We will continue to revise the writing for the future version of the paper. The revised parts are highlighted in blue in the paper. Here is a summary of the changes.

* Clarify the technical contribution and differences compared with Agarwal et al. in the introduction and also expand the explanation on the technical contribution and differences in the main text; See *Remark 2* in Section 4.1 and *Remark 3* in Section 5.2 of the revised paper.

* Restructure the original Section 3 *"Unregularized gradient play"* to be the new Section 3 *"Relationship between first-order stationary point and Nash equilibrium"*  and Section 4 *"Unregularized gradient play"*. The new section 3 focuses on presenting the non-uniform Łojasiewicz condition and its implication on the relationship between the stationary points and Nash equilibria. The new structure clears confusion on the relationship between stationary points and Nash equilibria and also helps us present the necessity of our asymptotic convergence results as well as the technical challenges and differences compared to the literature on the single agent setting and the MPG gradient play with direct parameterized policies.

* Emphasize the technical motivation and discussion of the choice of our $\log$-barrier regularizer; see the paragraph *"Discussion on the choice of the regularizer"* in Section 5 of the revised paper.

---

### Author Response · Authors · 2022-08-08
**Comment for Discussion Period**

Dear Reviewers,

Thanks again for your effort and valuable feedback during the review process. We have made our response and revised the paper accordingly. Since the deadline for reviewer-author discussion is approaching, we would like to kindly remind you to please let us know if your concerns have been resolved, or if you still have any questions that we can address during this period. (Our sincere apologies if you have already sent the feedback. We really appreciate your comments!) Thank you very much for your understanding!

---

### Meta-Review · Area_Chair_jb93 · 2022-08-27

**Recommendation:** Accept
**Confidence:** Certain

**Metareview:**

The reviewers appreciate the contribution of this paper to the theory of tabular MARL in MPGs, namely convergence rates of PG and NPG with and without a log-barrier regularizer. There were some concerns regarding clarity of the writing (in particular the textual descriptions of the theory and its implications) and of the contributions (and their distinction from existing results). There were also some concerns regarding the novelty of the proof techniques. Nevertheless, there's consensus that the paper makes important contributions.

The authors have started to address some of the reviewer feedback in their revision, and are encouraged to continue to do so.

**Award:**

No

---

### Decision · Program_Chairs · 2022-09-14

Accept